# COVID-19 amplified racial disparities in the US criminal legal system

Brennan Klein[1,2 ✉], C. Brandon Ogbunugafor[3,4,5,6,7 ✉], Benjamin J. Schafer[8], Zarana Bhadricha[9], Preeti Kori[9], Jim Sheldon[10], Nitish Kaza[1], Arush Sharma[1], Emily A. Wang[11,12,13], Tina Eliassi-Rad[1,5,6,14,15], Samuel V. Scarpino[1,5,6,10,14,15,16 ✉] & Elizabeth Hinton[2,8,13,17,18 ✉]

The criminal legal system in the USA drives an incarceration rate that is the highest on the planet, with disparities by class and race among its signature features[1–3]. During the first year of the coronavirus disease 2019 (COVID-19) pandemic, the number of incarcerated people in the USA decreased by at least 17%—the largest, fastest reduction in prison population in American history[4]. Here we ask how this reduction influenced the racial composition of US prisons and consider possible mechanisms for these dynamics. Using an original dataset curated from public sources on prison demographics across all 50 states and the District of Columbia, we show that incarcerated white people benefited disproportionately from the decrease in the US prison population and that the fraction of incarcerated Black and Latino people sharply increased. This pattern of increased racial disparity exists across prison systems in nearly every state and reverses a decade-long trend before 2020 and the onset of COVID-19, when the proportion of incarcerated white people was increasing amid declining numbers of incarcerated Black people[5]. Although a variety of factors underlie these trends, we find that racial inequities in average sentence length are a major contributor. Ultimately, this study reveals how disruptions caused by COVID-19 exacerbated racial inequalities in the criminal legal system, and highlights key forces that sustain mass incarceration. To advance opportunities for data-driven social science, we publicly released the data associated with this study at Zenodo[6].

Mass incarceration in the USA is distinguished by striking racial disparities and a rate of imprisonment that surpasses that of all other nations, with 2.12 million people in prisons and jails in 2019 (refs. 1–3,7–10). Owing to a combination of structural inequities and discriminatory enforcement, Black and Latino people are more likely to be stopped by police[11], held in jail pre-trial[12], charged with more serious crimes[13] and sentenced more harshly than white people[14,15]. These practices have made Black men in the USA six times as likely and Latino men 2.5 times as likely to be incarcerated as white men[16,17].

In this study, we demonstrate how the COVID-19 pandemic—which produced the largest, most rapid single-year decrease in prison population in US history—amplified existing inequities in the nation's criminal legal system[4]. Across nearly every state and federal prison system, we observe a convergent pattern: a substantial decrease in the overall number of people incarcerated (by approximately 200,000), but a meaningful increase in the proportion of incarcerated Black, Latino and other non-white people. We conclude that sentencing patterns are a central mechanism driving the racial disparity.

The trend we identify represents a substantial deviation from patterns preceding the pandemic. Before COVID-19, incarcerated Black people accounted for a declining share of the total prison population: roughly 41.6% of people incarcerated in state prisons were Black in March 2013, and by March 2020 this number had fallen to 38.9%—a decline of 2.7 percentage points over seven years. During the height of COVID-19 closures, from March 2020 to November 2020, this percentage increased by 0.9 points, erasing much of the progress over the last decade (Figs. 1b and 2 and Supplementary Fig. 2; see Supplementary Fig. 16 for comparison between effects among non-white versus Black populations). The trend we observe at the national level is reproduced exactly among states with the highest Black and Latino populations, and persists in some form in nearly every other state.

Data reporting methods on racial demographics in prisons have made it difficult for researchers to disentangle the various mechanisms driving observed disparities in incarcerated populations. We manually assembled and validated a dataset covering all 50 US states, the District of Columbia and the Federal Bureau of Prisons to both quantify the

[1]Network Science Institute, Northeastern University, Boston, MA, USA. [2]Institute on Policing, Incarceration & Public Safety, The Hutchins Center for African & African American Research, Harvard University, Cambridge, MA, USA. [3]Department of Ecology and Evolutionary Biology, Yale University, New Haven, CT, USA. [4]Public Health Modeling Unit, Yale School of Public Health, New Haven, CT, USA. [5]Santa Fe Institute, Santa Fe, NM, USA. [6]Vermont Complex Systems Center, University of Vermont, Burlington, VT, USA. [7]Department of Chemistry, Massachusetts Institute of Technology, Cambridge, MA, USA. [8]Department of History, Yale University, New Haven, CT, USA. [9]College of Engineering, Northeastern University, Boston, MA, USA. [10]Roux Institute, Northeastern University, Boston, MA, USA. [11]SEICHE Center for Health and Justice, Yale School of Medicine, New Haven, CT, USA. [12]Department of Medicine, Yale School of Medicine, New Haven, CT, USA. [13]Justice Collaboratory, Yale Law School, New Haven, CT, USA. [14]Khoury College of Computer Sciences, Northeastern University, Boston, MA, USA. [15]The Institute for Experiential AI, Northeastern University, Boston, MA, USA. [16]Department of Health Sciences, Northeastern University, Boston, MA, USA. [17]Department of African American Studies, Yale University, New Haven, CT, USA. [18]Yale Law School, New Haven, CT, USA. ✉e-mail: b.klein@northeastern.edu; brandon.ogbunu@yale.edu; s.scarpino@northeastern.edu; elizabeth.hinton@yale.edu

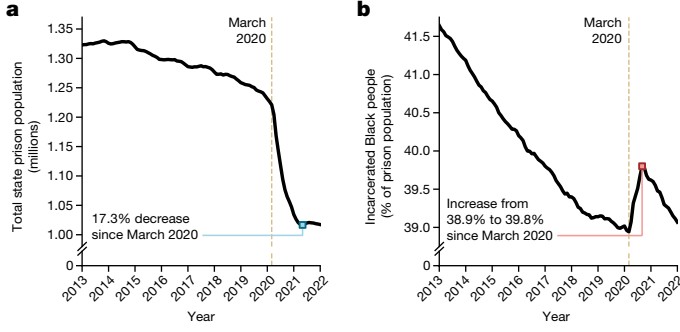

**Fig. 1 | Dynamics of the US prison population. a**, Total number of incarcerated people in the USA from January 2013 to January 2022. **b**, Total percentage of incarcerated Black people, as reported by states' Departments of Correction. According to data from the US census, Black people account for 13.4% of the total population[43]. This plot includes data from 49 states and the District of Columbia—data from Michigan are excluded as the state reports only "white" and "nonwhite" as race categories.

widening racial disparity observed during the first year of the COVID-19 pandemic and uncover its plausible causes. The result of this newly assembled, public dataset—comprising over 9,000 records across more than 20 years—is a view into the dynamics of prison populations before, during and after the onset of the pandemic. We publicly released the data associated with this study at Zenodo[6].

Overall, the number of incarcerated people decreased markedly in 2020. However, we show that the magnitudes of these declines were not equal by race, especially for incarcerated Black people (Fig. 3). We estimate that nearly 15,000 fewer Black people would have been incarcerated in January 2021 if the racial disparities we observe were not present (Supplementary Table 6). We discuss this observed disparity and related observations in light of the ethics of public health interventions, national debates about the future direction of policing and incarceration, and the importance of data infrastructure in responsible public policy. These discussions highlight how sentencing and other policies that seem to be 'race blind' can nonetheless lead to outcomes that are skewed by race[18]. We speculate that our findings transcend the influence of COVID-19, and discuss how large-scale disruptions can have a clear, quantifiable signature on extant inequalities.

## Declining incarcerated populations

The population of people incarcerated in US state prisons decreased by at least 17% between March 2020 and July 2021, from approximately 1.23 million to 1.02 million (Fig. 1a). A decrease in prison population occurred in every state, and, in most, started in early to mid-April 2020 (see Supplementary Fig. 1 for a state-by-state look at prison populations over time). This nationwide trend persisted despite stark differences in state-level trends pre-2020: For instance, many states entered 2020 with a steadily declining prison population (for example, total incarcerated population in Massachusetts fell from 11,403 in January 2013 to 8,292 in January 2020—declining 4.4% per year on average; in South Carolina: 22,146 in January 2013 to 18,041 in January 2020, an average of 2.9% decrease per year; in California: 134,534 in January 2013 to 124,027 in January 2020, an average of 1.1% decrease per year; and so on), others had relatively stable prison populations (for example, Virginia: 29,875 in 2013, 29,233 in 2020; Georgia: 56,951 in 2013, 55,218 in 2020) and many had growing prison populations before COVID-19 (for example, in Idaho, there were 8,030 people incarcerated in January 2013 and 9,502 in January 2020, increasing on average 2.5% per year; in Montana: 2,452 in January 2013 to 2,806 in January 2020, increases of 2.0% per year, on average; Supplementary Fig. 1). Nevertheless, we see large reductions

in the overall prison population across every state in the USA during the pandemic, with magnitudes ranging from a 5.8% reduction in Nebraska to 37.2% in New Jersey, and the median state's prison population falling by 18% of pre-pandemic levels. In Supplementary Table 2, we detail the scale and timing of each state's population decline.

As of January 2022, several states' prison populations continued to decrease (Arizona, Massachusetts, Washington, Louisiana and Pennsylvania, among others) and did so steadily throughout the pandemic. Other states' prison populations dropped sharply in the early months of the pandemic but saw their prison populations begin to increase again by January 2021 (Montana, Idaho, Iowa, Utah, Nebraska, West Virginia and California, among others) or by July 2021 (Wisconsin, Ohio, Texas, Connecticut, Rhode Island, Kentucky and so on). In Supplementary Fig. 1, we plot time series for each state's prison population over the past several years. Additionally, in Supplementary Table 1, we give an overview for each state's approach for reporting prison population statistics, along with how we collected each state's data.

## Changing racial demographics in prisons

Despite an overall decline in the total incarcerated population during the pandemic, there was an increase in the proportion of incarcerated Black people (Fig. 1b). This increase in racial disparity occurred nationally and in nearly every state, transcending vast differences in approach to crime and incarceration. In Fig. 2, we show the percentage of incarcerated Black people across 12 states (see Supplementary Fig. 2 for these trends in every state and the Federal Bureau of Prisons). However, the spike in the proportion of Black people in prison was temporary in most states, eventually returning to pre-pandemic levels by the end of 2021. We explore possible explanations for this reversal in subsequent sections, but the most likely reason is that the pace of prison admissions—which typically have a lower Black–white racial disparity than the overall incarcerated population[19]—began to approach pre-pandemic rates in early 2021.

Although the national trend we identify in Fig. 1 (that is, an abrupt increase in the proportion of Black people incarcerated) occurred in most state-level prison systems, there were meaningful differences that suggest possible mechanisms behind the disparity (Fig. 2 and Supplementary Figs. 2–4). In Fig. 2, we highlight several examples of state-level variability in the proportion of Black people incarcerated; for instance, trends in states such as Georgia, Kentucky and Texas resemble the shape seen nationally, whereas states such as Connecticut and Delaware saw an already-increasing trend in the percentage of incarcerated Black people increase even faster after March 2020. Five states—Maine, Maryland, Missouri, Oregon and Wyoming—are the only prison systems in the USA that do not clearly conform to the pattern we see across the country (a few states in Supplementary Fig. 2—for example, Missouri and Oklahoma—technically fit our criteria for exhibiting this trend but only weakly). These five states have a combined incarcerated population that amounts to roughly 5% of the national total and offer important insights into the underlying mechanisms behind the trends we see nationally. Namely, each of these states has either a relatively small proportion of incarcerated Black people or a prison system with fewer people with shorter-term (for example, fewer than 2 years) sentences compared to nationwide averages. We will show that the latter is probably the more powerful force contributing to the overall nationwide trend.

Ultimately, these observations lead us to outline three explanations that could bring about the trends from Fig. 1: who is admitted to prison; who is released from prison; and who remains in prison. These proposed mechanisms demonstrate different levers through which the pandemic may have influenced the racial composition of the incarcerated population, and dovetail with existing research on the dynamics of the American carceral state.

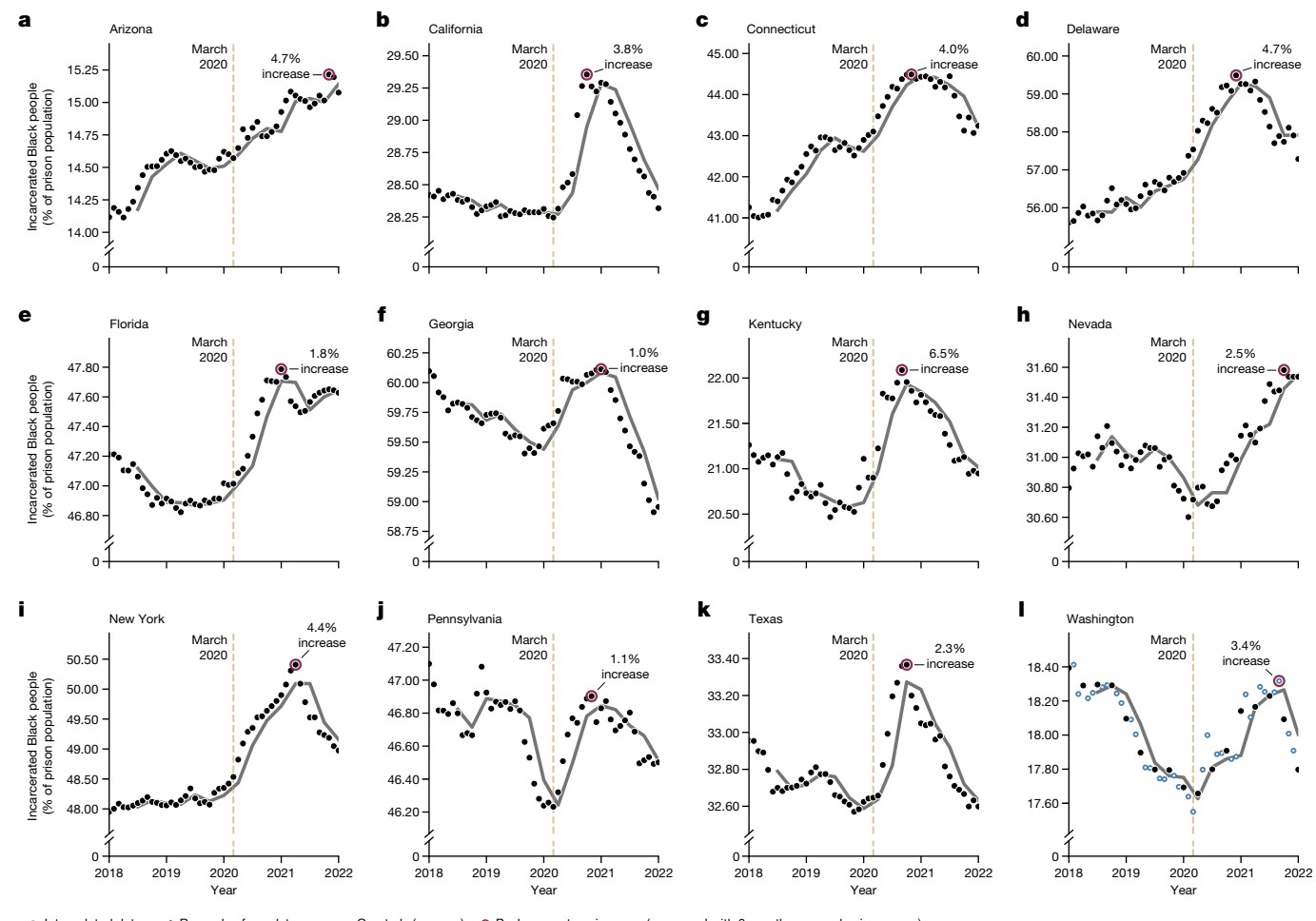

**Fig. 2 | Percentage of incarcerated Black people across 12 representative states.** In some states, the percentage of incarcerated Black people had been decreasing over the past several years. In others, this percentage had been increasing. Across a variety of pre-2020 trends, we see the same general trend across the USA: during the pandemic, incarcerated Black people accounted for an even larger share of the total prison population than in previous years.

**a–l,** We plot this trend in Arizona (**a**), California (**b**), Connecticut (**c**), Delaware (**d**), Florida (**e**), Georgia (**f**), Kentucky (**g**), Nevada (**h**), New York (**i**), Pennsylvania (**j**), Texas (**k**) and Washington (**l**)—states with consistent, frequent data reporting. Although each panel has a different vertical scale, the inset text indicates the peak percentage increase relative to 6-month pre-pandemic averages for each state.

## Mechanisms of disparity

Consider a time series of a state's prison population that does not notably change over several years. For this to occur, there needs to be approximately the same number of admissions and releases. For the demographic makeup of the prison population to remain stable, the relative number of admissions and releases by race also needs to be roughly equivalent over time. If there are sustained periods with more admissions (or releases) of a certain demographic, that will skew the overall distribution of the prison population.

Understanding the dynamics of admissions, releases and sentencing offers us a path towards identifying and isolating potential mechanisms that could bring about a steadily declining rate of incarceration of Black individuals (seen for nearly a decade before the pandemic), and the subsequent spike in the proportion of incarcerated Black people during the COVID-19 pandemic. Namely, the observed spike in Fig. 1b must be due to a disparity in who was admitted to prison during the pandemic, who was released or a combination of both.

## Admissions

In every state except Nebraska, courts closed at the beginning of the pandemic. These closures substantially reduced or altogether halted admissions into prisons for several months, starting around April 2020

(refs. 20–23). The Virginia Department of Corrections acknowledges the causal effect of court closures on the state's incarcerated population in their 2020 Annual Report[24], "The reduction in [average daily population] is directly attributed to the suspension of intake due to COVID-19". Similarly, a spokesperson for the Michigan Department of Corrections estimated that half of the reductions in incarcerated population were due to a decline in new admissions from courts and county jails[25].

On the basis of admissions data from 18 states, we estimate that the total monthly admissions to prison fell to about 30% of pre-pandemic averages by May or June 2020 (Fig. 4). This reduction in admissions provides a potential mechanism behind the sharp increase in the percentage of incarcerated Black people in Fig. 1b. Specifically, systematic racial differences occurring in monthly prison admissions during this period could drive changing disparity in the demographics of the incarcerated population. However, data from the 18 states presented in Fig. 4c actually show the reverse (that is, the percentage of Black individuals admitted to prison fell even lower than the corresponding rate for admissions of white individuals). Although we do see abrupt spikes in the percentage of Black individuals admitted to prison in a few states (for example, Wisconsin and Texas), this proposed mechanism seems not to be widespread enough to explain the nationwide trends we observe.

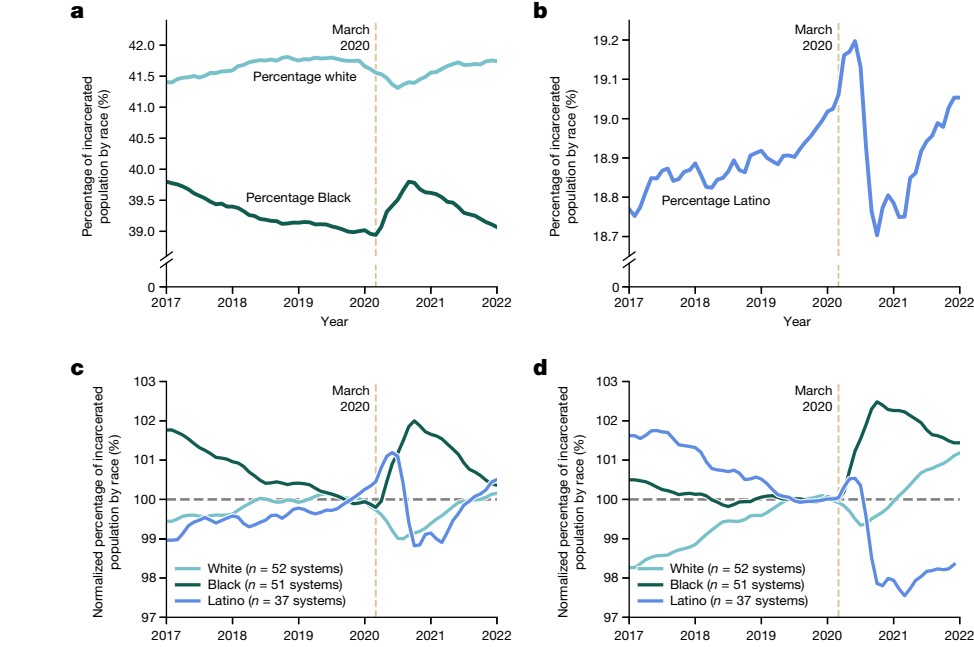

**Fig. 3 | Comparison of Black, white and Latino incarcerated populations over time. a**, Percentage of incarcerated population who are Black and white. Note especially that the effect size of the demographic changes during the COVID-19 pandemic are more pronounced in the incarcerated Black population (see Supplementary Fig. 16 for additional comparisons). **b**, Percentage of incarcerated Latino people. **c**, Percentage of white, Black and Latino incarcerated populations, normalized by the average value between March 2019 and February 2020. **d**, Percentage of white, Black and Latino incarcerated populations, normalized by the slope of each curve between March 2019 and February 2020.

Data from Florida offer another example of how changes to court proceedings influence prison admissions and the racial distribution of people admitted to prison. In Supplementary Fig. 18 we plot monthly trial statistics from circuit criminal defendants in Florida; after March 2020, we see sharp declines in the number of disposed defendants in Florida Circuit Criminal Courts, as well as the percentage of filed defendants that become disposed. Amid these declines, we see an abrupt increase in the percentage of cases that were dismissed before trial (that is, defendants whose charges were dropped). In Supplementary Fig. 19, we report that an increased proportion of the defendants with pre-trial case dismissals were white in the months after the start of the pandemic.

Prison admissions may also decline owing to policy changes or disruptions to a common source of prison admissions: county jails. Although there continues to be poor standards for reporting and maintaining these kinds of data, this potential source of prison admissions is important for a nationwide story of mass incarceration during the COVID-19 pandemic. Despite variability in admissions playing a role in the racial distribution of incarcerated populations, changing disparity in admissions alone does not seem to be widespread enough to account for the nationwide trends we observe in Fig. 1b. As we will see in the following section, a similar story emerges when looking at the demographics of people released from prison.

## Releases

In an effort to reduce the risk of severe acute respiratory syndrome coronavirus 2 (SARS-CoV-2) transmission, several states enacted policies designed to de-densify prisons. Depending on the state, these directives came from executive orders from the governor, state legislatures or governing boards. In Utah, for example, policies around releases are designed, approved and implemented by the Board of Pardons and Parole—an entirely separate entity from the courts and the Department of Corrections. According to the Board of Pardons and Parole, incarcerated people who are eligible for early release needed to

be already characterized as a non-violent offender, be within 90 days of release (this was later extended to 180 days[26]) and have an approved address to stay at after their release.

In Arkansas, an authorization from Governor Hutchinson (Executive Orders 20-06 and 20-16 (ref. 27)) made 1,243 incarcerated people eligible for early release as of 30 April 2020. Those deemed eligible needed to have a parole plan in place, be medically screened (that is, tested and screened for symptoms of COVID-19) and undergo final approval by the Arkansas Department of Corrections director to be released. In Supplementary Section 4.3 and Supplementary Fig. 23, we show that disproportionately more white people were released in Arkansas through this effort. The racial disparity in who was released by Governor Hutchinson's orders is due to the overlap between the state's release eligibility criteria and the racial differences in sentence classification—a tension we discuss further in the section in the Methods entitled Release policy data.

In Fig. 4d, we plot estimates of the nationwide change in monthly releases as a percentage of pre-pandemic values. What we see is that, despite efforts to reduce prison density through targeted releases, the rate of prisoner release was lower during much of the pandemic. At its lowest value (between February and May 2021), the number of people released from prison each month reached nearly 70% of pre-pandemic values. In the absence of changing admission patterns, this decline in releases should have led to an increase in the total incarcerated population in the USA, which is the opposite of the pattern we see in Fig. 1a. Therefore, we can conclude rather strongly that changing release rates did not drive the reduction in the incarcerated population during the pandemic. We also do not find meaningful differences in the relative number of releases by race during this time period; if anything, these data suggest that during the early months of the COVID-19 pandemic, Black people accounted for a higher percentage of monthly releases compared to pre-pandemic averages. As was the case for demographics of prison admissions, disparities in the monthly releases are unlikely to be driving the trends in Fig. 1b. However, data on prison releases

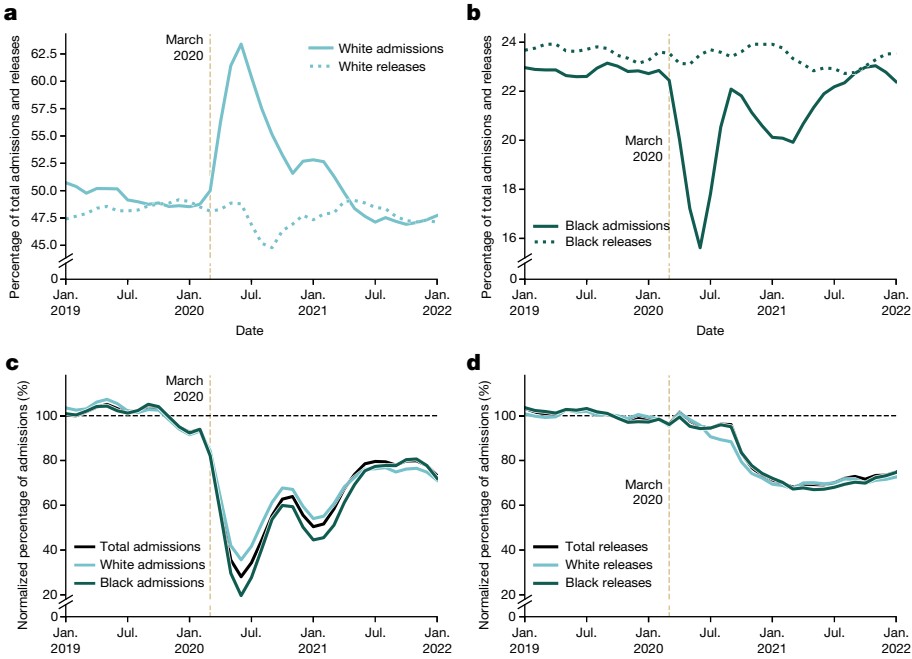

**Fig. 4 | Comparison of admissions and releases by race for 18 states.** Although data on monthly admissions to and releases from prison are less readily available than prison population data, we can nevertheless highlight the average dynamics of 18 states' data. **a**, Percentage of total monthly admissions (solid) and releases (dotted) for individuals who are white. **b**, Percentage of total monthly admissions (solid) and releases (dotted) for individuals who are Black. **c**, Normalized comparison of the change in monthly white, Black and total admissions. **d**, Normalized comparison of the change in monthly white, Black and total releases. Note: the vertical scales in **a** and **b** are different, whereas **c** and **d** share the same vertical scale.

point to an important, related process underlying the demographic patterns in incarcerated populations; in the next section, we focus less on those released from prison in any given month, but rather, those who remain.

## Sentencing

On the basis of demographic data from 18 state prison systems, racial disparities in admissions and releases alone are not able to explain the broad trends observed nationally (Fig. 4). In fact, if these were the only factors influencing prison population demographics, we would expect the opposite effect seen in Fig. 1b, as we observe a large increase in the proportion of admissions of white individuals after the start of the pandemic, amid large decreases in the proportion of admissions of Black individuals and relatively commensurate rates of releases. There are examples of individual states that show sudden increases in the relative amount of Black people admitted to prison at the start of the pandemic (see Texas, for example, in Supplementary Fig. 20). Similarly, there are examples of large-scale releases causing an abrupt increase in the percentage of incarcerated Black people (see a recent example in January 2022 in data from the Federal Bureau of Prisons; Supplementary Fig. 9). Nevertheless, we do not see these factors as being anywhere near as influential as disparities in sentencing of people already incarcerated at the start of the pandemic.

In short, the most important factor underlying the dynamics in Fig. 1b is related to differences in the average sentence length of incarcerated people by race. As a statistical observation this point is quite simple: provided there are differences in the average length of prison sentence by race (for example, the average incarcerated Black person serving a longer prison sentence than the average white incarcerated person; see Texas as an example in Supplementary Fig. 22) and sustained reductions in new admissions (as in Fig. 4c), then we will expect to see the effect observed in Fig. 1b. In addition to that basic mechanism, one can imagine factors that would exacerbate and/or attenuate the size and timing of the spike. These include: new or atypical patterns in prison admissions by race (relative to averages before the decline in admissions), or new or atypical changes in prison releases by race.

By casting sentencing differences as the driver behind the observations in this study, we are able to better understand why the main effect in Fig. 1b is so pronounced among incarcerated Black people and less so (although still present) when looking at incarcerated Latino people (Fig. 3). Illinois and Texas offer two particularly powerful examples that show increases in their proportion of incarcerated Black people. As one would expect given our proposed mechanism, the median sentence length for incarcerated Black people in each of these states is higher than that of white people. However, when we compare the median sentence lengths between incarcerated Black and Latino people, we find high Black–Latino overlap in Illinois but high white–Latino overlap in Texas. That is, white people in Illinois serve shorter sentences on average than Black and Latino people, but in Texas, white and Latino people serve shorter sentences than incarcerated Black people. According to the mechanism proposed above, we would expect this baseline difference in sentencing lengths to produce pandemic-related spikes in the percentage of Black and Latino people in Illinois and produce spikes only in the percentage of incarcerated Black people in Texas. This, in fact, is what we observe (Supplementary Fig. 7).

The policies, societal disruptions and behavioural changes that emerged following the onset of the COVID-19 pandemic amplified existing and long-standing racial disparities in the US carceral system. Consistent with other research, our findings show that disparities in sentencing by race are core to maintaining structural inequalities in incarcerated populations[14,19].

## Discussion

After declining steadily for the past decade, the percentage of Black and other non-white incarcerated people increased sharply during 2020, a trend that was present in almost every prison system across the country. To identify the mechanisms behind this increasing racial

disparity, we collected and validated a dataset that includes state-level information on police encounters, court proceedings and incarcerated populations. To obtain such granular information across all 50 states, the District of Columbia, and the Federal Bureau of Prisons, we manually collected data from individual Departments of Corrections and filed numerous Freedom of Information Act requests (Methods).

Before the COVID-19 pandemic, racial disparities in admissions were smaller than disparities in the prison population, as recent trends show a migration towards a class-driven disparity, with lower-educated white people steadily increasing in their rate of admission[28]. In a sense, courts had been serving as an instrument for decreasing the racial disparity in prisons before the pandemic; for example, the Black–white disparity in prison admissions is typically a ratio of 2:1, whereas it is closer to 6:1 for the total incarcerated population[19] (see ref. 29 for a recent exploration of several factors underlying this trend). Thus, when court proceedings or transfers from county jails are disrupted (that is, admissions are reduced), the racial disproportionality in the total prison population accelerates, as observed in Figs. 1 and 2 and Supplementary Figs. 1 and 2. These dynamics, happening within prison systems nationwide, that see incarcerated Black and other non-white people sentenced for longer periods of time on average[14], led to the abrupt nationwide increase in the percentage of incarcerated Black people, starting in March 2020. Differences in the length of sentence by race seem to be a key factor in producing the trends from Fig. 1b, but this effect will then be compounded if—in addition to overall decreases in admissions—there are also sudden changes in the typical distribution of the race of people admitted into prisons, which we see, for example, in Texas during the summer of 2020 (Supplementary Fig. 20b).

Understanding the role that racial disparities in sentencing play in producing the trends from Fig. 1b is key for making predictions about how sudden societal disruptions or policy changes in the future may impact prison population demographics (for example, continued pandemic, Supreme Court decisions, widespread social protests and so on). These findings can, in turn, help inform policy reform efforts. The sentencing disparity mechanism described in this work is even useful for explaining the dynamics behind the five states that did not conform to the overall national trend in Fig. 1b (Maine, Maryland, Missouri, Oregon and Wyoming; Supplementary Fig. 4). These five states maintain prison systems that incarcerate, on average, fewer people under shorter (less than 2-year) sentences, according to data from the National Corrections Reporting Program[30] (differences explored in Supplementary Figs. 5 and 6). This observation that states with fewer short-term prison sentences did not show the same racial disparity we found nationally has two subtle but important consequences. First, it suggests that a key reason why the disparities emerge is due to releases of incarcerated people who served shorter-term sentences (without a corresponding amount of admissions). This makes sense, because on any given day, a randomly selected person being released from prison is likely to have been sentenced for a shorter time period. Second, if white people are more likely to serve shorter sentences, then an overall reduction in the amount of people serving shorter-term prison sentences means there are fewer people serving shorter-term sentences who could be 'eligible' to drive the main effect in Fig. 1.

Although racial disparities in sentence lengths seem to be the most robust explanation behind the trend in Fig. 1b, we want to avoid disregarding the potential effects that racial disparities in prison admissions could have played during the COVID-19 pandemic. For example, there is another well-known mechanism through which court closures could have affected different states' relative rates of Black and Latino prison admissions during the pandemic (Supplementary Fig. 18): relative increases in pre-trial case dismissals (Supplementary Fig. 18d) and pre-trial plea deals. Plea deals in particular have long been demonstrated to result in a disproportionate number of Black defendants spending time in prison[13,31,32]. Interruptions in court proceedings may have contributed to the increased Black and Latino representation in prison populations by: reducing the increasingly large flux of admissions of white individuals to prison; amplifying processes—pre-trial case dismissals and pre-trial plea deals—that are long understood to be a leading contributor to disparities in judicial outcomes for Black individuals. Disruptions in the typical, pre-pandemic court proceedings also offer a compelling explanation as to why (as seen in Fig. 1b) we see the reversion to pre-2020 levels, starting in early 2021: the reduction in admissions stopped and, in most states, the total incarcerated population began to increase once again (Fig. 2 and Supplementary Fig. 1).

Beyond disparities in sentencing and admissions, the COVID-19 pandemic provided several specific challenges that shaped release patterns. Maintaining the largest and most expansive prison system in the world is a major challenge to public health, especially in the context of infectious diseases[4,33]. In particular, severely overcrowded conditions have presented a public health threat during the COVID-19 pandemic[34]. The physical and administrative structure of prisons provided constraints on ways to quarantine incarcerated people and de-densify congregate settings[34–38]. In recognition of these circumstances, several states enacted policies and initiated executive orders to release individuals who they deem eligible[23]. As a public health intervention, decarceration is a highly effective way to mitigate outbreaks inside and outside prisons[4,34,36–41]. During the pandemic, criteria for decarceration differed from state to state, but often included factors such as the age of the incarcerated person and the offence for which they were convicted (for example, non-violent drug offenders)[42]. We were able to quantify disparities in some states' efforts to de-densify prisons (for example, in Arkansas; see Supplementary Section 4.3 and Supplementary Fig. 23), which suggests that even decarceration policies widely understood to be consistent with effective and ethical public health practice (and that are assumed to be 'race blind') are susceptible to existing structural and racial inequalities. Moreover, one of the most important consequences of disparities in releases is not only about who is released, but who is left behind: the increase in the proportion of incarcerated Black and other non-white people translates to their being at a heightened risk of exposure to SARS-CoV-2.

Taken together, our findings reveal that the pandemic provided a 'stress test' for the criminal legal system. In engineering, stress tests involve exposing an apparatus to extreme conditions to reveal its fragilities; under these conditions, it can be easier to uncover the mechanisms that govern it. Using a range of data sources, we have argued that COVID-19 amplified underlying racial disparities in the carceral state. As is the case with many complex systems, the dynamics of prison populations are defined by interactions between multiple actors that, in combination, create unexpected or troubling results. In response to these findings, society has an ethical obligation to act, and reform sentencing practices and the broader criminal legal system towards more equitable ends.

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

## Methods

### State prison populations over time

Time series data about states' prison populations over time were collected manually through scraping Departments of Corrections websites, as well as direct requests to state officials through public record requests (for example, Freedom of Information Act requests and so on). For every state in our dataset, we sought the most temporally resolved data as possible. We collected population data at either weekly, monthly, quarterly or, for some states, yearly levels. The most common form of data we were able to collect is the number of people incarcerated at a given time in a given state, on a monthly timescale. In Supplementary Table 1, we link to the data source for every state in our dataset, and in Supplementary Section 1, we show how the prison population of every state has changed over time.

We compared the data collected here to data from other organizations that report statistics about the US prison population—the Bureau of Justice Statistics and the Vera Institute for Justice[44]—and find high overlap between all three of the datasets. In Supplementary Section 3, we identify areas in which our data differ from those of the Bureau of Justice Statistics, and we offer an explanation for why we are confident in accuracy of our approach (for example, in several cases, we received the data directly from the states' Departments of Corrections, through public records requests).

For every state in this dataset, the total prison population includes both male and female incarcerated people (something that is not always the case in studies about the US carceral system, which so often focuses on male incarcerated people). In Alaska, New Mexico, Vermont and California, "Transgender", "Other" or "Non-Binary" are also listed as gender categories, although this practice is not widely adopted in reporting statistics about the incarcerated population. In 27 states, incarcerated race statistics are separated by "male", "female" and "total", and further characterizing the interaction between race and sex in biases in admissions and releases during the COVID-19 pandemic remains future work.

### State policy data

**Court closures and reduced admissions.** Qualitative data on the closure and reopening of all 50 state court systems were collected primarily through the administrative orders and/or press releases of each state system's Supreme or Superior Court or chief judicial officer as well as through local news coverage. Most states suspended all in-person proceedings with the exception of limited emergency matters between 12 March and 20 March 2020. Several states that adopted policies early in this period issued increasingly strict guidance as the pandemic worsened. New Jersey, for example, suspended new trials on 12 March and issued a 2-week suspension on municipal court proceedings on 14 March before finally suspending all proceedings (with emergency exceptions) on 15 March. In addition to closing judicial buildings and suspending proceedings, most court closures also extended statute of limitations and filing deadlines owing to pandemic disruption. A handful of states, Pennsylvania and Texas among them, permitted or encouraged courts to begin conducting remote proceedings in their initial closure orders, although the adoption of remote proceedings was not widespread in this initial lockdown stage.

Court reopening policies were more heterogeneous than the initial closures, although trials remained suspended in most states through at least early summer 2020 (and in most cases substantially later). The earliest such policies appeared at the beginning of April 2020, with most aimed at giving regional and local judges discretion to begin hearing proceedings remotely (for example, Louisiana, Massachusetts, Florida and Iowa, among others). A substantially larger group of states adopted reopening guidelines between late April and mid May, many of which allowed essential judicial staff to return to offices following new public health guidance while also maintaining remote proceedings and expanding the number of non-trial proceedings that courts

could conduct remotely. Further reopenings and the resumption of limited in-person proceedings took place in many states throughout June, July and August 2020, although trial proceedings remained suspended. Notably, several states, especially those that adopted phased reopening plans, restricted in-person proceedings and further delayed trial resumption with the autumn–winter 2020–2021 COVID-19 surge. In many states, most administrative orders restricting court operations have at the time of publishing been rescinded, although others, California notably among them, still retain certain accommodations including the option for remote proceedings.

**Release policy data.** Data on COVID-19 release policies, when they existed, were collected from states' individual corrections and prison bureau systems, governors' executive orders and local news coverage. Fifteen states did not adopt any official release policy, although our data nevertheless show that there were still reductions in the overall prison population during the pandemic in all of these states. The remaining 35 states adopted policies with varying degrees of specificity and effectiveness, although many overlapped in their broadest contours, allowing consideration for early release to be granted to incarcerated people at increased public health risk (either due to age or underlying health condition) and for those nearing parole and/or the end of their prison sentences.

Almost all states with such policies did, however, adopt a restriction preventing the release of those incarcerated for violent crimes or sex offences. North Dakota was an outlier in this regard. Of the 120 people the state initially released from prison in March 2020, 14 were serving time for violent crime convictions and 11 were convicted of sex offences. New York's release policy was notably more restrictive (on paper at least) than that of many other states—only those incarcerated for "non-criminal technical parole violations" were eligible for COVID release. As an example of one state's release policy, we include below an excerpt from the Virginia Department of Corrections' policy on releases[45], from 24 April 2020.

"The Director of the Department of Corrections is authorized to consider early release for individuals with less than one year left to serve while the COVID-19 emergency declaration is in effect. Offenders convicted of a Class 1 felony or a sexually violent offense are not eligible for consideration. The exact number of individuals eligible for early release consideration will change depending on the length of the emergency declaration order. The [Department of Corrections] will identify those that are eligible for consideration using the procedures it has developed to ensure public safety and will notify offenders who are to be released under the early release plan. A diagnosis of COVID-19 is not a release factor.

The following Early Release Criteria will be utilized in considering an incarcerated person for early release pursuant to legislation:

- Release Date: The inmate's Good Time Release Date must be calculated and verified in order for the incarcerated to be considered.
- Inmate Medical Condition: The inmate's medical condition will be considered.
- Offense History: By legislative mandate, early release does not apply to inmates convicted of a Class 1 felony or a sexually violent offense. Consideration for early release will be based on the seriousness of the current offense, in descending order as follows: Non-violent Offense, Felony Weapons Offenses, Involuntary Manslaughter, Voluntary Manslaughter, Robbery, Felony Assault, Abduction, Murder, Sex Offense.
- Viable Home Plan: The incarcerated person must have a documented approved home plan to be considered.
- Good Time Earning Level: The inmate's current good time earning level must be I or II to be considered.
- No Active Detainers: Inmates must have no active detainer to be considered.
- No Sexually Violent Predator Predicate Offenses: Inmates convicted of one or more sexually violent offenses established in §37.2-903 of the Code of Virginia are not eligible pursuant to legislation.

• Recidivism Risk: Inmates must have a risk of recidivism of medium (5-7) or low (1-4), as identified by the validated COMPAS instrument, to be considered."

Note especially the inclusion of the COMPAS risk assessment tool, which is used in court systems across the USA as a way of quantifying an offender's likelihood of reoffending (recidivism). Over the past several years, we have seen a growing body of scholarly work devoted to identifying problematic and harmful racial and economic biases that arise when algorithmic risk assessment tools are used in practice[46–51]. COMPAS, in particular, has been the subject of a number of studies that take a critical look at the effectiveness—and ethics—of these risk assessment tools in the justice system[47,52]; in one study, COMPAS was found to predict recidivism 61% of the time, but at the same time, Black people were almost twice as likely to be labelled as high risk for reoffending but not actually reoffend[52].

Further research is needed to quantify demographic patterns in the incarcerated individuals who were released across different states, and because there was such high heterogeneity in different states' policies, it remains an open question whether we will see the same broad, systematic racial differences among the people who were released. However, as has been the case throughout the COVID-19 pandemic, the heterogeneity of policy responses across localities has typically had detrimental effects on our collective response to the pandemic[53].

## Study definitions of race and ethnicity

The data that we collected for the study used definitions of racial and ethnic groups that were determined by the agencies that collected the data. When the authors are discussing race and ethnicity in their interpretations, they are referring to the historical categories that have social, cultural and political consequences. We use the term Latino to describe people who are otherwise described as Hispanic in many settings. We have used the term non-white in select locations, as not all states had data disaggregated into the same set of categories. Thus, for some analyses, the term non-white directly describes the available data. For a list of the race categories reported by every state in our dataset, see Supplementary Table 7.

Recent advances in medical conventions have prompted discipline-wide introspection about the ways that race and ethnicity are discussed and used in research[54]. This is of critical importance to health equity and racial justice, and although in this work we rely on race statistics reported by states' Departments of Correction, future work will critically examine the differences in approaches for reporting race and ethnicity statistics of incarcerated populations. Notably, it is important to know whether a state's statistical reports use race categories that have been self-reported by the incarcerated person or whether it is interviewer-observed, which is often the case in administrative databases. These approaches are quite different and often result in inaccuracies in measurement of racial disparities[55]. Last, in Supplementary Section 3.3, we introduce a dataset that contains policies from 48 states and the Federal Bureau of Prisons about whether race data of incarcerated individuals are obtained through self-report or visual-assignment from administrators or staff.

## Reporting summary

Further information on research design is available in the Nature Portfolio Reporting Summary linked to this article.

## Data availability

The incarceration data used in this work are public records in each state, and we have included the source URLs in Supplementary Table 1.

Together, data from all 50 states, the District of Columbia and the Federal Bureau of Prisons create the The Dataset on Incarcerated Populations, which we have made publicly available at Zenodo[6] and at GitHub (https://github.com/jkbren/incarcerated-populations-data). The source data used to construct the Dataset on Incarcerated Populations are available through direct download using the links provided in Supplementary Table 1, by public records request or by request to the corresponding author(s).

## Code availability

The Python code to reproduce the analyses and construction of the database is available at GitHub (https://github.com/jkbren/incarcerated-populations-data) and at Zenodo[6]; these repositories contain several Jupyter notebooks with analyses and tutorials on how to automate the collection of some of the data used here.

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

**Acknowledgements** We thank H. Hartle, S. McCabe, T. LaRock, R. Gallagher, M. Frank, R. Rohlfs, E. Ross, B. Terry, S. Goel, D. Allen and members of the Northeastern University NULab for helpful conversations and tips with constructing the dataset. We thank members of the Justice Collaboratory at the Yale Law School, the Institute on Policing, Incarceration & Public Safety at The Hutchins Center for African & African American Research, Harvard University, the MLK Visiting Scholars Program at the Massachusetts Institute of Technology and the Vera Institute for Justice for support, helpful exchanges on the subject matter and feedback on the manuscript. Finally, we thank the Santa Fe Institute, New York University, University of Pittsburgh and the Yale School of Public Health for invitations to seminars and colloquia, at which aspects of this study were discussed.

**Author contributions** B.K., C.B.O., S.V.S. and E.H. conceived the project. C.B.O., S.V.S. and E.H. directed the project. B.K. directed the construction of the data science pipeline. B.K., B.J.S., Z.B., P.K., J.S., A.S. and N.K. collected data. B.K., C.B.O., B.J.S. and S.V.S. conducted analyses. B.K., C.B.O., E.A.W., T.E.-R., S.V.S. and E.H. interpreted and integrated the results. B.K., C.B.O., B.J.S., E.A.W., T.E.-R., S.V.S. and E.H. contributed to researching, writing and editing the final manuscript.

**Competing interests** The authors declare no competing interests.

**Additional information**
**Correspondence and requests for materials** should be addressed to Brennan Klein, C. Brandon Ogbunugafor, Samuel V. Scarpino or Elizabeth Hinton.

# Reporting Summary

## Statistics

For all statistical analyses, confirm that the following items are present in the figure legend, table legend, main text, or Methods section.

| n/a | Confirmed | |
|---|---|---|
| ☐ | ☒ | The exact sample size (*n*) for each experimental group/condition, given as a discrete number and unit of measurement |
| ☒ | ☐ | A statement on whether measurements were taken from distinct samples or whether the same sample was measured repeatedly |
| ☐ | ☒ | The statistical test(s) used AND whether they are one- or two-sided *Only common tests should be described solely by name; describe more complex techniques in the Methods section.* |
| ☒ | ☐ | A description of all covariates tested |
| ☐ | ☒ | A description of any assumptions or corrections, such as tests of normality and adjustment for multiple comparisons |
| ☐ | ☒ | A full description of the statistical parameters including central tendency (e.g. means) or other basic estimates (e.g. regression coefficient) AND variation (e.g. standard deviation) or associated estimates of uncertainty (e.g. confidence intervals) |
| ☐ | ☒ | For null hypothesis testing, the test statistic (e.g. *F*, *t*, *r*) with confidence intervals, effect sizes, degrees of freedom and *P* value noted *Give P values as exact values whenever suitable.* |
| ☒ | ☐ | For Bayesian analysis, information on the choice of priors and Markov chain Monte Carlo settings |
| ☒ | ☐ | For hierarchical and complex designs, identification of the appropriate level for tests and full reporting of outcomes |
| ☒ | ☐ | Estimates of effect sizes (e.g. Cohen's *d*, Pearson's *r*), indicating how they were calculated |

*Our web collection on statistics for biologists contains articles on many of the points above.*

## Software and code

Policy information about availability of computer code

| Data collection | We created a new dataset of the change in the prison population in all 50 states (and D.C.) over time. This involved a combination of web-scraping, public records requests, and manual data entry. The source of each state's data is documented in the manuscript and in the Github repository that contains the data. In this repository, we include time series of each prison system's population data. Additionally, we expanded on our analyses using datasets from states' departments of corrections, public records requests, and offices of the courts. The source, scope, and limitations of these datasets are documented in the text as they are introduced. |
|---|---|
| Data analysis | Data were analyzed using Python, and every analysis has a dedicated Jupyter notebook in the data repository explaining the various findings. This code is written in Python 3.7 and uses the following packages: Pandas 1.1.3, tabula 2.2.0, Numpy 1.19.2, beautifulsoup4 4.9.3, requests 2.24.0. Additionally, we assembled a series of tutorials about how to scrape pdf tables from public sources. Repository: https://github.com/jkbren/incarcerated-populations-data. This repository (Version 1.0.1.) is archived under the following Zenodo doi, 10.5281/zenodo.7675566. Note: the COMPAS risk assessment tool was referenced in the main text but is not used in this study. |

For manuscripts utilizing custom algorithms or software that are central to the research but not yet described in published literature, software must be made available to editors and reviewers. We strongly encourage code deposition in a community repository (e.g. GitHub). See the Nature Portfolio guidelines for submitting code & software for further information.

## Data

Policy information about availability of data

All manuscripts must include a data availability statement. This statement should provide the following information, where applicable:
- Accession codes, unique identifiers, or web links for publicly available datasets
- A description of any restrictions on data availability
- For clinical datasets or third party data, please ensure that the statement adheres to our policy

The incarceration data used in this work are public records in each state, and we have included the source urls in Table A.1. Together, data from all 50 states, the District of Columbia, and the Federal Bureau of Prisons create the "The Dataset on Incarcerated Populations", which we have made publicly available via an archived Zenodo repository, doi: 10.5281/zenodo.7675566 as well as through a Github repository (https://github.com/jkbren/incarcerated-populations-data). The source data used to construct the Dataset on Incarcerated Populations is available via direct download through the links provided in Table A.1, by public records request, or by request to the corresponding author(s).

## Human research participants

Policy information about studies involving human research participants and Sex and Gender in Research.

| | |
|---|---|
| Reporting on sex and gender | This study did not involve experimentation on human subjects -- instead, population-level summaries of incarcerated populations were used to generate the observational findings reported in this manuscript. |
| Population characteristics | While there were not human subject experiments in this work, we sought out data specifically about incarcerated persons' race in every state. |
| Recruitment | N/A |
| Ethics oversight | Because we use aggregated, already existing data in this manuscript, formal review was not sought. |

Note that full information on the approval of the study protocol must also be provided in the manuscript.

# Field-specific reporting

Please select the one below that is the best fit for your research. If you are not sure, read the appropriate sections before making your selection.

☐ Life sciences   ☒ Behavioural & social sciences   ☐ Ecological, evolutionary & environmental sciences

For a reference copy of the document with all sections, see nature.com/documents/nr-reporting-summary-flat.pdf

# Behavioural & social sciences study design

All studies must disclose on these points even when the disclosure is negative.

| | |
|---|---|
| Study description | This study is largely observational, identifying the signature of structural disparities in policy based on race using public records. We propose several hypotheses about potential mechanisms that could bring about the observed trends, and we use publicly available data to offer evidence for each mechanism. |
| Research sample | States' Departments of Correction release periodic statistical summaries about the total population of incarcerated people in the state prison system. We have data from 50 states, the District of Columbia, and the Federal Bureau of Prisons, spanning nearly 20 years. Table A.1 in the Supplementary Information details precisely the duration and reporting frequency of the data from various states' prison systems. |
| Sampling strategy | For each state, we collected the most temporally-resolved data that the state had. For most states in the dataset, we have monthly counts of the number of incarcerated people, by race and sex. For several, we have this data at the weekly level, and three states only report data at the yearly level. This procedure is detailed in Section A.1 of the Supplementary Information, as well as the "State prison populations over time" subsection of the Data & Methods section. |
| Data collection | Data were collected by the coauthors either manually (i.e., direct input from an online source or using web-scraping tools) or through public records requests (i.e., Freedom of Information requests). |
| Timing | The study period is primarily between March 2020 and January 2022, but for proper comparison with historical trends, we analyze time series data since 2013. |
| Data exclusions | N/A |

| Non-participation | N/A |
| Randomization | N/A |

# Reporting for specific materials, systems and methods

We require information from authors about some types of materials, experimental systems and methods used in many studies. Here, indicate whether each material, system or method listed is relevant to your study. If you are not sure if a list item applies to your research, read the appropriate section before selecting a response.

## Materials & experimental systems

| n/a | Involved in the study |
|-----|----------------------|
| ☒ | Antibodies |
| ☒ | Eukaryotic cell lines |
| ☒ | Palaeontology and archaeology |
| ☒ | Animals and other organisms |
| ☒ | Clinical data |
| ☒ | Dual use research of concern |

## Methods

| n/a | Involved in the study |
|-----|----------------------|
| ☒ | ChIP-seq |
| ☒ | Flow cytometry |
| ☒ | MRI-based neuroimaging |

