## [Peer Review File · Nature]

Manuscript Title: COVID-19 amplified racial disparities in the U.S. criminal legal system

Reviewer Comments & Author Rebuttals

Redactions – unpublished data

Reviewer Reports on the Initial Version:

Referees' comments:

Referee #1 (Remarks to the Author):

The question this paper is asking—how did Covid policies impact the racial composition of prison populations—is an incredibly important one. And the authors here have gone to great lengths to gather timely data to answer it, in a field where data almost always comes with a significant lag. Unfortunately, I have some serious concerns with the paper's data and conclusions that I think are important for the authors to address. Some of the concerns may be fairly easy to resolve, but others I fear may reflect deeper, more intractable problems with data on criminal legal system outcomes that often defy simple solutions.

1. The problems with administrative race data. This is one of the more serious concerns I have with the paper, and it is one that is distinctly hard to rectify. To examine how Covid impacted prison populations, the authors gathered administrative prison data from state departments of corrections through July 2021. There is much to applaud here, since the Bureau of Justice Statistics releases state level prison data with ~ 1 year lag (the official BJS data on prison populations for 2020, for example, was released in December 2021); the National Corrections Reporting Program (NCRP), the BJS's main dataset on prison populations at the monthly level—which is what is needed to answer the question posed by this paper—has an even bigger lag of over two years (the most recent data there is 2019).¹ In other words, those relying on official BJS data are still stuck mostly in the pre-Covid era.

However, the authors' reliance on administrative race data at the state level is problematic, because as work with BJS data has shown, administrative prison data alone *cannot be safely used to draw inferences about race in prisons*. If, for example, you look at Table 5 of the BJS's *Prisoners in 2020* report, it says that the national level data on the race of people in prison comes not just from the NCRP, but also from surveys of people in prison the BJS conducted in 2004 and again in 2016. This is because state administrative data, which is what the NCRP is built on as well, classifies a person's race based not on self-reports, but on what correctional officials assign to them as their race, a process that is known to produce misleading results. Table 5, then, calculates prison populations by race by using a lengthy algorithm to merge the administrative data with the race self-reports from the two surveys.

¹ It is worth pointing out that the authors are not entirely correct when they say, on p. 25, that the "scale of data we assembled in this work is unique among the available public datasets about state prison populations...." The NCRP has even more granular, person-level data available for people in prison for ~40 states, with data running back to 2000 for most states, and as far back as the 1970s for at least one. It is true that the NCRP is not entirely "public," since it is a restricted-use dataset housed at ICPSR, but it is one the authors would have access to, and the BJS is in the process of unveiling a public-facing tool to allow those without restricted access to still analyze the data.

Unfortunately, the two surveys are national-level, which is why BJS reports do not include state-specific data on the racial makeup of prison populations. One of the two surveys, the 2016 Survey of Prison Inmates, was designed to try to get state-specific estimates for California, Texas, and Florida, although resistance by Florida prison officials invalidated the Florida part. So, conceivably, the 2016 SPI can be used to adjust administrative data from California and Texas, but those are the only two states that can be adjusted individually. Which suggests that, absent other localized surveys of people in prison, it is hard to say anything state-specific about race in prison.

It is also hard to say in advance how using raw, uncorrected data will bias the results (or, conversely, to say with any confidence whether uncorrected results provide a ceiling or floor for the unbiased effect). My own work with the NCRP suggests that raw administrative data understates the magnitude of the decline in the incarceration rate for Black men and women that started in 2000.² Ostensibly, if administrative data tends to understate trends in Black incarceration more generally, this could mean the authors' results actually *understate* the post-Covid impact (both the 2020 increase and the 2021 reversion). But I'm not sure, at all, if that "more generally" holds; in general, simply assuming errors operate in this sort of symmetric way seems like a concerningly strong assumption.

2. During Covid, prison (alone) isn't really the right institution to focus on. As the authors note on p. 6, some (much? most?) of the decline in prison admissions was from jail holds: people were still being convicted, and people were still being incarcerated—it was just that state departments of corrections refused to accept people from county-run jails after the conviction, forcing many with felony convictions to start serving their (state) prison term in (county) jails. Now, to be clear, prisons and jails are not interchangeable institutions, especially during Covid (where jails were likely even less able or willing to take protective measures like social distancing). And I realize that however bad our data is for prisons, it is orders of magnitude worse for jails. But given the systemic shift towards jails during the pandemic, it seems important to account for them if—as I think is the case—what this paper is really asking is "how did post-conviction confinement change by race during the pandemic?" Given that Black people held in prison tend to serve longer sentences, a widespread reliance on jails for "admissions" in 2020 would cause the Black share in *prisons specifically* to rise even if the overall racial composition of those confined post-conviction in *prisons and jails* didn't change (since if Blacks serve longer average sentences, whites make up a larger share of prison admission cohorts than they do of prison populations as a whole). Accounting for jail populations in general will be hard, and measuring the prison-admission-hold part even harder, but it is an important part of the story here.

3. The 2021 reversion. The result in Fig. 1 is dramatic—but two parts of it are dramatic: not just the increase in 2020, but the reversion almost back to the 2019 ratio that takes place in 2021. While the authors say on p. 5 that they will talk about this reversion in more depth later in the paper, I did not see any subsequent discussion of it. The authors talk in depth about trends in policing, court cases, and prison releases, but as far as I could tell nothing in detail about admissions (and the court processing section, which is the one most closely related to admissions, does not discuss the reversion).

The authors suggest, on p. 5, that the reversion is most likely the result of new inflows in admissions as the jail holds get lifted—which would reinforce my point above that it is essential to figure out a way to incorporate jail populations here as well. Regardless, the sharp and sudden reversion almost back to pre-Covid trends while we were still very much in the midst of Covid is something that needs more discussion and analysis.

² Perhaps a small point, but the authors are not entirely correct on p. 2 when they say that the share of Black prisoners has been declining "for nearly a decade"—the incarceration rates for Blacks has been dropping for more than two decades, since 2000, as is discussed here: <https://www.washingtonpost.com/opinions/2019/04/30/weve-made-remarkable-progress-black-incarceration-now-we-need-know-why/>.

4. The important role of counties. While we tend to think about prison populations as state populations, in many ways they are *county* populations. While it is true that prisons are run by the state government, the path to prison comes from being arrested by city police or county sheriffs, charged and (almost always) taking a plea from a county prosecutor, and then being sentenced by a judge whose courtroom is generally county-specific. If, then, during the pandemic, urban counties kept cases moving faster than rural counties—due to better infrastructure at every level—and were more likely to send those convicted to prison rather than holding them in local jails, then some of the change in the disparity in prison populations could be the result of systemic differences in county-level institutional capacity and competence. Often state-level data masks important intra-state county-level heterogeneity, some (or much?) of which could be central to the causal mechanisms at play here.

5. Lack of state-specific analogs to Fig. 1. In the data appendix, the authors provide state-specific trends in state prison populations. But they do not provide similar graphs that replicate Fig. 1 at the state level. Given the concentrations of prison populations across states, it is important to provide this data as well. At the end of 2019, just three states held over 30% of all people in US prisons, and just nine held just over 50% of all people in prison (which should not surprise us, since those same states held 27% and 50% of the nation's overall population). This raises the very real risk that the results we see in Fig. 1 are driven by just a handful of states that hold the bulk of the nation's prison populations (especially given the relatively small absolute value of the change: less than one percentage point, or about 2.5% from the 2019 baseline). This sort of concentrated-effect issue is actually a common problem with data about the criminal legal system. About 30% of all life sentences are just in California; historically, about five counties accounted for over half of all children sentenced to life without parole; in the 2000s, the BJS had to report admissions to prison off parole separately for California, because California's large and idiosyncratic results had a huge impact on the national average. I think it important for the authors to provide a state-level breakdown of Fig. 1, so readers can better see what the full national story looks like in a system where populations are highly concentrated in just a few states, and where effects can be quite heterogeneous.

6. The case studies. A corollary to that last bullet point is that I am concerned about what conclusions we can draw from the authors' case studies. The authors rightly note on p. 10 that the "heterogeneity in the structures of criminal justice systems across states" makes it hard to draw firm conclusions from state-specific case studies. I fear that this heterogeneity may actually be quite a big impediment—and not just across states, but within them too. For example (as I explain in more detail below), state highway patrol troopers have significantly different jobs than city police officers, and face substantially different populations and types of calls. So not only might it be hard to generalize from Texas to, say, New York, but it's likely hard to generalize from the Texas Highway Patrol even to the Austin Police Department.

6. At the very least, the authors should give some sense of whether these states are at all representative of the country as a whole for the factors they are using them to illuminate. If someone used, say, California in the 2000s as a case study in prison admissions off parole, or Philadelphia as a case study in juvenile life without parole sentences, it would give a significantly misleading view of the country as a whole, since each of those was a major outlier from whatever sort of general national trends existed.

7. A related concern I had with the case studies is that I got the sense—and I could very well be wrong here—that they were chosen in part because they had the sort of granular data necessary for the case study. If so, this raises additional concerns about non-representativeness, since it is important to ask why, given how generally shoddy and unavailable data on the criminal legal system is, these specific states happened to gather and make public these specific sources of data. It could be for reasons uncorrelated with what the authors are examining, but if so, the authors should make that clear; if not, they should try to give readers a sense of the ways in which the case-study states may be outliers.³

³ As a general matter, I fear that disclaimers like the ones they provide, saying that these case studies should not be read as generalizations, are ineffective absent some sort of explicit discussion about the risks these specific case

* *

I had a few more-specific concerns as well.

7. Texas Highway Patrol Data. If the focus of this paper is on what drives prison populations, or what drives the confinement populations of those convicted of felonies (to account for the jail-hold issue), I'm not sure trends in highway stops are particularly informative. The authors correctly note on p. 11 that "traffic stops are not the only way for people to enter the judicial system — nor are they the primary source of prison admission..." but that last phrase understates things. Given the comparatively-serious nature of offenses that lead to prison admission, traffic stops likely play a very small role in those admissions. It's also worth noting that the shocks of Covid make it hard to interpret what, exactly, to infer from the THP data. As the authors correctly note, the pandemic led to a significant shift in who was commuting, and in a direction that surely led to an increase in the share of Black drivers. This would suggest that even a race-blind THP would stop a growing fraction of Black drivers during the pandemic. To be clear, I'm not saying the THP was race-blind! But to understand the dynamics of systemic racism, especially for the (prison-centric) question this paper is addressing, it is important to separate out what is being driven by the criminal legal system itself vs. how that system interacts with other structural problems (like how Black Americans were less likely to have work-from-home jobs).

8. Arkansas parole data. On p. 9, the authors state that the percent of whites released early in Arkansas under Covid rules was greater than their share of the prison population, something "we would not expect to see in a prison system absent biases in sentencing." This is not necessarily true. Many states' Covid early-release rules applied only to people in prison for non-violent crimes, and at least at the national level, at the end of 2019 (the year with the most current offense-specific data), 64% of Black people in prison were serving time for a violent conviction, vs. 50% of white people in prison (see Table 14 in the BJS's *Prisoners in 2020* report, which has offense-level data at a one-year lag). Now, of course, it is true that Blacks receive longer sentences than whites for the same crimes, that the same conduct is more likely to lead to a conviction for violence (or a more serious degree of violence) for Black person than a white one (due to plea deals offered), and that racial disparities in violence (which are unambiguously true for murder, though the data gets much more nebulous after that) are inarguably the product of systemic racism as well. But from a policy perspective, nailing down the relative importance of these various mechanisms is important. For example, if the racial disparities in Covid releases is due to racial bias within the pool of those eligible (which was rarely if ever the entire prison population, given the restrictions often imposed on releasing those convicted of violence), then the solution is to, say, figure out a way to limit problematic discretion. If the problem is in who is in the eligibility pool—if releases are proportionate to the racial makeup of those eligible for Covid release, and the disparities reflects differences in who is eligible—then addressing bias at the decision-making level will do little, and we need to focus instead on determining what the eligibility pool should look like. This latter question, however, becomes increasingly tough to answer, both politically and normatively, the more the key issue is disqualifying those with violent convictions.

9. Texas severity data. I don't think the Texas severity data can actually answer the question of whether the spike is due to a change in offending patterns across race. That answer requires us to look not at differences in the crime severity of people sent to prison, but at differences in the crime severity of those who are or are not sent to prison. That the relative increase seems to come at the lower end of the sent-to-prison severity does not rule out the possibility that there were racial differences in those committing felonies that merited any sort of prison time.

studies face. Without that, I fear readers will too often instinctively take the case study to be at least roughly representative.

To be clear, I don't think changes in offending can explain the spike the paper shows for Texas, either. If nothing else, at least in the short run, changes in offending drive prison populations through admissions, and there is inevitably a lag—likely often several months—between offense and admission. So an admissions-driven spike right at the Covid outbreak would be the result of offending changes months before. Not only would that sort of precise spurious-correlation timing be surprising, but it's not clear there was any sort of spike, at least in the major Texas cities, at the right time.⁴ But I still think it is also important to note that the Texas data given in the paper does not in fact refute the possibility that changing offense patterns could explain the 2020 disparity in-increase in Texas.

* * *

As I mentioned at the top of this report, I think the question the authors are asking is an incredibly important one. But I also fear that collides with several significant challenges that arise in almost any paper trying to use administrative data on the criminal legal system. I hope that the authors are able to figure out ways to address these issues, alt-hough in some cases I fear that there may be no real fix—the authors may need to think instead about ways to express the uncertainty in their results that arise from the ines-capable systemic errors in this sort of administrative data.

⁴ See, for example, the seven-day moving averages for all violent incidents in 2019 and 2020 for Austin, Dallas, Houston, and Fort Worth, available on-line at <https://citycrimestats.com/covid/> (unfortunately, it is impossible to provide a direct link to the specific set of graphs for those four cities, so the authors will need to recreate them themselves).

Referee #2 (Remarks to the Author):

This paper documents a new fact: the Black and Latino share of the state prison population increased immediately following the onset of the COVID-19 pandemic. To show this, the authors collect data on the prison population in all 50 states and the District of Columbia. Using more detailed data from three states (Florida, Texas, Arkansas), the authors examine three potential explanations: disruption in court operations, changes in police interactions, and decarceration.

The new fact that the paper documents is important and interesting. My main concern with the paper is that, because of limitations of the data the authors use, the paper does not provide much concrete evidence on the underlying mechanisms that produce this pattern. As the authors emphasize, understanding the mechanism is important to understanding what to make of this pattern moving forward. The authors focus on and present state-specific case studies for each of three mechanisms: disruption in court operations, changes in police interactions, and decarceration. However, in each case, it's not clear from the data if that mechanism is empirically important for explaining the stylized fact of interest and, if so, why, or what that tells us about the nature of structural racism in criminal justice. Moreover, a more detailed and informative examination is feasible because the more detailed data required are available to researchers, at least in some states (though perhaps with some lag).

I expand on these points for each mechanism below.

Disruption in court operations

The authors use monthly data on case dispositions in Florida Circuit courts and show that the number of dispositions declines sharply in March 2020, with a corresponding increase in the backlog of cases. This drop in dispositions coincides with an increase in the nonwhite share of the prison

Unfortunately, as the authors point out, without more detailed data—including data on the race of defendants—it is difficult to understand why a decrease in dispositions would increase the nonwhite share of the prison population. Is it that white defendants experience a disproportionate increase in case dismissals? Are white defendants more likely to have their cases delayed or to receive non-confinement sentences (relative to pre-pandemic)? Can racial differences in treatment be explained by racial differences in the distribution of the charges (e.g. charge severity)?

One hypothesis is that new prison admissions are disproportionately white relative to the prison population because white admits tend to have short prison sentences (either due to differences in charges, or differences in sentencing conditional on charge severity). The same would be true for releases (see Figure A7). If the pandemic stopped new admissions and accelerated releases, those left in prison would tend to have long sentences, a population that may be disproportionately nonwhite.

It is important to note that the kind of detailed microdata required to answer these questions are available in several states (for example, Lee and McCrary 2017; Feigenberg and Miller 2021; Rose and Shem-Tov 2021). It's possible that there would be some lag in accessing these data relative to the data used here, but I think the ability to unpack mechanisms would be well worth the wait.

A side note: the authors note that they "observe the same correlations between percent of dismissed cases and percent non-white incarcerated people prior to the COVID-19 pandemic". I thought this was an interesting point, but could not figure out where it was demonstrated in the paper. It would be helpful to document this finding more explicitly in the paper.

Changes in police interactions

The authors use data on traffic stops conducted by Texas Highway Patrol to understand how changes in policing and police interactions with civilians may have contributed to changes in the racial composition of prisons. They show that the percentage of traffic violations that involve Black and Latino drivers increases significantly after March 2020.

An important shortcoming of this analysis is that only an extremely small percentage of traffic stops lead to a prison sentence, so it is not at all clear how traffic stop patterns are related to the motivating stylized fact. If there was some meaningful change in policing that contributed to an increase in the nonwhite share of prisoners, these would not be the data to show it. It is true that, compared to court records, it is much less straightforward to obtain data on policing. But it would be

possible to get data on 911 calls and stops conducted by city police departments (rather than highway patrol) (Hoekstra and Sloan, forthcoming).

A side note: the authors reference a 2015 investigative report documenting that Texas officers were systematically misreporting Latino drivers as white and suggest that the traffic stop data may underestimate the true proportion of traffic violations. My understanding is that the Texas Department Public Safety responded to this revelation by requiring officers to ask drivers to self-identify, which sharply increased the recorded Latino share of stopped drivers (Luh 2020).

Decarceration

Here the authors show that the population of incarcerated people that were eligible for early release in Arkansas is disproportionately white compared to the overall prison population. It would be helpful to know what the criteria for early release in Arkansas were in practice. For example, the authors state that "eligible incarcerated people needed to have a parole plan in place". How does one get a parole plan in place? Are these incarcerated people that were nearing the end of their prison sentence? In that case, it would be important to document whether racial differences in early release simply reflect racial differences in average sentence length, which may in turn reflect racial differences in the composition of charges. Consistent with the idea that this pattern reflects differences in average sentence length, the fact that those released from incarceration are disproportionately white appears to not be unique to the pandemic (Figure A7).

These questions could be answered with a more detailed accounting of individual-level prison admissions and releases, which could be done with available state department of corrections data on the prison population (potentially including the bulk data from Arkansas that the authors reference).

References

CJARS, 2022. <https://cjars.isr.umich.edu/data-documentation-download/>

Feigenberg, Benjamin and Conrad Miller, 2021. "Racial Divisions and Criminal Justice: Evidence from Southern State Courts", *American Economic Journal: Economic Policy*, 13(2).

Hoekstra, Mark and CarlyWill Sloan, forthcoming. "Does Race Matter for Police Use of Force? Evidence from 911 Calls", *American Economic Review*.

Lee, David S. and Justin McCrary, 2017. "The Deterrence Effect of Prison: Dynamic Theory and Evidence," *Advances in Econometrics*.

Luh, Elizabeth, 2020. "Not so Black and White: Uncovering Racial Bias from Systematically Misreported Trooper Reports", unpublished manuscript.

Rose, Evan K. and Yotam Shem-Tov, 2021. "How Does Incarceration Affect Reoffending? Estimating the Dose-Response Function", *Journal of Political Economy*, 129(12).

Referee #3 (Remarks to the Author):

I appreciate the opportunity to review this manuscript. It is well-written and understandable, even for non-specialists. This paper can serve as a model for using the Covid-19 pandemic as a natural experiment for testing theories related to crime and criminal justice. As the authors argue, their creation of the unique dataset is a contribution in itself. The core argument, that whites benefited more than Blacks from the decrease in the prison population during the first year of the COVID-19 pandemic, is interesting and important, and I think it would draw widespread attention from academics, journalists, legal practitioners, and advocates.

The authors' use of case studies is effective for proposing mechanisms that could have caused the racial disparity. It does appear, as the authors indicate, that disruption in the court system is the most probable mechanism. I'm curious if Massachusetts experienced similar disruption in the court system. A brief analysis of the Massachusetts case could strengthen the manuscript (if that's possible given data constraints).

The authors may want to emphasize the importance of structural inequality when discussing the decarceration case. Since only people with an "approved address to stay upon release," and a parole plan in place (which likely included having housing, job opportunities, and documentation like a driver's license) were eligible for release in Arkansas, the policy likely privileged prisoners with at least limited social and economic resources (perhaps more white than Black people?). Also, I found the last sentence in the last paragraph on p. 9 a bit confusing. Does "biases in sentencing" mean that biased sentencing led to longer sentences for Black vs. white defendants with similar cases, and/or that Blacks were more likely than whites to be classified as "violent" (vs. "non-violent"). If I understand the analysis correctly, these are the two sentence-related factors that affected release decisions.

Referee #4 (Remarks to the Author):

This paper examines shifts in the racial composition of incarceration in the United States during and directly after a critical juncture – the disruptions posed by the Covid-19 pandemic. While rates of incarceration were steadily declining prior to the pandemic (as well as racial disparities in prison admissions), the pandemic created "the largest, most rapid single-year decrease in prison population in American history." Here's where the paper makes its boldest claim: that that the sudden fall in incarceration rates occurred alongside and worsened racial disparities in incarceration "erasing much of the progress from the declines over the last decade." Throughout the paper, they conclude that the pattern they locate – falling incarceration accompanied by rising disparity -- is due to racial bias in features of the criminal legal system: who gets released, who police stop, and who gets processed. To summarize their findings, the pandemic shifted all three, amplifying existing inequalities. Using an original data compilation of all states incarceration rates by race as well as

three state case studies to explore potential mechanisms for amplified racial disparities, they argue that a combination of three factors were at work: decarceration policies that released more white incarcerated people, court closures in the early months of the pandemic, and policing interactions during the pandemic.

Central claims:

- Substantial fall in prison populations across every state during the pandemic (nationally, a 16.3% decline)
- The people who benefitted most from falling admissions and releases were white
- Thus, the relative number of black incarcerated people increased temporarily
- The race-based patterns cannot be explained by crime severity

The paper makes an important contribution to our understanding of racially disproportionate incarceration and delivers extremely compelling findings. The authors have done an incredible service to our fields by constructing this dataset! This was effortful and entailed filing FOIA requests across multiple states where the data by race were not publically available, cross-checking data against existing measures for accuracy, compiling monthly data on trial statistics in Florida to dig deeper into mechanisms, navigating incredibly thorny issues related to the data, and anticipating possible errors. A reader can see a full listing of how the data were collected in Table A.1 (and one quickly sees what a patchwork of different state practices on how race is categorized and classification systems). The resulting figures and measures and Appendix display considerable care and precision. The authors provide back of the envelope calculations to show how much more the black prison population would have decreased if it matched the white reduction in incarceration. In short, over 14,000 fewer black people would have been incarcerated.

Because the argument is such important claim and set of findings, I have several ideas for ways the paper could improve and really hone in on its most important findings.

The paper would do well to actually frontload the main finding: a dramatic surge in black composition of the prison across every state, despite (or perhaps because of) a decline in the scale of incarceration. This is a novel finding and very troubling. It is remarkably consistent across states, regardless of whether the fraction of black incarcerated people was increasing, stable, or decreasing in a particular state prior to the onset of the pandemic in March 2020. Unearthing the complex mechanisms for this is a very difficult undertaking and the authors did their level best to unpack possible explanations and examine each with fascinating mini case studies (case studies because the data for every state do not exist and they are right to end with a call for states to release more and better data on race). I want to emphasize here as a scholar of incarceration just how difficult this data collection is.

After staring at the figures and the text for some time, I have an alternative hypothesis for what's going on: by temporarily starving the system of a source of falling racial disparity in prison admissions, cutting off prison admissions through court closures affected the racial composition of the prison.

The logic here is a bit complex so let me explain. Prison admissions (flow) contain much less racial

disparity than total prison confinement (stock). The racial disparity in prison admissions (2:1) is much lower than the black/white disparity in total imprisonment (6:1). And for several years prior to the pandemic, black prison admissions were falling dramatically from their peak in the late 1990s, while white prison admissions were growing especially among those whites without college education (though from a much lower level), which resulted in a narrowing of racial inequality. For example, the Muller & Roehrkasse data show that admission rate among black people without college education fell from 4,494 people per 100,000 in the year 2000 to 2,511 per hundred thousand in 2015 while the rate among similarly educated whites actually grew. Let us not be confused: Racial disparities in prison admissions are still severe and black people continue to be admitted to prison more than any other group, reflecting the ongoing legacies of racially oppressive policies and practices. However, the black/white disparity in prison admissions reached its lowest level in decades in 2015.

So, if you just stop admitting people to prison through court closures, you are effectively cutting off a key source of lessening racial disparity in incarceration. This is counterintuitive. Otherwise stated, if we stopped admitting people to prison at the rates we were seeing prior to the onset of the pandemic, which showed dramatically falling racial disparities (see Muller and Roehrkasse's 2021 Social Forces article), we would see a plateauing of the racially moderating effect of prison admissions on total prison population. Since the black/white disparity in admissions is lower than the black/white disparity in incarceration, if you just halt prison admissions, falling racial disparities in imprisonment would stagnate. If, on the other hand, covid court closures had not interrupted prison admissions, the pre-covid trend of steadily increasing admissions of low-education white people would have continued (Muller and Roehrkasse 2021). But with the pandemic (and prison populations not being replenished with relatively more low education white admissions), the declining share of black people making up the prison population would necessarily stabilize. That is a numerical property that does not have to do with intentional racial bias (though everything to do with the lasting effects of the drug war and racialized punitive policies on total confinement). To put the point more dramatically, if all courts closed again tomorrow, we should see the same effect (a slowdown in the decline of black/white incarceration disparities). Now, what I am suggesting would only explain a flattening in the pre-covid downward trend in the black percent of incarcerated people; it would not explain why the black share of incarcerated people actually rises as the authors show. For that, we need to turn to who is being released, which the authors do in looking at racially unequal early release patterns, and this is what I see as one of the most promising mechanisms in the paper. In combination with what I'm suggesting above, emergency decarceration orders that gave disproportionately more white incarcerated people the option to be released would have resulted in the upsurge in the share of the prison that was comprised of black people.

One suggestion then is for the authors to acknowledge this possibility in the paper or say why they do not agree or think it is driving a portion of the results. Can the authors show prison admissions as well as total incarceration rates and total prison population? If a big claim in the paper surrounds who gets admitted, then why not show prison admissions by race, not just total incarceration (and black share of prison population)?

A related concern is that one of the most prominently discussed mechanisms in the paper is the idea that court closures and disruptions in prisoner intake from courts and county jails due to the

pandemic exacerbated existing biases. The claim, as I read it, is this: when courts closed, the sheer number of defendants fell sharply, and in the state they have data for, “more than 99% of cases did not go to trial,” so more people agreed to guilty pleas or had charges dropped. The authors are right to say that plea deals and which people have their cases dropped are heavily racialized processes. There is no argument there and I agree that there are “multiple potential sources of bias that can stem from disruptions in the court system”. But I have read this part several times and am still very confused. Is the idea that many more whites had their cases dropped than black defendants, who were more likely to take plea deals? That would take more than a month to be reflected in the upswing in the black share of the incarcerated prison population, no? And it would have to be quite large to affect the racial composition of the prison in the months afterwards. But more troubling as I read it is that Table A.4. suggests that pre-trial guilty pleas as a share of total defendants filed did not increase after March 2020; indeed, quite the opposite as I’m reading the table: guilty pleas immediately drop from 10,277 in Feb 2020 to 7,412 in March 2020 and drop further in April to 2,344. It also drops if we calculate it as a share of total defendants filed. Case dismissals pre-trial also fall sharply, so sharply that the racial bias would have to be absolutely huge to explain any shift in the racial composition of the prison.

This brings me to what I see as the strongest evidence in the paper regarding mechanisms: release policies that favored white incarcerated people. The idea is that states took measures to de-densify prisons given the public health crisis in prisons and these measures advantaged white prisoners given how they reflected and heightened structural biases. For example, in many states to be released or even considered for release, one has to be nearing parole release, have a lesser offense, have good time credits, have low recidivism risk, and show evidence of an address one can go to (a “viable home plan”) – all features that would disadvantage black populations. The release criteria example from Virginia on page 14 was incredibly illuminating. The authors own evidence in the Arkansas case study support this claim; they find that nearly three-quarters of those eligible for release were white, much greater than would be commensurate with their share of the prison population in that state (57%). I also appreciated their discussion of state variation and analysis in several other states (p. 10), showing that some states like Massachusetts showed the possibility of decarcerating “in a way that does not exacerbate existing racial inequalities.” They provide areas for other scholars and future research to examine further.

1. The paper doesn’t make much of the fact that the black share of incarcerated people does not rise and keep rising; in fact, the data shown reveal that it rises for several months in 2020 from just under 38% to just over 38.5% and then turns back down to its original level by the first months of 2021. The way the paper is written focuses entirely on the rise and not the hill-like pattern depicted in almost all of the figures. Every state except Arizona in Figure 2 depicts this inverted U shaped pattern. (The authors acknowledge this only briefly on page 5 where they say “this spike is temporary, eventually returning to its pre-pandemic level”). The rise was actually quite short-lived – and that is not to discount its importance. But it means that whatever caused the shift in the black share of the incarcerated population was not an enduring feature or that the pandemic caused a lasting change in the demographic makeup of the prison. This point is important because it shifts how we understand the dramatic (but not durable) change in the racial composition of the prison. It may give additional clues to the importance of potential mechanisms. The timing of the “turnaround” in relative rises in black share of the imprisoned could help decipher causal

mechanisms. Can the authors do more to discuss and interpret this?

2. What is the broader story for how we should understand how a crisis with deeply structural dimensions intersects with another site of structural violence?

3. The traffic stops data are very interesting but theoretically, readers will have difficulty understanding why traffic safety stops would have much to do with incarceration rates (most of these, as the authors acknowledge, do not lead to confinement or even arrest!).

4. At several points, the authors say bias (which readers could interpret as intentional malevolent racism) but I think what their data are really pointing to are what they say in the conclusion – “the structural nature of racial disparities in the United States carceral system.” One that releases people based on their housing access; one that continues to reflect extreme levels of racialized incarceration rates (on the order of 6:1) due to policies and practices that targeted black people and communities; one that meant that more black people were on the road and potentially being stopped by police because they had to work and carry on in a pandemic where some could isolate and work remotely. I suggest not leaving this to the conclusion.

5. Table A.2 – in future work, the authors might explore why some states had larger reductions in their prison populations. There is a lot of variation to explore here, ranging from a drop in New Jersey that was 65.7% of pre-pandemic levels to Wyoming which only dropped to 93.1% of the pre-pandemic level.

6. Figure 3D is also confusing – the authors should be clear about what it is depicting – does it represent court dismissals at the county level?

7. How does discretion play a role in releasing practices? I was surprised that the authors don't really consider this, only the structural features of release policies that prioritize certain kinds of offense, viable homes upon release, etc.

8. In Figure A.7, the release patterns by race in Arkansas seem to get better after March 2020, which goes against the main argument. What is going on here? And “incarcerated Black people are released at lower than expected rates” as far back as 2018. The authors should acknowledge and explain that this is a long-standing pattern.

Overall, a terrific and compelling paper and one that should be published and will make a significant contribution to our knowledge base.

Vesla M. Weaver

Author Rebuttals to Initial Comments:

Reviewer 1 -- Nature 2022-01-00306

The question this paper is asking—how did Covid policies impact the racial composition of prison populations—is an incredibly important one. And the authors here have gone to great lengths to gather timely data to answer it, in a field where data almost always comes with a significant lag. Unfortunately, I have some serious concerns with the paper’s data and conclusions that I think are important for the authors to address. Some of the concerns may be fairly easy to resolve, but others I fear may reflect deeper, more intractable problems with data on criminal legal system outcomes that often defy simple solutions.

We appreciate the careful and thorough expertise expressed in this review, which have resulted in multiple improvements to the original manuscript. We note, as well, that the phrasing used in the first sentence above (“how did Covid policies impact...”) inspired a few wording changes in the Introduction, as our work is not as directly about COVID-19 policies, per se, as it may seem. We use the societal changes as a result of the COVID-19 pandemic as a vehicle to look closely at structural issues in the U.S. prison system—some of these changes were the introduction or absence of policies, but many were not.

1. The problems with administrative race data. This is one of the more serious concerns I have with the paper, and it is one that is distinctly hard to rectify. To examine how Covid impacted prison populations, the authors gathered administrative prison data from state departments of corrections through July 2021. There is much to applaud here, since the Bureau of Justice Statistics releases state level prison data with ~ 1 year lag (the official BJS data on prison populations for 2020, for example, was released in December 2021); the National Corrections Reporting Program (NCRP), the BJS’s main dataset on prison populations at the monthly level—which is what is needed to answer the question posed by this paper—has an even bigger lag of over two years (the most recent data there is 2019). (It is worth pointing out that the authors are not entirely correct when they say, on p. 25, that the “scale of data we assembled in this work is unique among the available public datasets about state prison populations....” The NCRP has even more granular, person-level data available for people in prison for ~40 states, with data running back to 2000 for most states, and as far back as the 1970s for at least one. It is true that the NCRP is not entirely “public,” since it is a restricted-use dataset housed at ICPSR, but it is one the authors would have access to, and the BJS is in the process of unveiling a public-facing tool to allow those without restricted access to still analyze the data.) In other words, those relying on official BJS data are still stuck mostly in the pre-Covid era.

However, the authors’ reliance on administrative race data at the state level is problematic, because as work with BJS data has shown, administrative prison data alone

cannot be safely used to draw inferences about race in prisons. If, for example, you look at Table 5 of the BJS's Prisoners in 2020 report, it says that the national level data on the race of people in prison comes not just from the NCRP, but also from surveys of people in prison the BJS conducted in 2004 and again in 2016. This is because state administrative data, which is what the NCRP is built on as well, classifies a person's race based not on self-reports, but on what correctional officials assign to them as their race, a process that is known to produce misleading results. Table 5, then, calculates prison populations by race by using a lengthy algorithm to merge the administrative data with the race self-reports from the two surveys.

Unfortunately, the two surveys are national-level, which is why BJS reports do not include state-specific data on the racial makeup of prison populations. One of the two surveys, the 2016 Survey of Prison Inmates, was designed to try to get state-specific estimates for California, Texas, and Florida, although resistance by Florida prison officials invalidated the Florida part. So, conceivably, the 2016 SPI can be used to adjust administrative data from California and Texas, but those are the only two states that can be adjusted individually. Which suggests that, absent other localized surveys of people in prison, it is hard to say anything state-specific about race in prison.

It is also hard to say in advance how using raw, uncorrected data will bias the results (or, conversely, to say with any confidence whether uncorrected results provide a ceiling or floor for the unbiased effect). My own work with the NCRP suggests that raw administrative data understates the magnitude of the decline in the incarceration rate for Black men and women that started in 2000. (Perhaps a small point, but the authors are not entirely correct on p. 2 when they say that the share of Black prisoners has been declining "for nearly a decade"—the incarceration rates for Blacks has been dropping for more than two decades, since 2000, as is discussed here.) Ostensibly, if administrative data tends to understate trends in Black incarceration more generally, this could mean the authors' results actually understate the post-Covid impact (both the 2020 increase and the 2021 reversion). But I'm not sure, at all, if that "more generally" holds; in general, simply assuming errors operate in this sort of symmetric way seems like a concerningly strong assumption.

This comment contains several substantive, important points, and we've made multiple additions to the manuscript in response (both in terms of new data and new analyses). Below we summarize the overall changes then provide point-by-point responses.

- First, we contacted every state's Department of Corrections and directly asked how the race of incarcerated people is collected and recorded. Nearly every state replied to us saying that their policy is that race is self-reported at intake into the prison.

- Second, we have added a large section to the Appendix that uses NCRP data as a baseline for comparison and validation of several of the states in our dataset.
- The newly collected data, together with the validation analyses done with the NCRP data, both reassure us about the quality and validity of our data as well as open new questions for future work—including how to improve standardization and accuracy of NCRP data.

First, we instinctively agreed with Reviewer 1’s main point about the problematic nature of administrative race data (e.g. “This is because state administrative data, which is what the NCRP is built on as well, classifies a person’s race based not on self-reports, but on what correctional officials assign to them as their race, a process that is known to produce misleading results.”). The lack of self-report data has been a hindrance in all sorts of research, and it can be especially confounding when asking questions about racial disparities. However, as we dove deeper into this question, we decided that the easiest thing to do would be to contact every state’s Department of Corrections (as well as DC and Federal BOP) to find out the actual policy and how it’s implemented. In nearly every state, the explicit policy is to collect self-reported race data at intake. There are a few states who did not return our communication and a few states with either ambiguous policy language (e.g. Minnesota, which explicitly writes “Race information may be self-identified or classified by an observer.”) or policy language that is suggestive of self-report but not entirely (e.g. Massachusetts: “It is mostly self-reported, however, if the county sends a face sheet the Booking Officer will use that.”). Only one state (Texas) explicitly referred to staff members visually assigning someone’s race “...during intake, the [Texas Department of Criminal Justice] will visually determine the race of the individual.”

We have added this result as an additional table in the Supplemental Information, as well as a brief discussion about why this is a crucial addition to our dataset. We sincerely appreciate the original comment, as we would not have been driven to collect this data without it. Note: we also acknowledge the difference between a state’s written policy (e.g. from a spokesperson from the Indiana Department of Correction: “The offenders self-identify at the time of entry into our predetermined categories (not free form).”) and how it is actually enacted by staff at the prisons (e.g. from a spokesperson from the Massachusetts Department of Correction: “It is mostly self-reported, however, if the county sends a face sheet the Booking Officer will use that.”). So while merely collecting data about different states’ written policies is useful, there is likely heterogeneity in the compliance to any specific policy. We have added a note about this nuance in the 4.3 Study Definitions of Race and Ethnicity section.

If, however, we do not take the states’ recording policy at face value (there are, after all, a number of ways where we can imagine the written policy of collecting race data via self-report diverging from the on-the-ground practice), there is more data validation that can be done by comparing our dataset to the NCRP data—as Reviewer 1 mentions. In this spirit of thoroughness, we selected several states with large prison populations that appear to have high coverage in the

NCRP data, and we used the historical trends in these states as a comparison / further validation tool for our data. We have added a section to the Supplemental Information where we introduce the NCRP data, justify the selection of the states for comparison, and show several plots comparing the two datasets. We see this as a validation with multiple benefits: First, it grounds the data we have collected in a known to a well-studied companion dataset (to give a snapshot, below is a comparison between 12 states' total prison populations until end of 2019—blue curves are our dataset, red is NCRP):

States with high overlap between NCRP data and Klein et al. (2022) data

Immediately we are reassured by the high correspondence between these twelve states. Additionally, we see roughly equivalent trends when looking at the percent incarcerated population who are Black:

States with high overlap between NCRP data and Klein et al. (2022) data

Next, there are several states where the overall trend in the prison population is similar, just shifted uniformly up or down (i.e. states with the same or similar-shaped curves but shifted by a fixed amount). For example:

States with shifted overlap between NCRP data and Klein et al. (2022) data

However, what is noticeable about several states with systematic differences is that a key measure of interest—the percent of the incarcerated population who are Black—remains more or

less the same (or, if not exactly overlapping, the curves do not appear to be *systematically* different):

States with shifted overlap between NCRP data and Klein et al. (2022) data

In the figure above, note that for the most part, we see overlapping values for the percent of Black people incarcerated in each state. Lastly, there are several states where the NCRP data is clearly not capturing the same information that our dataset contains. These are states that—we suggest—do not have high coverage or high data quality in the NCRP dataset or have otherwise changed their reporting protocol during the duration of their inclusion in the NCRP. For example:

States with little overlap between NCRP data and Klein et al. (2022) data

Here, we see an opportunity to use our data to augment / supplement / help fill in states with known reporting irregularities or other issues in the NCRP dataset. While the insights from these comparisons between the two datasets were not the intended contribution of our paper, we do think that adding these analyses to the manuscript strengthens the work and offers a sort of roadmap for future work.

Ultimately, if we only analyze states with high overlap between NCRP and our data (a proxy for overall reporting quality: California, Colorado, Florida, Georgia, Illinois, Indiana, Kentucky, Maine, Minnesota, Mississippi, Nevada, New York, North Carolina, North Dakota, Oregon,

Pennsylvania, South Carolina, and Tennessee), we see the same qualitative result, but we also observe a much stronger effect size (42.5% of state prison populations were Black right before March 2020, increasing to 43.7% in October 2020; compared to 37.8% to 38.6% when including every state)—as Reviewer 1 hypothesized.

We have updated the Supplemental Information section with more of this discussion. To conclude, we offer a brief statistical point about the possibility of systematic mis-reporting of race statistics by states. Following up on Reviewer 1’s comment, “My own work with the NCRP suggests that raw administrative data understates the magnitude of the decline in the incarceration rate for Black men and women that started in 2000... Ostensibly, if administrative data tends to understate trends in Black incarceration more generally, this could mean the authors’ results actually understate the post-Covid impact (both the 2020 increase and the 2021 reversion).” While it has been frustrating during data collection that there is almost no uniformity across states’ race reporting (except, as discussed above, that most states’ policy is to record race via self-report), it may actually be useful from a statistical standpoint. That is, if we assume widespread misreporting or underreporting of the race of incarcerated people in each state, we can pose a probabilistic question about the trends we’re seeing: what is the joint probability that not only does every state mis-report the race of incarcerated people, but also that we observe a similarly timed spike in percent Black population (Fig. 1b) in almost every state? While we ultimately do not have these probabilities, we feel—in light of the extensive further analyses and data collected on this point—that this offers an intuitive understanding as to why we see such a broad validation of our original main results.

2. During Covid, prison (alone) isn't really the right institution to focus on. As the authors note on p. 6, some (much? most?) of the decline in prison admissions was from jail holds: people were still being convicted, and people were still being incarcerated—it was just that state departments of corrections refused to accept people from county-run jails after the conviction, forcing many with felony convictions to start serving their (state) prison term in (county) jails. Now, to be clear, prisons and jails are not interchangeable institutions, especially during Covid (where jails were likely even less able or willing to take protective measures like social distancing). And I realize that however bad our data is for prisons, it is orders of magnitude worse for jails. But given the systemic shift towards jails during the pandemic, it seems important to account for them if—as I think is the case—what this paper is really asking is “how did post-conviction confinement change by race during the pandemic?” Given that Black people held in prison tend to serve longer sentences, a widespread reliance on jails for “admissions” in 2020 would cause the Black share in prisons specifically to rise even if the overall racial composition of those confined post-conviction in prisons and jails didn't change (since if Blacks serve longer average sentences, whites make up a larger share of prison admission cohorts than they do of prison populations as a whole). Accounting for jail populations in general will be hard, and measuring the prison-admission-hold part even harder, but it is an important part of the story here.

We've thought about this comment quite a bit—and we're cautious of being too dismissive of this important point—but there are a couple of caveats that we would like to push back on. First, we agree that jails likely played a crucial role in exactly the way Reviewer 1 suggests: diverting people who would have otherwise been admitted into state prison, thereby causing an overflow in the county-run jails after conviction. However, we suggest that due to the heterogeneity in how this played out across different states, it would not be wise to devote a large section of this work to untangling these mechanisms (indeed, this phenomenon deserves its own deep dive). Instead, for this paper, we should use data from county jails only insofar as it lets us rule in or rule out alternative hypotheses that may change our overall understanding of this process.

Reviewer 1 is correct in describing the dismal state of standardized data for county jails (esp. for data disaggregated by race, sex, etc.); we do not have the same quality data as we do for state prison populations as for jails. However, to explore the hypothesis presented above—that a substantial disruption to typical prison admissions was from the buildup of people in county jails (post-conviction) who were awaiting their transfer to state-run prisons—we can draw on data from the Jail Data Initiative at the NYU Public Safety Lab (<https://jaildatainitiative.org/>), shown below. Here, we use data from over 1500 counties (averaged based on the counties' urban-rural designation) to look at the incarceration rate and how it changed over the early pandemic period.

Broadly, we would have expected to see the signature of state prisons’ refusal to admit people from county jails who have already been convicted. However, in counties across the United States, the average jail population dropped by about 25% of their pre-pandemic levels. (Note: In the figure above, we’ve broken this down by counties’ urban/rural designation to highlight the broad similarities in the jail population declines, regardless of urban/rural differences that are known to be associated with incarceration rate). While this is not an entirely comprehensive analysis (e.g. it’s possible that if state prisons were *not* refusing transfers, we could have seen reductions of 30%, 40%, etc.), we do think it addresses the question of whether our interpretation of the phenomena we’re observing needs to change substantially.

A second point is that, despite occurring at a reduced volume (especially around April 2020), people were still being admitted into prisons. For example, we can identify new admissions in the Texas High Value datasets that we used, based on the listed sentence dates and when individuals were added to the dataset. A natural question arises, then, which is who is being admitted during this period? One way of measuring who is being admitted is through the newly added figure A.16, subplots E-H, where we show the average sentence length of people who were admitted by month. Across all low, medium, high, and highest offense severity categories, we see a spike in the average length of sentence of people who are new admissions towards the start of the pandemic. This is primarily influenced by the small sample sizes (admissions were in the 100s), but there were nevertheless prison admissions. Note as well, we can plot the race of the people admitted during this period, and we see the expected spike in the percent of newly admitted people who are Black (shown below).

A final point in this discussion about admissions into state prisons (though not exactly measuring admissions, per se), the Florida courts data show that despite an almost complete cessation of jury and non-jury trials in April and May 2020, pre-trial guilty pleas still made up over 70% of the total disposed cases.

Since our overall framing is quite general—that the observed bias from Figure 1b may be due to biases in admissions, biases in releases, or both—we don’t feel that the county jail mechanism rules any of that out, though it certainly adds an important nuance to the mechanism. We note this in the Discussion section, and we highlight Reviewer 1’s point as an important area for future work.

3. The 2021 reversion. The result in Fig. 1 is dramatic—but two parts of it are dramatic: not just the increase in 2020, but the reversion almost back to the 2019 ratio that takes place in 2021. While the authors say on p. 5 that they will talk about this reversion in more depth later in the paper, I did not see any subsequent discussion of it. The authors talk in depth about trends in policing, court cases, and prison releases, but as far as I could tell nothing in detail about admissions (and the court processing section, which is the one most closely related to admissions, does not discuss the reversion).

The authors suggest, on p. 5, that the reversion is most likely the result of new inflows in admissions as the jail holds get lifted—which would reinforce my point above that it is essential to figure out a way to incorporate jail populations here as well. Regardless, the sharp and sudden reversion almost back to pre-Covid trends while we were still very much in the midst of Covid is something that needs more discussion and analysis.

This is astute, and since it was brought up several times in the reviews, we've added much more discussion in the main text (also, the note on page 5 now references where we discuss this more—thanks for pointing that out). Specifically, we added a passage about this reversion in the discussion, as well as when the observation is first introduced.

4. The important role of counties. While we tend to think about prison populations as state populations, in many ways they are county populations. While it is true that prisons are run by the state government, the path to prison comes from being arrested by city police or county sheriffs, charged and (almost always) taking a plea from a county prosecutor, and then being sentenced by a judge whose courtroom is generally county-specific. If, then, during the pandemic, urban counties kept cases moving faster than rural counties—due to better infrastructure at every level—and were more likely to send those convicted to prison rather than holding them in local jails, then some of the change in the disparity in prison populations could be the result of systemic differences in county-level institutional capacity and competence. Often state-level data masks important intra-state county-level heterogeneity, some (or much?) of which could be central to the causal mechanisms at play here.

For much of this response, we point to our longer discussion of county jails above, but there's a few additional points we'd like to bring up after looking at other datasets. First, we note that in several states' court closures, the directive was statewide, meaning that all county courts followed the same orders. Second, using data from COVID AMP—a repository that aggregates time series of policy responses during the pandemic (<https://covidamp.org/>)—we theoretically can ask whether it is actually the case that, for example, urban counties kept moving cases faster than their rural counterparts. Here, we are limited by the fact that “court closures” are not listed as one of the policy subcategories, but we can nevertheless get a heuristic sense of how prudent / restrictive a county's COVID response was. Again we see that, for most counties with policy data in COVID AMP, *urban* counties had COVID-based restrictions (e.g. public service closures, essential and non-essential business closures, etc.) that last for longer periods of time and are stronger / more stringent. Note here that we classify urban using Rural-Urban Continuum Codes of “large central metro” and “large fringe metro”.

Relatedly, we can use other forms of digital trace data to estimate the extent to which certain types of counties are “back to normal” relative to others. Here, we point to recent work that our group has done about mobility and social contact patterns during the early months of the pandemic. Briefly, we analyzed peoples' mobile device data to estimate commuting patterns, mobility range, and social contacts. Below, we attach a plot that looks at these quantities, averaged at the county level and separated by urban/rural designation.

We see that urban counties experienced larger declines in mobility and social contacts, whereas rural counties returned to their pre-pandemic levels (the 100% horizontal line above) relatively quickly after the start of the pandemic. Of course, this plot again does not directly address the difference in court proceedings of rural vs. urban counties, but it contributes to a broader understanding of how quickly counties returned to normal after the initial set of restrictions in March 2020—a key part of why we think states resumed their pre-pandemic demographic trends after 2020.

5. Lack of state-specific analogs to Fig. 1. In the data appendix, the authors provide state-specific trends in state prison populations. But they do not provide similar graphs that replicate Fig. 1 at the state level. Given the concentrations of prison populations across states, it is important to provide this data as well. At the end of 2019, just three states held over 30% of all people in US prisons, and just nine held just over 50% of all people in prison (which should not surprise us, since those same states held 27% and 50% of the nation’s overall population). This raises the very real risk that the results we see in Fig. 1 are driven by just a handful of states that hold the bulk of the nation’s prison populations (especially given the relatively small absolute value of the change: less than one percentage point, or about 2.5% from the 2019 baseline). This sort of concentrated-effect issue is actually a common problem with data about the criminal legal system. About 30% of all life sentences are just in California; historically, about five counties accounted for over half of all children sentenced to life without parole; in the 2000s, the BJS had to report admissions to prison off parole separately for California, because California’s large and idiosyncratic results had a huge impact on the national average. I think it important for the authors to provide a state-level breakdown of Fig. 1, so readers can better see what the full national story looks like in a system where populations are highly concentrated in just a few states, and where effects can be quite heterogeneous.

We appreciate reviewer one for raising these points, which will be addressed in two parts. First, we added Figure A.2, which contains the state-specific analogs to Figure 1. For several states

with either low numbers of incarcerated Black people (e.g. Hawaii, Idaho, South Dakota, Vermont, etc.) or states that don't report Black as a racial category (e.g. Michigan only reports white and non-white), we plot the percent of incarcerated people who are non-white. For added focus, we plot 24 states with the most consistent data (monthly data, multiple race categories, large prison populations) in Figure A.3. We add a brief discussion of this in Section A.1.

Second, in Figure A.1 and A.2, as well as A.5, we have added additional analyses about the data we recently acquired from the Federal Bureau of Prisons (FBOP). Apart from being a useful validation of our main observation, we think the FBOP data is unique in that it necessarily comprises prison populations from different states instead of relying on data from a single state's prison system. An example of the FBOP data is shown below.

- The case studies. A corollary to that last bullet point is that I am concerned about what conclusions we can draw from the authors' case studies. The authors rightly note on p. 10 that the "heterogeneity in the structures of criminal justice systems across states" makes it hard to draw firm conclusions from state-specific case studies. I fear that this heterogeneity may actually be quite a big impediment—and not just across states, but within them too. For example (as I explain in more detail below), state highway patrol troopers have significantly different jobs than city police officers, and face substantially different populations and types of calls. So not only might it be hard to generalize from Texas to, say, New York, but it's likely hard to generalize from the Texas Highway Patrol even to the Austin Police Department.

At the very least, the authors should give some sense of whether these states are at all representative of the country as a whole for the factors they are using them to illuminate. If someone used, say, California in the 2000s as a case study in prison admissions off

parole, or Philadelphia as a case study in juvenile life without parole sentences, it would give a significantly misleading view of the country as a whole, since each of those was a major outliers from whatever sort of general national trends existed.

A related concern I had with the case studies is that I got the sense—and I could very well be wrong here—that they were chosen in part because they had the sort of granular data necessary for the case study. If so, this raises additional concerns about non-representativeness, since it is important to ask why, given how generally shoddy and unavailable data on the criminal legal system is, these specific states happened to gather and make public these specific sources of data. It could be for reasons uncorrelated with what the authors are examining, but if so, the authors should make that clear; if not, they should try to give readers a sense of the ways in which the case-study states may be outliers. (As a general matter, I fear that disclaimers like the ones they provide, saying that these case studies should not be read as generalizations, are ineffective absent some sort of explicit discussion about the risks these specific case studies face. Without that, I fear readers will too often instinctively take the case study to be at least roughly representative.)

This is a great point that caused us to make several data additions and more exposition in the main text. First, we added more justification as to why we chose a case studies-based approach to interpret the observations we document. These problems are multifaceted, ones that will require extensive further work that must include a broad, inter-disciplinary set of researchers to disentangle (which, we should note, is why we think *Nature* is the ideal venue to publish this work). We wanted to approach this complex problem with as generic hypotheses as possible (e.g. suggesting that the bias observed in Figure 1b could be due to some combination of disruptions in court proceedings during the pandemic, changes in how/who is policed during the pandemic, bias in who is released); from there, we sought out specific states who report data that could offer evidence for each hypothesis. At that point, we have provided evidence that rules-in a given hypothesis. In such a data-sparse field, where we cannot access comprehensive data for every state, this approach allows us to at the very least rule-in hypotheses that then serve as the starting point for further research. There will surely be philosophical differences about how many case studies are enough and/or when to stop looking for more mechanisms, but we took deliberate care to present zoomed-out, general hypotheses about what might contribute to these trends.

Putting aside questions about the burden of proof in this work, we do appreciate the concerns regarding some of the case studies. In response to these comments, we have incorporated a very similar dataset to the Texas Highway Patrol data, this time from California. We added a description of this dataset in SI Section A.3.4, briefly introducing how the California Racial and Identity Profiling Advisory Board (RIPA) requires each state and local agency to report traffic

stop incident data to the attorney general, who then posts the data in bulk. Put simply, we find the same effect as in Texas—if anything, California shows a much more pronounced spike.

IMAGE REDACTED

Additionally, we downloaded data about traffic stops from the New Jersey Office of the Attorney General (<https://www.njoag.gov/trafficstops/>), which is similar to the Texas and California data, but is 1) aggregated to the monthly level and 2) includes additional information about what happened *after* the traffic stop. Because of this, we can subset the data to only include traffic stops *that result in summons*. Below, we plot data about the racial breakdown of traffic stops that resulted in summons for the drivers, highlighting once again that there is a sudden and sharp increase in Black and Latino drivers' share of the police interactions, this time extending that finding to include legal summons.

IMAGE REDACTED

As for how to incorporate this specific finding into the paper, we are of two minds about it: On the one hand, this result is useful when justifying why one might associate changes in the racial distribution of who interacts with police with changes to the racial makeup of prison populations, however slight this association may be. On the other hand—as we discuss in the following comment—focusing too much of our attention on the New Jersey example risks losing the broader message of why we use traffic stops data in the first place. Namely, it is a clear example of broader, systemic forces that influence policing (and who is “eligible” to be policed) in the United States: To put it bluntly, you’re not getting pulled over when you’re working from home.

I had a few more-specific concerns as well.

7. Texas Highway Patrol Data. If the focus of this paper is on what drives prison populations, or what drives the confinement populations of those convicted of felonies (to account for the jail-hold issue), I’m not sure trends in highway stops are particularly informative. The authors correctly note on p. 11 that “traffic stops are not the only way for people to enter the judicial system — nor are they the primary source of prison admission...” but that last phrase understates things. Given the comparatively-serious nature of offenses that lead to prison admission, traffic stops likely play a very small role in those admissions. It’s also worth noting that the shocks of Covid make it hard to interpret what, exactly, to infer from the THP data. As the authors correctly note, the pandemic led to a significant shift in who was commuting, and in a direction that surely led to an increase in the share of Black drivers. This would suggest that even a race-blind THP would stop a growing fraction of Black drivers during the pandemic. To be clear,

I'm not saying the THP was race-blind! But to understand the dynamics of systemic racism, especially for the (prison-centric) question this paper is addressing, it is important to separate out what is being driven by the criminal legal system itself vs. how that system interacts with other structural problems (like how Black Americans were less likely to have work-from-home jobs).

Related to our response above, we want to re-emphasize the role that the traffic stop data has in this case: We hypothesized that one possible mechanism contributing to the sudden shift in prison population demographics could be abrupt changes in who is interacting with police and/or how those interactions are taking place. And while we're also not claiming that the THP was race-blind, in a sense it strengthens our argument *if they were*. As for delineating what is being driven by the criminal legal system vs. what is being driven by structural biases, we hope to convey a message that is somewhat adjacent: The goal of the paper is to use the pandemic as a novel lens through which we can understand broader structural biases in society. This is somewhat of a subtle point, which multiple reviewers touched on, and as a result we have put in some work to make sure the point is clearly stated up front. To put it another way, we did not set out to argue "If only this one policy were changed, then we would not have seen the worrying effects in Figure 1b..." Instead, we are pointing out a variety of ways that our large-scale societal response—together with existing race-based, structural, systemic biases—interact in order to apply the justice system disproportionately based on race.

8. Arkansas parole data. On p. 9, the authors state that the percentage of whites released early in Arkansas under Covid rules was greater than their share of the prison population, something "we would not expect to see in a prison system absent biases in sentencing." This is not necessarily true. Many states' Covid early-release rules applied only to people in prison for non-violent crimes, and at least at the national level, at the end of 2019 (the year with the most current offense-specific data), 64% of Black people in prison were serving time for a violent conviction, vs. 50% of white people in prison (see Table 14 in the BJS's Prisoners in 2020 report, which has offense-level data at a one-year lag). Now, of course, it is true that Blacks receive longer sentences than whites for the same crimes, that the same conduct is more likely to lead to a conviction for violence (or a more serious degree of violence) for Black person than a white one (due to plea deals offered), and that racial disparities in violence (which are unambiguously true for murder, though the data gets much more nebulous after that) are inarguably the product of systemic racism as well.

But from a policy perspective, nailing down the relative importance of these various mechanisms is important. For example, if the racial disparities in Covid releases is due to racial bias within the pool of those eligible (which was rarely if ever the entire prison

population, given the restrictions often imposed on releasing those convicted of violence), then the solution is to, say, figure out a way to limit problematic discretion. If the problem is in who is in the eligibility pool—if releases are proportionate to the racial makeup of those eligible for Covid release, and the disparities reflects differences in who is eligible—then addressing bias at the decision-making level will do little, and we need to focus instead on determining what the eligibility pool should look like. This latter question, however, becomes increasingly tough to answer, both politically and normatively, the more the key issue is disqualifying those with violent convictions.

Again, a very astute point, which we agree with. We have added language that echoes these sentiments directly, pointing once again to the fact that the process itself—much like the THP data—could very conceivably be race-blind. As shown in the newly added Figure A.16 (subplots A-D) using data from Texas, for people with low/medium/high offense severity charges, we see that Black people are more likely to be in prison for a longer sentence (note: the figure does not show a sentence-by-sentence comparison based on race, simply a comparison of the average sentence lengths by race and offense severity). This translates to Black people in prison, on any given day, having a lower probability of being released. Coupled with stricter requirements about COVID-19-based release criteria, as in the Arkansas example, it is clear how a race-blind policy can generate outcomes that differ according to race. *But*, again, we emphasize that the strength of this work lies in its ability to use the pandemic and the changes that it brought about as a lens through which to view long-standing, structural racial biases in this country.

9. Texas severity data. I don't think the Texas severity data can actually answer the question of whether the spike is due to a change in offending patterns across race. That answer requires us to look not at differences in the crime severity of people sent to prison, but at differences in the crime severity of those who are or are not sent to prison. That the relative increase seems to come at the lower end of the sent-to-prison severity does not rule out the possibility that there were racial differences in those committing felonies that merited any sort of prison time.

To be clear, I don't think changes in offending can explain the spike the paper shows for Texas, either. If nothing else, at least in the short run, changes in offending drive prison populations through admissions, and there is inevitably a lag—likely often several months—between offense and admission. So an admissions-driven spike right at the Covid outbreak would be the result of offending changes months before. Not only would that sort of precise spurious-correlation timing be surprising, but it's not clear there was any sort of spike, at least in the major Texas cities, at the right time (See, for example, the seven-day moving averages for all violent incidents in 2019 and 2020 for Austin, Dallas, Houston, and Fort Worth, online at <https://citycrimestats.com/covid/> (unfortunately, it is

impossible to provide a direct link to the specific set of graphs for those four cities, so the authors will need to recreate them themselves).). But I still think it is also important to note that the Texas data given in the paper does not in fact refute the possibility that changing offense patterns could explain the 2020 disparity increase in Texas.

This is a good point. We also don't think the Texas severity data necessarily answers the question of whether the spike is due to a change in offending patterns—that also wasn't the reason for its inclusion. Re-reading our manuscript, we realized that this was not well-explained on our part, and we really appreciate this point. To clarify, this was another example of a dataset used to rule in/out a given hypothesis. We were hoping to preempt a common reply about how these observations could just be due to racial differences in crime severity. We added discussion that echoes the points made here, similar to our response in comment #6 and #7.

Additionally, we have added new analyses looking specifically at sentencing using the Texas data. In short, starting after March 2020, the average person in prison in Texas had a longer sentence than the average sentence prior to the pandemic. This is due to two main factors: 1) An abrupt drop in the number of admissions without 2) a commensurate decrease in the number of releases. This means, on average, that someone who is still in prison after March 2020 is someone who is more likely to have been assigned a longer sentence. This, coupled with the fact that in any given month white people make up about 55% of the releases in Texas (despite being only 33% of the overall population) contributes to a broader mechanistic story about the Texas prison population. Below we're adding the new Figure A.16, which shows a comparison of average sentence lengths in Texas, by race and offense severity. In the top row of subplots A-D we plot the average sentence length of the incarcerated population in Texas state prisons (excluding life-sentences and sentences over 80 years), by race and offense severity. In the bottom row of A-D, we plot the change relative to pre-pandemic values (here, 100% indicates no change). For all offense types, we see increases in the average sentence length at the start of the pandemic due to the drop in monthly prison admissions without commensurate decreases in releases. In subplots E-H, we show the average sentence length of new prison admissions, by offense severity and race.

Case study: Offense Severity Data, Texas — Sentence length and admissions

As I mentioned at the top of this report, I think the question the authors are asking is an incredibly important one. But I also fear that it collides with several significant challenges that arise in almost any paper trying to use administrative data on the criminal legal system. I hope that the authors are able to figure out ways to address these issues, although in some cases I fear that there may be no real fix—the authors may need to think instead about ways to express the uncertainty in their results that arise from the inescapable systemic errors in this sort of administrative data.

A quick personal thank-you: These were very critical reviews, and even after substantial edits there will still be some issues that remain under-addressed or left for future work. But it is a really valuable exercise to have someone read this work as closely as you did. We learned a lot from the comments provided, and we look forward to hearing any further thoughts.

Reviewer 2 -- Nature 2022-01-00306

This paper documents a new fact: the Black and Latino share of the state prison population increased immediately following the onset of the COVID-19 pandemic. To show this, the authors collect data on the prison population in all 50 states and the District of Columbia. Using more detailed data from three states (Florida, Texas, Arkansas), the authors examine three potential explanations: disruption in court operations, changes in police interactions, and decarceration.

The new fact that the paper documents is important and interesting. My main concern with the paper is that, because of limitations of the data the authors use, the paper does not provide much concrete evidence on the underlying mechanisms that produce this pattern. As the authors emphasize, understanding the mechanism is important to understanding what to make of this pattern moving forward. The authors focus on and present state-specific case studies for each of three mechanisms: disruption in court operations, changes in police interactions, and decarceration. However, in each case, it's not clear from the data if that mechanism is empirically important for explaining the stylized fact of interest and, if so, why, or what that tells us about the nature of structural racism in criminal justice. Moreover, a more detailed and informative examination is feasible because the more detailed data required are available to researchers, at least in some states (though perhaps with some lag).

Thank you for the deep read and the critical comments provided below. We have responded to each—including some that required the collection and analysis of new data—and we believe that the inclusion of these new findings and general edits have greatly improved this work.

I expand on these points for each mechanism below.

Disruption in court operations

The authors use monthly data on case dispositions in Florida Circuit courts and show that the number of dispositions declines sharply in March 2020, with a corresponding increase in the backlog of cases. This drop in dispositions coincides with an increase in the nonwhite share of the prison population.

Unfortunately, as the authors point out, without more detailed data—including data on the race of defendants—it is difficult to understand why a decrease in dispositions would increase the nonwhite share of the prison population. Is it that white defendants experience a disproportionate increase in case dismissals? Are white defendants more likely to have their cases delayed or to receive non-confinement sentences (relative to pre-pandemic)? Can racial differences in treatment be explained by racial differences in the distribution of the charges (e.g. charge severity)?

One hypothesis is that new prison admissions are disproportionately white relative to the prison population because white admits tend to have short prison sentences (either due to differences in charges, or differences in sentencing conditional on charge severity). The same would be true for releases (see Figure A7). If the pandemic stopped new admissions and accelerated releases, those left in prison would tend to have long sentences, a population that may be disproportionately nonwhite.

It is important to note that the kind of detailed microdata required to answer these questions are available in several states (for example, Lee and McCrary 2017; Feigenberg and Miller 2021; Rose and Shem-Tov 2021). It's possible that there would be some lag in accessing these data relative to the data used here, but I think the ability to unpack mechanisms would be well worth the wait.

These are great points, and they have influenced some of our edits and new analyses a great deal. Thank you. To summarize our longer response:

1. We received an incredibly rich dataset from Florida via public records request. As alluded to in our manuscript (but—as Reviewer 2 correctly points out—not directly shown), there was a spike in the proportion of dismissed cases of white defendants.
2. We did a deep dive into the relationship between differences in sentencing based on race and how that may relate to the relative increase in the Black incarcerated population. Reviewer 2's hypothesis is spot on: when there are underlying differences in sentencing length based on race, *mere reductions in admissions* are sufficient to qualitatively reproduce the main results (e.g. in Figure 1b). We show this through a more in depth analysis of Texas data as well as a brief modeling/simulation exercise.

First, Florida: We requested data from the same office that provided the Florida trial statistics data for Figure 3, and we think these new data directly address the comments above. The Office of State Court Administrator (OSCA) in Florida provided us with data from the Criminal Transaction System for 2018, 2019, 2020, and 2021; these data contain a column for defendants' race, the action taken by the court, and the date each case was decided (among many other variables). [Note: OSCA stressed to us that they do not create the dataset, they are just the stewards of the dataset; clerks of the court record each defendant's case data via the Offender Based Transaction System. OSCA is the organization that compiled the data and are not the custodians of the data. They are also requiring us to report a disclaimer that “Any conclusions or analysis that will derive from this dataset are solely those of the individual author(s) or the person(s) who did the analysis and not of OSCA.”]

Disclaimer aside, this dataset allows us to run a simple analysis: Among defendants with cases that were dismissed, between 2018-2021, what percent are recorded as white? We plot this in subplot (B) below—subplot (A) is Figure 3C reproduced; subplot (C) shows the two curves atop

one another, rescaled using min-max scaling in order to highlight the timing and relative increase that both measures show after March 2020.

(trial data from trialstats.flcourts.org; defendant records from OSCA)

Next: With respect to the comment about the sentence lengths of people newly admitted into prison, we dove a bit more into the data from Texas to get at this question, and we’ve added Figure A.16 (A-D shown below) to the Supplemental Information. To walk through what this figure shows: On the top row, we plot the average sentence length of the incarcerated population in Texas state prisons (excluding life-sentences and sentences over 80 years), by race and offense severity; below, we plot each curve’s relative change since March 2020 (100% indicates the same average sentence length). We see that after March 2020, the average person in prison in Texas had a longer sentence than the average sentence prior to the pandemic—with Black people typically serving sentences that are 1-2 years longer than white people, depending on the offense severity.

Offense severity data from: https://www.tdcj.texas.gov/bpp/parole_guidelines/Offense_Severity_Class.pdf

As astutely mentioned in Reviewer 2’s comment, “One hypothesis is that new prison admissions are disproportionately white relative to the prison population... If the pandemic stopped new

admissions and accelerated releases, those left in prison would tend to have long sentences, a population that may be disproportionately nonwhite.” This is particularly insightful and it is indeed what we see in Texas, where—even if releases were not accelerated, which we did not find evidence of in Texas—reduced admissions *alone* can increase the average sentence length of people in prison, which would in turn lead to a relative increase in the share of Black people in Texas prisons. Put another way, on average, someone who is still in prison after March 2020 is someone who is more likely to have been assigned a longer sentence, and someone who—in Texas—is more likely to be Black. Note that this is still the case even if we do not observe differences in prison admissions based on race during the pandemic: fewer admissions overall, coupled with race-based differences in average sentence length, is enough to produce the effects we observe. In any event, we *do* see differences in admissions based on race during the pandemic in Texas (figure below).

A final note about the key role of sentencing here: Because we observe the relationship between reduced admissions and differences in sentence length by race, we are now in a position where we can *model* these dynamics via simulation. Specifically, using Texas as an example, on average, we know: 1) how many people are admitted into prison each month, by race, 2) the distribution of sentence lengths of newly admitted people, by race 3) how many people are released from prison each month, by race, and 4) the total number of incarcerated people, by race. We can simulate admission-release dynamics in the following way: each month, N_i people of race i are added into the prison population, each of is assigned a sentence of length l_i months, which is a number drawn from the typical distribution of sentence lengths for race i (according to the data). Using these data, we construct and run a simulation where people are released each month, if they have been in the simulation for longer than their sentence length. While holding relative number of admissions and releases constant by race, we can look at the expected effect of *merely* reducing admissions (e.g. by 50%, 60%, etc.). If we match the reductions in prison

admissions to exactly match what was observed in Texas during 2020 (and then resume to fixed average monthly admissions), we see the following (right) compared to the true data (left):

While this simple simulation is able to reproduce the qualitative effect (the relative increase in the proportion of Black people in prison), it does not reproduce the magnitude of the effect. On the one hand, we would not expect such an unsophisticated model of prison population dynamics to exactly reproduce the observed data; on the other hand, this also points to other potential sources of bias that we did not include in our simulations (e.g. skewed admissions by race, as shown in the previous figure two-above).

Ultimately, this proved to be a promising exercise, which ended up validating Reviewer 2’s instinct. We are hesitant to put this whole analysis into the main text, as we fear bloating the main observations, but we did use these insights to reframe some of our hypotheses, strengthening the case for the sentencing story.

A side note: the authors note that they “observe the same correlations between percent of dismissed cases and percent non-white incarcerated people prior to the COVID-19 pandemic”. I thought this was an interesting point, but could not figure out where it was demonstrated in the paper. It would be helpful to document this finding more explicitly in the paper.

This was indeed under-explained in the figure and text, and we have adjusted the figure caption and text to reflect it. The point we were making was what is in Figure 3D, which is that there is a strong positive correlation (at a 1-month lag) between the percent of total cases that are dismissed and the percent non-white incarcerated population.

Changes in police interactions

The authors use data on traffic stops conducted by Texas Highway Patrol to understand how changes in policing and police interactions with civilians may have contributed to changes in the

racial composition of prisons. They show that the percentage of traffic violations that involve Black and Latino drivers increases significantly after March 2020.

An important shortcoming of this analysis is that only an extremely small percentage of traffic stops lead to a prison sentence, so it is not at all clear how traffic stop patterns are related to the motivating stylized fact. If there was some meaningful change in policing that contributed to an increase in the nonwhite share of prisoners, these would not be the data to show it. It is true that, compared to court records, it is much less straightforward to obtain data on policing. But it would be possible to get data on 911 calls and stops conducted by city police departments (rather than highway patrol) (Hoekstra and Sloan, forthcoming).

Since this point overlaps quite a bit with one of Reviewer 1's comments, we'll reiterate and expand on a few of the changes we made in response. However, a brief digression: we went through the Hoekstra and Sloan piece, which offers another interesting and compelling angle on this topic—one that, we feel, would similarly benefit from honing in on the time frame from 2020 to 2022. Reiterating one of our responses to Reviewer 1, the pandemic and societal disruptions that followed from it gives us a relatively unprecedented lens through which we can view existing (structural) disparities in the United States—in essence, using the pandemic to see the often hidden structural forces that existed well before the pandemic. We see the question of how policing, use-of-force, etc. (*a la* Hoekstra & Sloan, 2022) changed or was potentially exacerbated during the pandemic as another one of these questions that can tell us a lot about the nature of policing, race, and justice *in general*—as opposed to simply being confined to March 2020 onward. In any event, we did make specific changes and additions based on these and similar comments (copy/pasting some of our responses to Reviewer 1 below):

“...In response to these comments, we have incorporated a very similar dataset to the Texas Highway Patrol data, this time from California. We added a description of this dataset in SI Section A.3.4, briefly introducing how the California Racial and Identity Profiling Advisory Board (RIPA) requires each state and local agency to report traffic stop incident data to the attorney general, who then posts the data in bulk. Put simply, we find the same effect as in Texas—if anything, California shows a much more pronounced spike.”

IMAGE REDACTED

“Additionally, we downloaded data about traffic stops from the New Jersey Office of the Attorney General (<https://www.njoag.gov/trafficstops/>), which is similar to the Texas and California data, but is 1) aggregated to the monthly level and 2) includes additional information about what happened *after* the traffic stop. Because of this, we can subset the data to only include traffic stops *that result in summons*. Below, we plot data about the racial breakdown of traffic stops that resulted in summons for the drivers, highlighting once again that there is a sudden and sharp increase in Black and Latino drivers’ share of the police interactions, this time extending that finding to include legal summons.”

IMAGE REDACTED

“As for how to incorporate this specific finding into the paper, we are of two minds about it: On the one hand, this result is useful when justifying why one might associate changes in the racial distribution of who interacts with police with changes to the racial makeup of prison populations, however slight this association may be. On the other hand—as we discuss in the following comment—focusing too much of our attention on the New Jersey example risks losing the broader message of why we use traffic stops data in the first place. Namely, it is a clear example of broader, systemic forces that influence policing (and who is “eligible” to be policed) in the United States: To put it bluntly, you’re not getting pulled over when you’re working from home.”

A side note: the authors reference a 2015 investigative report documenting that Texas officers were systematically misreporting Latino drivers as white and suggest that the traffic stop data may under-estimate the true proportion of traffic violations. My understanding is that the Texas Department Public Safety responded to this revelation by requiring officers to ask drivers to self-identify, which sharply increased the recorded Latino share of stopped drivers (Luh 2020).

Good catch, and thanks for the citation! Looking at the data, we do see much more stability post-2015 in the data (figure below, if interested—also highlights even more how relatively out of the ordinary the spike is what we show in Figure 4B) We’ve added clarified language around this dataset, and have included a short mention of the Luh 2020 finding.

Decarceration

Here the authors show that the population of incarcerated people that were eligible for early release in Arkansas is disproportionately white compared to the overall prison population. It would be helpful to know what the criteria for early release in Arkansas were in practice. For example, the authors state that “eligible incarcerated people needed to have a parole plan in place”. How does one get a parole plan in place? Are these incarcerated people that were nearing the end of their prison sentence? In that case, it would be important to document whether racial differences in early release simply reflect racial differences in average sentence length, which may in turn reflect racial differences in the composition of charges. Consistent with the idea that this pattern reflects differences in average sentence length, the fact that those released from incarceration are disproportionately white appears to not be unique to the pandemic (Figure A7).

These questions could be answered with a more detailed accounting of individual-level prison admissions and releases, which could be done with available state department of corrections data on the prison population (potentially including the bulk data from Arkansas that the authors reference).

This is another key point, which gets again at a broader theme in this work. That is, it is possible for states to require seemingly straightforward—even “race blind”—criteria to be eligible for early release *and still* observe bias in who gets released. As Reviewer 2 mentions above, this could arise simply due to differences in sentencing, under the assumption that sentence length is a proxy for some of the release criteria (e.g. offense severity). Ultimately, we use this to argue in favor of our broader thesis that *this* is one of the levels that structural, systemic issues operate at. Unfortunately, the data we have for Arkansas is not as rich as our data for several other states,

and we were unable to get our hands on the same level of granularity. However, we did add discussion of this point, expanding on how exactly this is evidence of broader, systemic issues.

References

CJARS, 2022. <https://cjars.isr.umich.edu/data-documentation-download/>

Feigenberg, Benjamin and Conrad Miller, 2021. “Racial Divisions and Criminal Justice: Evidence from Southern State Courts”, *American Economic Journal: Economic Policy*, 13(2).

Hoekstra, Mark and Carly Will Sloan, forthcoming. “Does Race Matter for Police Use of Force? Evidence from 911 Calls”, *American Economic Review*.

Lee, David S. and Justin McCrary, 2017. “The Deterrence Effect of Prison: Dynamic Theory and Evidence,” *Advances in Econometrics*.

Luh, Elizabeth, 2020. “Not so Black and White: Uncovering Racial Bias from Systematically Misreported Trooper Reports”, unpublished manuscript.

Rose, Evan K. and Yotam Shem-Tov, 2021. “How Does Incarceration Affect Reoffending? Estimating the Dose-Response Function”, *Journal of Political Economy*, 129(12).

Reviewer 3 -- Nature 2022-01-00306

I appreciate the opportunity to review this manuscript. It is well-written and understandable, even for non-specialists. This paper can serve as a model for using the Covid-19 pandemic as a natural experiment for testing theories related to crime and criminal justice. As the authors argue, their creation of the unique dataset is a contribution in itself. The core argument, that whites benefited more than Blacks from the decrease in the prison population during the first year of the COVID-19 pandemic, is interesting and important, and I think it would draw widespread attention from academics, journalists, legal practitioners, and advocates.

We are humbled to hear this appreciation of our work, and we are motivated to continue pulling this thread at the intersection of public policy, public health, data science, and criminal justice. As the Reviewer notes, the pandemic (and its downstream consequences in our behavior, policy, and everyday life) is a useful new lens to view a very old problem. Like so many systemic status quos in our world, however, it can be difficult to see their effects without some sort of jolt to the system. In this case, the pandemic was a jolt to the typical operation / manifestation of mass incarceration in the United States. We see this moment as an opportunity to shine a new light on questions about racial bias in the justice system.

The authors' use of case studies is effective for proposing mechanisms that could have caused the racial disparity. It does appear, as the authors indicate, that disruption in the court system is the most probable mechanism. I'm curious if Massachusetts experienced similar disruption in the court system. A brief analysis of the Massachusetts case could strengthen the manuscript (if that's possible given data constraints).

Nearly every state closed its courts in some capacity during the early months of the pandemic, and Massachusetts in particular was a state where in-person court proceedings were disrupted for an especially long time. (As a rough guide, we found that states with more stringent COVID mitigation policies were also those with longer disruptions to typical court proceedings.) As you mention above, we do believe that disruptions in the court system were the core contributor to the disparities we observed (e.g. our main result, Figure 1b), and as such, we also saw an increase in the percent of Black incarcerated people in Massachusetts during the early months of the pandemic. Note: this occurs despite Massachusetts' lower incarceration rates compared to other states. We added Figure A1 & A2 in the Supplemental Information that show state-by-state trends in the total incarcerated population and the percent of incarcerated people who are Black. Strikingly, as we mentioned in the original manuscript, we see this spike in nearly every state.

The authors may want to emphasize the importance of structural inequality when discussing the decarceration case. Since only people with an "approved address to stay upon release," and a parole plan in place (which likely included having housing, job opportunities, and documentation

like a driver's license) were eligible for release in Arkansas, the policy likely privileged prisoners with at least limited social and economic resources (perhaps more white than Black people?). Also, I found the last sentence in the last paragraph on p. 9 a bit confusing. Does "biases in sentencing" mean that biased sentencing led to longer sentences for Black vs. white defendants with similar cases, and/or that Blacks were more likely than whites to be classified as "violent" (vs. "non-violent"). If I understand the analysis correctly, these are the two sentence-related factors that affected release decisions.

Thanks for these important comments. The question of structural inequality has come up in the other Reviewers' comments, and we have taken this to heart in the updated manuscript (because, as you mention, *it is so core to our story*). We have clarified two key frames based on these comments: First, we have changed the wording around "biases in sentencing" in that passage, which now reads, "...despite the fact that 57.2% of the Arkansas prison population was white, over 72% of the incarcerated people eligible for early releases were white—a disparity that we would not expect to see in a prison system absent of release policies that favored incarcerated white people. These policies may manifest in multiple ways: sentencing patterns that create longer sentences for incarcerated Black and Latino individuals, different classifications (e.g., violent or nonviolent), and other categorizations that may drive a disparity in those released. How these policies drive inequality is likely dictated by the particulars at the state level." In doing so, we have removed imprecise use of the term "biases" and replaced it with text that states the problem much more explicitly.

The second framing that has been reemphasized in this version of the manuscript is this idea of *structural* or *systemic* inequality. The point about release criteria is a terrific one that gets at this broader notion of societal / systemic differences in outcomes based on race. It also pairs well with our newly expanded discussion about systemic factors influencing peoples' contact with police during the pandemic (via the Traffic Stops data from Texas—and now California as well). Reviewers 1 and 2 both had interesting comments about how much we can interpret traffic stops as being at all associated with the demographics of people who are incarcerated. We now emphasize that we did not include the traffic stops analyses to suggest that there were sudden shifts in police officers' prejudice toward Black & Latino drivers starting in March 2020. Instead, we feel that our point is made even stronger if we assume that policing in Texas and California *is* race-blind, because the differences we observe would need to be explained by some higher-order, *systemic* difference in how race affects outcomes in our criminal justice system. In the case of traffic stops, Black and Latino workers more often are employed as "essential workers" and/or in jobs that cannot otherwise transition to remote work; conversely, white people are more likely to work remotely in every state, and therefore are less likely to encounter police during e.g. their commute, etc. This broader point about pre-pandemic differences in race and class is echoed both in the Arkansas release data as well as the traffic stops data.

Reviewer 4 -- Nature 2022-01-00306

This paper examines shifts in the racial composition of incarceration in the United States during and directly after a critical juncture – the disruptions posed by the Covid-19 pandemic. While rates of incarceration were steadily declining prior to the pandemic (as well as racial disparities in prison admissions), the pandemic created “the largest, most rapid single-year decrease in prison population in American history.” Here’s where the paper makes its boldest claim: that the sudden fall in incarceration rates occurred alongside and worsened racial disparities in incarceration “erasing much of the progress from the declines over the last decade.” Throughout the paper, they conclude that the pattern they locate – falling incarceration accompanied by rising disparity -- is due to racial bias in features of the criminal legal system: who gets released, who police stop, and who gets processed. To summarize their findings, the pandemic shifted all three, amplifying existing inequalities. Using an original data compilation of all states incarceration rates by race as well as three state case studies to explore potential mechanisms for amplified racial disparities, they argue that a combination of three factors were at work: decarceration policies that released more white incarcerated people, court closures in the early months of the pandemic, and policing interactions during the pandemic.

Central claims:

- Substantial fall in prison populations across every state during the pandemic (nationally, a 16.3% decline)
- The people who benefited most from falling admissions and releases were white
- Thus, the relative number of black incarcerated people increased temporarily
- The race-based patterns cannot be explained by crime severity

The paper makes an important contribution to our understanding of racially disproportionate incarceration and delivers extremely compelling findings. The authors have done an incredible service to our fields by constructing this dataset! This was effortful and entailed filing FOIA requests across multiple states where the data by race were not publically available, cross-checking data against existing measures for accuracy, compiling monthly data on trial statistics in Florida to dig deeper into mechanisms, navigating incredibly thorny issues related to the data, and anticipating possible errors. A reader can see a full listing of how the data were collected in Table A.1 (and one quickly sees what a patchwork of different state practices on how race is categorized and classification systems). The resulting figures and measures and Appendix display considerable care and precision. The authors provide back of the envelope calculations to show how much more the black prison population would have decreased if it matched the white reduction in incarceration. In short, over 14,000 fewer black people would have been incarcerated.

The authors appreciate the praise, but much more importantly, appreciate the care and thoughtfulness of this summary. It is reassuring to learn that the Reviewer 4 understood our intentions and gravitated to our attempts to communicate the story transparently. Before we get into details, we must say explicitly that these comments have been incorporated into many aspects of our study and have significantly improved our manuscript.

In our response, we will address and elaborate a few points about the thorough description you provide, but by in large we agree with the hypothesis and believe that our dataset reflects the three factors Reviewer 4 identified: “decarceration policies that released more white incarcerated people, court closures in the early months of the pandemic, and policing interactions during the pandemic.”

Below we will respond to various points, and highlight places in the revised manuscript where Reviewer 4’s comments influenced the structure of our arguments.

Because the argument is such an important claim and set of findings, I have several ideas for ways the paper could improve and really hone in on its most important findings.

The paper would do well to actually frontload the main finding: a dramatic surge in black composition of the prison across every state, despite (or perhaps because of) a decline in the scale of incarceration. This is a novel finding and very troubling. It is remarkably consistent across states, regardless of whether the fraction of black incarcerated people was increasing, stable, or decreasing in a particular state prior to the onset of the pandemic in March 2020. Unearthing the complex mechanisms for this is a very difficult undertaking and the authors did their level best to unpack possible explanations and examine each with fascinating mini case studies (case studies because the data for every state do not exist and they are right to end with a call for states to release more and better data on race). I want to emphasize here as a scholar of incarceration just how difficult this data collection is.

We appreciate these comments and accompanying suggestions. We struggled to decide which data to prioritize, and in which order. Given the number of intriguing results in our study and space limitations, we considered many permutations for how the data should be presented to the reader.

We decided on the following structure:

- Demonstrate the nationwide finding (Figure 1)
- Show representative states (Figure 2) with a mention of the richer nationwide dataset in the supplementary information

- Dedicate main text Figures 3-5 to demonstrating the individual mechanisms that could contribute to the findings in Figures 1, 2 and A1 (and newly included A2 and A3).

We considered other frames for the manuscript but stuck with this one. However, our revision features clearer language that more clearly communicates the results. In addition, our revision contains much improved and expanded data for several of the mechanisms, especially for court closures and traffic stops. We elaborate on some of these changes (many in direct response to Reviewer 4's comments, including multiple updates that draw from new, rich sources of data) later in our response.

After staring at the figures and the text for some time, I have an alternative hypothesis for what's going on: by temporarily starving the system of a source of falling racial disparity in prison admissions, cutting off prison admissions through court closures affected the racial composition of the prison.

The logic here is a bit complex so let me explain. Prison admissions (flow) contain much less racial disparity than total prison confinement (stock). The racial disparity in prison admissions (2:1) is much lower than the black/white disparity in total imprisonment (6:1). And for several years prior to the pandemic, black prison admissions were falling dramatically from their peak in the late 1990s, while white prison admissions were growing especially among those whites without college education (though from a much lower level), which resulted in a narrowing of racial inequality. For example, the Muller & Roehrkasse data show that the admission rate among black people without college education fell from 4,494 people per 100,000 in the year 2000 to 2,511 per hundred thousand in 2015 while the rate among similarly educated whites actually grew. Let us not be confused: Racial disparities in prison admissions are still severe and black people continue to be admitted to prison more than any other group, reflecting the ongoing legacies of racially oppressive policies and practices. However, the black/white disparity in prison admissions reached its lowest level in decades in 2015.

The point about differences in disparities (2:1 vs 6:1) is now a key argument in our Discussion, which was further bolstered by the inclusion of new data and analyses from Texas. (This point is echoed in our response to Reviewer 2.) Briefly, we asked whether it is possible that the result we found could have been produced *only due to differences in sentence duration, by race*. That is: Can we simulate a synthetic prison population where each month an average number of people are admitted and released (and whose simulated race is based on actual data from Texas)? Once we have this very simple setup, we show—using actual data about sentence length by race—that a brief decrease in monthly admissions is *alone* sufficient to produce the spike in percent Black population in prison; that is, if Black people are, on average, in prison for longer sentences, any brief decrease to the rate of new admissions will trigger the spike in percent Black population. Below we're attaching the output of this simulation (right) compared to the actual

data from Texas (left). For the simulation, we based admission numbers off of the exact data from admissions in Texas.

Returning to the point about the 2:1 vs 6:1 difference, this means that even if racial disparities in admissions are much lower than that of the total prison population, we would still expect to see this trend. In the specific case of Texas, we see all of these factors aligning (i.e., a sudden change in the admission disparity). Below, we show the percent of White/Black/Latino admissions during the pandemic, highlighting the notable spike in the relative number of people admitted who are Black at the start of the pandemic.

(While this is not precisely the example you brought up, it is highly consistent with the overall hypothesis being posed.)

So, if you just stop admitting people to prison through court closures, you are effectively cutting off a key source of lessening racial disparity in incarceration. This is counterintuitive. Otherwise stated, if we stopped admitting people to prison at the rates we were seeing prior to the onset of the pandemic, which showed dramatically falling racial disparities (see Muller and Roehrkasse's 2021 Social Forces article), we would see a plateauing of the racially moderating effect of prison admissions on total prison population. Since the black/white disparity in admissions is lower than the black/white disparity in incarceration, if you just halt prison admissions, falling racial disparities in imprisonment would stagnate. If, on the other hand, covid court closures had not interrupted prison admissions, the pre-covid trend of steadily increasing admissions of low-education white people would have continued (Muller and Roehrkasse 2021). But with the pandemic (and prison populations not being replenished with relatively more low education white admissions), the declining share of black people making up the prison population would necessarily stabilize. That is a numerical property that does not have to do with intentional racial bias (though everything to do with the lasting effects of the drug war and racialized punitive policies on total confinement). To put the point more dramatically, if all courts closed again tomorrow, we should see the same effect (a slowdown in the decline of black/white incarceration disparities).

Now, what I am suggesting would only explain a flattening in the pre-covid downward trend in the black percent of incarcerated people; it would not explain why the black share of incarcerated people actually rises as the authors show.

For that, we need to turn to who is being released, which the authors do in looking at racially unequal early release patterns, and this is what I see as one of the most promising mechanisms in the paper. In combination with what I'm suggesting above, emergency decarceration orders that gave disproportionately more white incarcerated people the option to be released would have resulted in the upsurge in the share of the prison that was comprised of black people.

Again, this is such a keen insight—one that took us several months (and many gigabytes of data) to arrive at—and we are glad that we have restructured much of our Discussion to elaborate on it. Reviewer 4 gracefully sifts through larger results and points out a contradiction: that courts have actually been a source of *declining* disparity. And the manuscript that they point us towards (Muller and Roehrkasse 2021; cited in the revision) details how low-education whites have been increasing in admissions in recent years. This is an argument that we did not consider in detail initially, but we have managed to accommodate it in our revised Discussion section.

One suggestion then is for the authors to acknowledge this possibility in the paper or say why they do not agree or think it is driving a portion of the results. Can the authors show prison admissions as well as total incarceration rates and total prison population? If a big claim in the

paper surrounds who gets admitted, then why not show prison admissions by race, not just total incarceration (and black share of prison population)?

As the reviewers have astutely highlighted, our prior draft lacked important nuance around how court closures could be the main driver of the dynamics. While we stand behind the inclusion of references from the Virginia and Michigan Departments of Corrections (References 20 and 21) that implicated court closures as the main cause; however, without more granular state-level data, we feel that suggesting any one mechanism is the true driver of the main finding would be premature.

In addition, we note that we do include an analysis addressing this point in Figure A7, but we have not had the same luck getting standardized admissions / release data separated by race. As mentioned above, we did this analysis for Texas, and doing so validated this new framing about admission disparities vs sentencing disparities.

A related concern is that one of the most prominently discussed mechanisms in the paper is the idea that court closures and disruptions in prisoner intake from courts and county jails due to the pandemic exacerbated existing biases. The claim, as I read it, is this: when courts closed, the sheer number of defendants fell sharply, and in the state they have data for, “more than 99% of cases did not go to trial,” so more people agreed to guilty pleas or had charges dropped. The authors are right to say that plea deals and which people have their cases dropped are heavily racialized processes. There is no argument there and I agree that there are “multiple potential sources of bias that can stem from disruptions in the court system”. But I have read this part several times and am still very confused. Is the idea that many more whites had their cases dropped than black defendants, who were more likely to take plea deals? That would take more than a month to be reflected in the upswing in the black share of the incarcerated prison population, no? And it would have to be quite large to affect the racial composition of the prison in the months afterwards. But more troubling as I read it is that Table A.4. suggests that pre-trial guilty pleas as a share of total defendants filed did not increase after March 2020; indeed, quite the opposite as I’m reading the table: guilty pleas immediately drop from 10,277 in Feb 2020 to 7,412 in March 2020 and drop further in April to 2,344. It also drops if we calculate it as a share of total defendants filed. Case dismissals pre-trial also fall sharply, so sharply that the racial bias would have to be absolutely huge to explain any shift in the racial composition of the prison.

Reviewer 4’s detailed breakdown of the plea data is illuminating, and we fortunately have newly FOIA-ed data (and an overall stronger set of arguments) that allow us to clarify this point in the text and our analyses. Reviewer 2 brought up a similar point, and we will restate some of our comments to them below.

First, we received an incredibly rich dataset from Florida via public records request. As alluded to in our manuscript (but not directly shown), there was a spike in the proportion of dismissed cases of white defendants. We know this now because of data that we had requested from the same office that provided the Florida trial statistics data for Figure 3, and we think these new data directly address the comments above. The Office of State Court Administrator (OSCA) in Florida provided us with data from the Criminal Transaction System for 2018, 2019, 2020, and 2021; these data contain a column for defendants’ race, the action taken by the court, and the date each case was decided (among many other variables). [Note: OSCA stressed to us that they do not create the dataset, they are just the stewards of the dataset; clerks of the court record each defendant’s case data via the Offender Based Transaction System. OSCA is the organization that compiled the data and are not the custodians of the data. They are also requiring us to report a disclaimer that “Any conclusions or analysis that will derive from this dataset are solely those of the individual author(s) or the person(s) who did the analysis and not of OSCA.”]

Disclaimer aside, this dataset allows us to run a simple analysis: Among defendants with cases that were dismissed, between 2018-2021, what percent are recorded as white? We plot this in subplot (B) below—subplot (A) is Figure 3C reproduced; subplot (C) shows the two curves atop one another, rescaled using min-max scaling in order to highlight the timing and relative increase that both measures show after March 2020.

(trial data from trialstats.flcourts.org; defendant records from OSCA)

This brings me to what I see as the strongest evidence in the paper regarding mechanisms: release policies that favored white incarcerated people. The idea is that states took measures to de-densify prisons given the public health crisis in prisons and these measures advantaged white prisoners given how they reflected and heightened structural biases. For example, in many states to be released or even considered for release, one has to be nearing parole release, have a lesser offense, have good time credits, have low recidivism risk, and show evidence of an address one can go to (a “viable home plan”) – all features that would disadvantage black populations. The

release criteria example from Virginia on page 14 was incredibly illuminating. The authors own evidence in the Arkansas case study support this claim; they find that nearly three-quarters of those eligible for release were white, much greater than would be commensurate with their share of the prison population in that state (57%). I also appreciated their discussion of state variation and analysis in several other states (p. 10), showing that some states like Massachusetts showed the possibility of decarcerating “in a way that does not exacerbate existing racial inequalities.” They provide areas for other scholars and future research to examine further.

This comment is actually a great summary of many core changes that are in this revised version of the manuscript. First off, given the additional analyses we’ve conducted since receiving these reviews, we both *agree and disagree* with the points raised above. For one, we now know that it is sufficient to produce the spike in percent Black incarcerated population with only two ingredients: sudden drop in admissions and a baseline difference in length of prison sentences, by race. In that sense, the *decarceration* side of the story is featured less prominently, giving way instead for a story about disruption in admissions. And while we suspect that there was likely bias in pandemic-related decarceration (i.e., in the Arkansas case study), we doubt that the magnitude of these releases were large enough to drive the main effect we observe. However, if we consider this point more carefully, we surely see that disparities in sentencing *are* questions of decarceration. This subtle point is actually quite useful for the narrative of this work, and we have used it to rethink the structure of our Discussion section. Thank you for this critical insight.

1. The paper doesn’t make much of the fact that the black share of incarcerated people does not rise and keep rising; in fact, the data shown reveal that it rises for several months in 2020 from just under 38% to just over 38.5% and then turns back down to its original level by the first months of 2021. The way the paper is written focuses entirely on the rise and not the hill-like pattern depicted in almost all of the figures. Every state except Arizona in Figure 2 depicts this inverted U shaped pattern. (The authors acknowledge this only briefly on page 5 where they say “this spike is temporary, eventually returning to its pre-pandemic level”). The rise was actually quite short-lived – and that is not to discount its importance. But it means that whatever caused the shift in the black share of the incarcerated population was not an enduring feature or that the pandemic caused a lasting change in the demographic makeup of the prison. This point is important because it shifts how we understand the dramatic (but not durable) change in the racial composition of the prison. It may give additional clues to the importance of potential mechanisms. The timing of the “turnaround” in relative rises in black share of the imprisoned could help decipher causal mechanisms. Can the authors do more to discuss and interpret this?

This point was brought up a few times in the reviews, and we think we’re closer to getting a complete understanding, through new data and interpretation. First, to reference our simulation

of the Texas prison population, we see that once admissions begin to approach typical monthly levels, the sentence-disparity-based spike in the percent Black population begins to return to typical values; the timing and duration of the second half of the inverted-U shape, then, is based on the extent to which the race distribution in prison admissions *also* returns to typical (pre-pandemic) levels.

Note, we are highlighting admissions as especially important for the inverted-U shape, as opposed to releases, for two reasons. For one, we again find ourselves in a data-sparse setting and simply have more reliable data on admissions. Second, in the data we do have about releases, we do not see a particularly dramatic relationship between the timing of releases and any sort of inverted-U shape. In the newly added Figure A.5 (based on a public records request that was recently fulfilled, plotted below), we plot the total incarcerated population in the Federal Bureau of Prisons (FBOP) over the past 5 years (A), the number of monthly releases (B), and the percent Black population (C). If some sort of demographic differences in the FBOP releases were the key factor influencing the inverted-U shape, we would expect to see the signature of that in the releases data, which, based on this dataset, we do not.

2. What is the broader story for how we should understand how a crisis with deeply structural dimensions intersects with another site of structural violence?

This question is perhaps the one most fundamental to not only this study, but to research that we are actively building out based on this study. While the current manuscript isn't charged with solving this problem per se, we do hope to responsibly articulate a pattern that can fuel more detailed inquiries into these structural causes. The pandemic and its (ongoing) aftermath have highlighted so much structural inequality across society, from disparities in healthcare and health outcomes, to widening divides in our economic, justice, and democratic systems. While the main thrust of our manuscript continues to be focused on identifying and explaining our core results, we have added a few additional references to other scholars' work, which touch on COVID-19's especially pernicious impacts in prisons across the United States. One example we've been discussing internally for a while now is to compare today's COVID-19 outcomes to historical

health reports from prisons during / after the flu pandemic starting in 1918 in the United States (see pg 49 of the pdf “Annual Report of the New Jersey State Prison Trenton N.J. Fiscal Year 1919” which can be found at <https://dspace.njstatelib.org/xmlui/handle/10929/49629>. Note especially the passage about “Owing to rigid enforcement of the quarantine regulations during the influenza epidemic only one case of the disease developed at the Prison, and the patient recovered.”)

From a recent report (which we now cite): the death rate from COVID-19 among people who are incarcerated is on average 2.47 times that of the general population. It remains urgent and ongoing work to understand how our modern instantiation of carceral and structural violence can produce such horrific disparities.

3. The traffic stops data are very interesting but theoretically, readers will have difficulty understanding why traffic safety stops would have much to do with incarceration rates (most of these, as the authors acknowledge, do not lead to confinement or even arrest!).

We reflected deeply on this, since it’s been brought up in almost every review, and we made several changes to the writing and analyses in an attempt to clarify. We repurpose a version of our reply to Reviewers 1 and 2 below:

“...we have incorporated a very similar dataset to the Texas Highway Patrol data, this time from California. We added a description of this dataset in SI Section A.3.4, briefly introducing how the California Racial and Identity Profiling Advisory Board (RIPA) requires each state and local agency to report traffic stop incident data to the attorney general, who then posts the data in bulk. Put simply, we find the same effect as in Texas—if anything, California shows a much more pronounced spike.”

IMAGE REDACTED

“Additionally, we downloaded data about traffic stops from the New Jersey Office of the Attorney General (<https://www.njoag.gov/trafficstops/>), which is similar to the Texas and California data, but is 1) aggregated to the monthly level and 2) includes additional information about what happened *after* the traffic stop. Because of this, we can subset the data to only include traffic stops *that result in summons*. Below, we plot data about the racial breakdown of traffic stops that resulted in summons for the drivers, highlighting once again that there is a sudden and sharp increase in Black and Latino drivers’ share of the police interactions, this time extending that finding to include legal summons.”

IMAGE REDACTED

“As for how to incorporate this specific finding into the paper, we are of two minds about it: On the one hand, this result is useful when justifying why one might associate changes in the racial distribution of who interacts with police with changes to the racial makeup of prison populations, however slight this association may be. On the other hand—as we discuss in the following comment—focusing too much of our attention on the New Jersey example risks losing the broader message of why we use traffic stops data in the first place. Namely, it is a clear example of broader, systemic forces that influence policing (and who is “eligible” to be policed) in the United States: To put it bluntly, you’re not getting pulled over when you’re working from home.”

4. At several points, the authors say bias (which readers could interpret as intentional malevolent racism) but I think what their data are really pointing to are what they say in the conclusion – “the structural nature of racial disparities in the United States carceral system.” One that releases people based on their housing access; one that continues to reflect extreme levels of racialized incarceration rates (on the order of 6:1) due to policies and practices that targeted black people and communities; one that meant that more black people were on the road and potentially being stopped by police because they had to work and carry on in a pandemic where some could isolate and work remotely. I suggest not leaving this to the conclusion.

This is a terrific point. We’ve taken this advice and have both clarified our use of “*bias*” as a descriptor as well as moved discussions about systemic forces (e.g. housing, employment, etc.) earlier in the document.

5. Table A.2 – in future work, the authors might explore why some states had larger reductions in their prison populations. There is a lot of variation to explore here, ranging from a drop in New Jersey that was 65.7% of pre-pandemic levels to Wyoming which only dropped to 93.1% of the pre-pandemic level.

This is a great idea, and is indeed the subject of ongoing future work. In a sense, there's two main shapes in Figure A1: states with prison populations that are—even in 2022—still declining, and states with prison populations that sharply declined then shot right back up. Here, again, we think the primary driver is prison admissions, but that is not the whole story because in states that have completely resumed normal court activity, we do not always see the same reversion.

6. Figure 3D is also confusing – the authors should be clear about what it is depicting – does it represent court dismissals at the county level?

We've changed that figure / discussion in three ways: First, we added an entire new year of data from June 2020 to June 2021 (which strengthens our original point / analyses). Second, we have highlighted more about the source / content of the data in the caption (state-wide data from the criminal circuit defendants). Third, in the main text and Supplemental Information, we now refer to the newly acquired data from Florida, which adds great depth to the correlation shown in Figure 3D.

7. How does discretion play a role in releasing practices? I was surprised that the authors don't really consider this, only the structural features of release policies that prioritize certain kinds of offense, viable homes upon release, etc.

This is an important question, but one that is difficult to answer due to sparse data. It requires us to design new studies that collect the appropriate data.

Though we haven't addressed this in our study, we can offer a few thoughts. On the one hand, it may seem obvious why most states have adopted an *algorithmic* approach to releases—removing agency, bias, and discretion from judges, all in the name of efficiency and (maybe superficially) fairness. And given the vast literature on the relationship between algorithmic (un)fairness, justice, and mass incarceration, we are trying to sort out just how to pose this question in a data-centric manner. We look forward to thinking through this issue more in future work. attempting to address this in ongoing followup work.

8. In Figure A.7, the release patterns by race in Arkansas seem to get better after March 2020, which goes against the main argument. What is going on here? And “incarcerated

Black people are released at lower than expected rates” as far back as 2018. The authors should acknowledge and explain that this is a long-standing pattern.

The authors appreciate that Reviewer 4 caught this. We added more explanation to the figure caption and more exposition in its discussion in the main text, highlighting that, indeed, this is a long-standing problem. (But also that a deeper understanding of this pattern is crucial for influencing successful or unsuccessful policies in the future.) While it is important here to revisit our argument about admissions + sentencing disparities being the most likely contributing factors, we do acknowledge that further work is needed to look at this case in particular and whether it can potentially offer any more mechanistic insights in general.

Overall, a terrific and compelling paper and one that should be published and will make a significant contribution to our knowledge base.

We cannot express our gratitude at the depth of Reviewer 4’s comments. As a general point that is indirectly related to these comments: in the revised manuscript’s conclusion, we have now included a holistic analogy about how the pandemic caused a rupture in the system that could have affected the incarcerated populations in several ways, analogizing the pandemic to a “stress test” from engineering:

“Taken together, our findings reveal that the pandemic provided a “stress test” for the criminal legal system. In engineering, stress tests involve exposing a system to extreme conditions in order to reveal its fragilities. Using a range of data sources, we have argued that COVID-19 amplified underlying racial disparities in the carceral state.”

We concur with, and have tried to communicate more clearly in our revision: that our results demonstrate that the carceral state is a complex system whereby stressors like COVID-19, create a break on some aspects, and a motor for others, all of which interact towards the findings that we report.

The reason we point this out is that, rather than suggest that any one mechanism is the driver of the outcome (our previous submission implicated closures as the main cause), we close with an analogy that emphasizes the global effects of the pandemic in revealing structural violence.

We’re very grateful to reviewer #4 for taking the time to read the manuscript so carefully, for their excellent interpretation, and for providing many useful suggestions. We appreciate the kind words, and hope that our revision deems our study worthy of sharing with the world.

Reviewer Reports on the First Revision:

Referees' comments:

Referee #1 (Remarks to the Author):

I appreciate the efforts the authors of this study have taken to address the comments I raised in my initial report. However, I feel like some of the issues still have not received sufficient attention. I don't want to just reiterate my prior points here, so I will be sure to engage with the authors' lengthy response to my comments (which I also appreciated).

1. The reversion. I know the authors added a few sentences to the paper about the reversion, but I think it still needs much more analysis. If the point of the paper is that Covid exacerbated underlying inequities in the criminal legal system, then the fact that Covid's effect appears to have vaporated in 2021, while the legal system and life in general was still very much impacted by the virus, is an equally important issue to address.

The authors briefly argue that the reversion is further evidence that their leading causal candidate—court slowdowns and shutdowns—is the right one. But (1) given the heterogeneity in state-level outcomes, I'm not sure it's possible to draw any strong conclusions from the national trend given in Fig. 1B, and (2) the heterogeneity in state outcomes actually allows the authors to test this claim more rigorously. Start with (1). I appreciate the inclusion of Fig. A2, but I also think it now raises even more questions about what, exactly, happened from the outset of Covid to (roughly) the end of 2021. I was able to count only 15 states (and the BOP) whose state-level pattern easily fit that of the aggregate national data in Fig. 1B: CA, CO, GA, IL, ID, KS, KY, NJ (though it is flat thru 2020), NY (flat through 2020 as well), ND, PA TN, TX, WA, WV. That could go as high as 19 if you count the four states that were flat prior to 2020, saw a short spike in early 2020, and then dropped to new lows in 2021: AR, LA, MS, and WI. But that leaves over half the states with different patterns. Some saw the %-Black or %-non-white grow pretty steadily from 2018 to 2021 (AK, AZ, HI, ME, OK, RI, UT). Some saw the %-B/NW grow until 2020 then revert (MO, OR) or from 2018-2021 and then drop (CT, DE, MD, MN, MT, and somewhat VT, though VT then jumps up at the end). SD and WY seem to follow an inverted pattern, VA declines until 2021 and then spikes, and so on.

Which is to say: there is no single national story here. It's true that many of the large-prison population jurisdictions follow the pattern in Fig 1B (CA, TX, and the BOP in particular—but not really FL), but that could be because they have an outsized impact on Fig. 1B.

This, then, leads directly to point (2): there is a lot of variation here the authors could try to exploit to see if their hypotheses are correct, without having to rely on (possibly-idiosyncratic) case-studies. For example, the authors note that while lockdowns and court shutdowns went into effect at roughly the same time, the reopenings were far more heterogenous in timing and scope. Does the variation in reversions track any of these variations in reopenings? If the authors are correct that the spike is timed to court shutdowns, then openings should matter too. Similarly, were the shutdowns different in states that don't see a spike? Were the shutdowns the same, but other relevant factors different?

There are many ways to approach this. But the variation in Fig. A2 suggests that the authors can try to analyze the impact of court shutdowns/slowdowns more rigorously than looking at a single case-study (taken from a state that doesn't exactly fit Fig. 1B: FL has a reversion, but a small one, and one that quickly reverses after just four months). It also allows the authors to see if there are other important factors that might have moderated or aggravated the impact of the shutdowns.

Also, one style issue I had with Fig. A2: given the different scales on the y-axes, it is hard to really compare state experiences well: they all look roughly the same, but in some cases the spike is, say, roughly +1.2 percentage points (WI) or +0.6 points (NM). Perhaps a graph of months absolute change in percentage points would allow each state to be on the same axes. (It's obviously impossible to use a unified y-axis for %-B/NW, since the ranges vary too much—I get why the authors didn't do that here.)

2. Perhaps somewhat relatedly, how relevant is the shift from Black to Non-White? I appreciate why this happens—criminal legal race data is exhausting to work with—but Fig. A4 made me fear that the choice could have real impacts. My takeaway from Fig. A4 is that the onset of Covid led to a relative decline in the percent of non-white/non-Black people in prison too! The white share drops by ~0.3 percentage points, and the Black share rises by ~0.6 percentage points (I'm eyeballing peaks-and-troughs). Since racial shares have to add to 100%, this means the people not represented here (who make up ~20% of all people in prison) must have seen a ~0.3 percentage point decline as well—the same, roughly, as white people experienced.

From a methodological perspective, this makes the blurring of Black and non-White concerning, because it certainly looks like Black and non-Black non-white people had opposite experiences! What does aggregating states with %-Black data with states providing only %-NW data mean for the results in, say, Fig 1B? It can't just be that the %-NW states have such small Black prison populations that we can sort of "ignore" them, since among the %-NW states are IL, IN, MI, and OH, which are states likely to have large Black populations in prisons.

It's also a really intriguing causal question. What was it about Covid that led to Hispanics (white Hispanics, I'd assume—though in states with complete race data, how are those who identify as Black and Hispanic classified?) getting treated almost identically to white people? And is that even what happened, or is this some sort of artifact of how states varied in their reporting of race data? Is it the impact of just a few states (I noticed that percent-Latino for TX falls, according to the figure on p. 10 of the reply letter—I bet TX has one of the largest Latino prison populations)?

(Relatedly, that TX is the one state that openly admits to not using race self-identification is somewhat concerning. The authors frame it as "only one state" on p. 36, but when that one state has one of the largest prison populations overall and surely the largest Latino population, it's not really "only one state," it's a critical data-point that admits to a serious potential source of significant bias.)

3. Jail holds. The authors' reply to my concern about jail holds—that if those were a serious issue, we should not expect to see a 25% drop in jail populations (see their reply letter on p. 8-9)—is,

unfortunately, not persuasive, and I think it is a critical issue they still need to address. It is likely they cannot address it empirically, since the data simply are not there. But they still need to wrestle with the issue much more directly, especially for a journal like Nature, where many of the readers may not be familiar with many of the more in-the-weeds policy issues involving the criminal legal system.

My basic concern with their reply is that jail populations are likely driven in large part by the “churn” of low-level misdemeanors coming and going, and the entire misdemeanor system ground to a halt in 2020: not just on the court side, but on the street-level enforcement side as well (which, to be clear, their focus on state-trooper traffic stops—which are not just limited to traffic stops, but traffic stops on highways or in small towns lacking their own police forces—cannot observe). To put the scale of misdemeanor enforcement in perspective, misdemeanor courts process about ~13 million cases a year, vs. ~1.5 million in felony courts—but only (often fairly serious) felony cases make it to prison. The near-total shutdown of misdemeanor enforcement means that local jails could simultaneously see relatively large increases in jail holds for felony convictions alongside really substantial declines in overall jail populations.

In other words, I think that the authors’ parenthetical “e.g.” on p. 9 about how they aren’t ruling out that jail drops might have been bigger still but for jail holds is not really a parenthetical aside: I think it could actually be the real story.

And it’s not immediately clear to me how Fig. A16 E-H address this; if anything, I think it may be evidence of the importance of jail holds. I think what those show is that, given the holds, prisons only admitted the most serious cases within each category. That’s an interesting point to make (esp that it is true across all four categories, not just the highest—I’d expect jail managers would work hard to move out those convicted of the most serious violent crimes, but looks like the effect is across all levels), but I don’t see how that shows that jail holds didn’t matter. In fact, it suggests that the types of people coming from jails changed, that some (non-random) set of people were remaining in jails. Of course, some of this reflects changes in who the police were arresting during this time, but I’m not sure it explains all of it; I doubt that it does.

Now, it is true that we have little to no data on jail populations that is directly on point for what the authors want to say here. Which is really disappointing. But that doesn’t change the fact that it poses a serious challenge to their causal claim that court slowdowns drove the change in prison populations, since courts have no control over how a person’s confinement is handled post-conviction. The courts have some (sometimes quite a lot of) say over whether they are detained at all, but not where.

I don’t have a good solution, empirically, to this. I think it may reflect a critical empirical black hole that arises from a combination of poor county-level jail data and the macro-level upheaval of Covid. But it still deserves a much more extensive discussion.

4. Traffic stop data. I appreciate that the authors looked to other agencies to confirm what they saw with Texas. But even the NJ data focused on summonses doesn’t really address my core concern here, which is that traffic stops almost never result in the sorts of charges that lead to prison time

(the most likely prison-eligible traffic stop is the one that finds sufficient quantities of drugs, but drug offenses comprise only ~15% of prison populations, making it unlikely that any sort of marginal change in drug-producing traffic stops would do much in 2020). Even summonses don't really produce prison time. They may lead to more jail time, but as the authors note, that is a separate issue.

There are ways to tackle the racial impact of policing during Covid, but I don't think that traffic stop data isn't the most promising path forward. See, for example, this working paper by Massenkoff and Chalfin (http://maximmassenkoff.com/papers/victimization_rate.pdf), which looks at how victimization risk changed over the pandemic. One could invert their approach for enforcement, too: how did the risk of non-traffic stops change during the pandemic, by looking at city-level data on race and nature of arrests and the population that was at risk of such stops. (The NYPD, at least, provides detailed annual data on stops and arrests; I'm sure other departments do too.)

To be clear. I think the causal mechanism they are suggesting makes absolute sense! But I don't think traffic stop data is a good way to examine the sorts of arrests that lead to prison time. And since prisons are the focus of this paper, the case-studies need to be more closely tied to that issue. And, again, I think this is especially true for a journal like Nature; if this were for Criminology, the readers would all be able to immediately appreciate the limitations of traffic stop data for prison growth models. But the authors argue for publication in Nature to try to reach out to a wider array of researchers—likely ones who are less familiar with the complicated jurisdictional fracturing in the criminal legal system, and thus less likely to appreciate the fundamental limitations of traffic stop data.

5. Problems with the NCRP data. I had some concerns with the authors' use of the NCRP, because some of their numbers seem off. For example, in Fig A9, the NCRP file I have shows none of the errors they get for any of the five states. For all five, my NCRP dataset tracks the Klein numbers closely. In fact, all five states are viewed by BJS and Abt Associates (who manage the NCRP) as being "term file" states (the more reliable form of data) since before 2010, and my NCRP data comports with that, not the finding in Fig A9.

Which makes me concerned about Figs A7 and A8. It's not clear to me why they use so few states, when the BJS has 30 states with "term file" data starting at least by 2010, often earlier; it has 33 if you count the three states that have good data to 2018. These states are AL, AZ (to 2018), CA, CO, DE, FL, GA, IL, IN, IA, KY, MA, MI (to 2018), MN, MS, MO, MT, NE, NV, NJ (to 2018), NY, NC, OH, OK, PA, RI, SC, TN, TX, UT, WA, WI, WY. Note the authors also include ME, which the BJS says is reliable as of 2012 (when it starts in the paper).

It's unclear to me why the authors drop many of these, using only CA, CO, FL, GA, IL, KY, ME, MN, NY, OH, SC, along with ND and OR. Given the clear errors in Fig. A9, I'm concerned the reason may be a problem with the data file or the code, not the underlying data itself. I'd also note that Figs A7 and A8 include two states that the BJS does not think has good data during this period: North Dakota (which they say has a break in the data in 2014-2015 making comparisons hard, and had no data in 2016), and Oregon (which submitted no data in 2015 and 2016).

I realize that the NCRP data is a robustness check here, but I still wanted to flag some obvious errors for them to address.

Referee #2 (Remarks to the Author):

I thank the authors for seriously engaging with and addressing my comments. More generally, I found their responses to the reviewer comments very clarifying. However, I think the authors can do more to incorporate the insights from these responses in the paper itself.

I have three comments on the authors' responses to my prior comments.

1. I appreciate the authors thoroughly investigating the hypothesis that racial differences in average sentence length can go a long way in "explaining" the findings. I think the message of the paper would be much more clear if this point were a larger part of the paper's narrative. The bulk of the story seems to be that prison admissions decreased substantially during the pandemic (perhaps more so for minor offenses) and that admissions are less racially skewed than the prison population. This point isn't discussed directly in the text until page 11. I think it should come up as early as the abstract or introduction.

2. By the same token, can the authors provide statistics on the relative importance of changes in admissions versus changes in releases for the change in the prison population? It seems they can do this in Texas, at least. It would be great to see statistics on admissions and releases in other states, if possible.

3. I appreciate the authors' attempt to expand the discussion of traffic stops and tie that evidence to court outcomes. However, I still think the paper would be stronger if it dropped the discussion of traffic stops. It is just very unclear what implications these patterns have for prison statistics, if any. The California and New Jersey data, while nice additions, do not make this connection clear. Even among stops that lead to summons, very few will lead to any confinement sentence, and even fewer will lead to a prison sentence. I understand the point the authors are trying to make—that the pandemic affected the composition of who is interacting with police—but the interactions that the authors are measuring are not really the relevant ones for thinking about prison spells.

In summary, I am suggesting a paper that:

- a. Documents changes in the racial composition of the prison population across all states
- b. Points out how slowing down admissions (and potentially speeding up releases) would be expected to change the racial composition of the prison population given racial differences in sentence length
- c. Gives some sense for the relative importance of changes in admissions versus releases
- d. Uses the Florida and Arkansas case studies for a more granular discussion of changes in admission and release policy

Referee #3 (Remarks to the Author):

The authors have sufficiently responded to my comments on the original manuscript. I recommend that Nature publish the paper, as I think it makes important empirical and analytical contributions. The article should appeal to specialist and non-specialist readers alike.

Referee #4 (Remarks to the Author):

Editorial Note: Reviewer 4 was unable to review the revision but based on editorial discussion and advice from the other referees, we felt that R4's concerns were addressed sufficiently to continue considering the paper for publication.

Authors response to the second round of review:

Reviewer 1 -- Nature 2022-01-00306

I appreciate the efforts the authors of this study have taken to address the comments I raised in my initial report. However, I feel like some of the issues still have not received sufficient attention. I don't want to just reiterate my prior points here, so I will be sure to engage with the authors' lengthy response to my comments (which I also appreciated).

1. The reversion. I know the authors added a few sentences to the paper about the reversion, but I think it still needs much more analysis. If the point of the paper is that Covid exacerbated underlying inequities in the criminal legal system, then the fact that Covid's effect appears to have evaporated in 2021, while the legal system and life in general was still very much impacted by the virus, is an equally important issue to address.

The authors briefly argue that the reversion is further evidence that their leading causal candidate—court slowdowns and shutdowns—is the right one. But (1) given the heterogeneity in state-level outcomes, I'm not sure it's possible to draw any strong conclusions from the national trend given in Fig. 1B, and (2) the heterogeneity in state outcomes actually allows the authors to test this claim more rigorously. Start with (1). I appreciate the inclusion of Fig. A2, but I also think it now raises even more questions about what, exactly, happened from the outset of Covid to (roughly) the end of 2021. I was able to count only 15 states (and the BOP) whose state-level pattern easily fit that of the aggregate national data in Fig. 1B: CA, CO, GA, IL, ID, KS, KY, NJ (though it is flat thru 2020), NY (flat through 2020 as well), ND, PA TN, TX, WA, WV. That could go as high as 19 if you count the four states that were flat prior to 2020, saw a short spike in early 2020, and then dropped to new lows in 2021: AR, LA, MS, and WI. But that leaves over half the states with different patterns. Some saw the %-Black or %-non-white grow pretty steadily from 2018 to 2021 (AK, AZ, HI, ME, OK, RI, UT). Some saw the %-B/NW grow until 2020 then revert (MO, OR) or from 2018-2021 and then drop (CT, DE, MD, MN, MT, and somewhat VT, though VT then jumps up at the end). SD and WY seem to follow an inverted pattern, VA declines until 2021 and then spikes, and so on.

Which is to say: there is no single national story here. It's true that many of the large-prison population jurisdictions follow the pattern in Fig 1B (CA, TX, and the BOP in particular—but not really FL), but that could be because they have an outsized impact on Fig. 1B.

A brief note on this point, before our larger discussion below. Here, we attach a figure that excludes California, Texas, and the Bureau of Prisons:

In addition to a version that excludes Florida as well:

In fact, we can exclude we exclude over 50% of the U.S. population (by also excluding Georgia, Ohio, New York, and Pennsylvania) in order to generate this figure:

Ultimately, by excluding over 90% of the total U.S. prison population, we create this plot:

Earnestly, we do not want to use these to try and push back on the valid points Reviewer 1 makes above, we wanted just to add them to contextualize our longer response below.

This, then, leads directly to point (2): there is a lot of variation here the authors could try to exploit to see if their hypotheses are correct, without having to rely on (possibly-idiosyncratic) case-studies. For example, the authors note that while lockdowns and court shutdowns went into effect at roughly the same time, the reopenings were far more heterogenous in timing and scope. Does the variation in reversions track any of these variations in reopenings? If the authors are correct that the spike is timed to court shutdowns, then openings should matter too. Similarly, were the shutdowns different in states that don't see a spike? Were the shutdowns the same, but other relevant factors different?

There are many ways to approach this. But the variation in Fig. A2 suggests that the authors can try to analyze the impact of court shutdowns/slowdowns more rigorously than looking at a single case-study (taken from a state that doesn't exactly fit Fig. 1B: FL has a reversion, but a small one, and one that quickly reverses after just four months). It also allows the authors to see if there are other important factors that might have moderated or aggravated the impact of the shutdowns.

Also, one style issue I had with Fig. A2: given the different scales on the y-axes, it is hard to really compare state experiences well: they all look roughly the same, but in some cases the spike is, say, roughly +1.2 percentage points (WI) or +0.6 points (NM). Perhaps a graph of months absolute change in percentage points would allow each state to be on the same axes. (It's obviously impossible to use a unified y-axis for %-B/NW, since the ranges vary too much—I get why the authors didn't do that here.)

This is a good point, which we can begin to address using newly collected admissions data at the monthly level, but zooming out, we have reframed a lot of the core arguments in the paper away from the question of whether there’s one “national story” / many independent stories / several regionally-clustered stories, etc. We discuss this much more substantially in the Supplemental Information, and in the end, it may not be the case that the data shows one single, unified story across every state (we continue to think there is). The new framing and emphasis in the main text gets around this issue slightly.

There may not be one nationwide trend that each state follows; however, there is a *nationwide mechanism* underlying the dynamics of prison population demographics. The mechanism is simple, and it leads to the patterns seen in Fig. 1B and Fig. A2. If there are 1) differences in the average time spent in prison based on race and 2) a sudden decline in prison admissions, then we will see some variation of the trend in Fig. 1B. *In addition* to that basic mechanism, one can imagine factors that can exacerbate and/or attenuate the size and timing of the spike. These include: New or atypical patterns in prison admissions by race (relative to averages prior to the decline in admissions), or new or atypical changes in prison releases by race. Whether this mechanism manifests as the dramatic spikes from Fig. 1B depends on how these factors interact. The importance of reframing the question in terms of mechanisms rather than whether or not 100% of states reproduce the trend in Fig. 1B is useful because it allows for added nuance—see, for example, the new comparison of median sentence length distributions in Illinois and Texas, which we have added to the Supplementary Information.

This figure is created by 1) randomly sampling 100 White/Black/Latino incarcerated people from the bulk population datasets released frequently by each state (the High Value Data Sets in Texas and Prison Population Datasets in Illinois). 2) We then record the median length of sentence from each of these three groups of 100 randomly selected people and 3) repeat this sampling procedure 1000 times. From this, we create a histogram showing the distribution of median sentence lengths for White, Black, and Latino people across each state. (A) and (B) show this histogram for both states. We see

higher median sentence lengths for Black incarcerated people than White in both states; in Illinois, we see a high overlap between the distributions of median sentence lengths for Black and Latino incarcerated people, whereas in Texas, White and Latino incarcerated people have similar median sentence length distributions. According to our point about there being a nationwide *mechanism* (as opposed to trend, per se), these baseline differences in sentence length distributions in the two states would give us clear predictions about what would happen come March 2020. That is, based on differences in sentencing *alone*, we would expect to see spikes in the curves for %Black and %Latino incarcerated population in Illinois (C & D below) but not in Texas (E & F below).

While we currently include this figure in the Supplemental Information, we note that this (current) person-level data on incarcerated populations over time is rarely reported across states, and as such, it does not feature heavily in the main text. However, the figure itself is simply an additional justification of our framing of the national mechanism.

Now, aside from the “national story” vs “national mechanism” question, there’s still quite a bit to say about the presence or absence of a single national story. For instance, we attached the figures above that exclude more and more states’ data, all the while broadly reproducing the original trends from Fig 1B. The trouble with this approach, however, is that we are inevitably left studying only a fraction of all people in prison in order to highlight a few states that do not follow precisely the same trend as Fig. 1B. Another way of phrasing that is to put this in terms of probability: Randomly sample someone in prison during 2020: What is the probability that they are incarcerated in a state that is undergoing the trend from Fig. 1B? Here, again, this points to a national story.

Another way of posing this question is to highlight the following figure again:

Note at the top of the figure, there are sloped lines based on the data from the different colored time windows (2017-2019, 2019-2020, 2020-2021, 2021-2022). Isolating these slopes is useful again for nationwide comparison for two reasons: 1) because it can allow us to ask questions about what we would expect to happen if we did not observe some sort of spike in the %Black population in 2020-2021, and 2) because we can use the slope of 2019-2020 as a way of normalizing the entire time series, allowing for a simple test of whether or not we observe a spike in 2020.

Below, each violin plot represents every state and federal prison system's slope of the percent incarcerated Black population curves. From 2020 to 2021, we observe a statistically significant difference in the slope from before the pandemic.

This plot shows that nationally—with few exceptions, which we discuss below—the slope of the %Black incarcerated curves increased during 2020, compared to the 2019 averages. We define this as an increase relative to the slope of the pre-pandemic curve, sustained for at least five out of the first six months after March 2020. After this, the same slopes decreased during 2021.

As we mention above, the other benefit of highlighting the slopes of the pre-pandemic % Black population curves is as a normalization/standardization tool. In the figure below, we divide each state’s time series of % Black population by its corresponding value in the best-fit line from the slope from the year prior to March 2020. Visually, this bundles the states’ curves to the 100% value before the pandemic (100% of normalized pre-pandemic values). After March 2020, there are five states that do not at least briefly show a spike in their percent Black population: Maine, Maryland, Missouri, Oregon, and Wyoming.

These five states maintain prison systems that incarcerate, on average, fewer people under shorter (less than 2-year) sentences, according to NCRP (differences explored in snapshot Figures below). This observation has two subtle but important consequences: First, it suggests that a key reason why the disparities emerge is due to releases of incarcerated people who served shorter-term sentences (without a corresponding amount of admissions). This makes sense, because on any given day, a randomly-selected person being released from prison is likely to have been sentenced for a shorter time period. Second, if white people are more likely to serve shorter sentences, then an overall

reduction in the amount of people serving shorter-term prison sentences means there are fewer people serving shorter-term sentences who could be “eligible” to drive the main effect in Fig1B.

Figure above: comparison of sentence length distributions by race in the five states that do not follow the main trends in Fig 1B. Note especially the relatively small number of people in these states serving sentences that are less than 2 years in length. Additionally, these states differ from the more general pattern in the United States, which typically show more monotonic trends in their comparison in Black-White sentence distributions, as shown in the figure below. The trend of decreasing proportion of white people serving longer sentence lengths coupled with increasing proportions of Black people serving long sentences is typical of many state prison systems.

The benefit of this new focus on a simple mechanistic story (“simple”—it’s tough to say anything is simple when dealing with U.S. mass incarceration) is that if we don’t see a sudden, widespread bias in new prison admissions, then the key factor influencing the trends from Fig. 1B has to be related to who is being released and/or who is remaining in prison. This not only clarifies important dynamical properties of the U.S. prison population, but it also informs new data collection, hypotheses, and data-informed policy down the road.

- Perhaps somewhat relatedly, how relevant is the shift from Black to Non-White? I appreciate why this happens—criminal legal race data is exhausting to work with—but Fig. A4 made me fear that the choice could have real impacts. My takeaway from Fig. A4 is that the onset of Covid led to a relative decline in the percent of non-white/non-Black people in prison too! The white share drops by ~0.3 percentage points, and the Black share rises by ~0.6 percentage points (I’m eyeballing peaks-and-troughs). Since racial shares have to add to 100%, this means the people not represented here (who make up ~20% of all people in prison) must have seen a ~0.3 percentage point decline as

well—the same, roughly, as white people experienced.

From a methodological perspective, this makes the blurring of Black and non-White concerning, because it certainly looks like Black and non-Black non-white people had opposite experiences! What does aggregating states with %-Black data with states providing only %-NW data mean for the results in, say, Fig 1B? It can't just be that the %-NW states have such small Black prison populations that we can sort of “ignore” them, since among the %-NW states are IL, IN, MI, and OH, which are states likely to have large Black populations in prisons.

It's also a really intriguing causal question. What was it about Covid that led to Hispanics (white Hispanics, I'd assume—though in states with complete race data, how are those who identify as Black and Hispanic classified?) getting treated almost identically to white people? And is that even what happened, or is this some sort of artifact of how states varied in their reporting of race data? Is it the impact of just a few states (I noticed that percent-Latino for TX falls, according to the figure on p. 10 of the reply letter—I bet TX has one of the largest Latino prison populations)?

(Relatedly, that TX is the one state that openly admits to not using race self-identification is somewhat concerning. The authors frame it as “only one state” on p. 36, but when that one state has one of the largest prison populations overall and surely the largest Latino population, it's not really “only one state,” it's a critical data-point that admits to a serious potential source of significant bias.)

Addressing the last point first: We appreciate this, and we've changed our language to make it less as “only one state” and are more careful to emphasize that it's a big state, which may have a big impact on any aggregated story. While Texas indeed may report race data differently than the other surveyed states, it is especially worrying if there were reason to think that there are heterogeneities about race reporting within Texas, which would make large scale inferences about statewide trends difficult. We do not have evidence that this is the case, but we have added a sentence to clarify where this could prove to be an issue in our analyses. However, ultimately, we can remove Texas entirely and still reproduce our main effect.

In regards to the other points above, we have changed the following, which dovetails with our more lengthy discussion above: First, after applying the same standardization procedure (divide by slope of best-fit line for one year prior to March 2020), we still find a brief and abrupt spike in the percent Latino incarcerated population among the 37 states that report Hispanic/Latino as a separate race category (Panel D in the figure below, which now replaces the previous Fig. A3). Because of this, we have changed some of our

“percent of Black and Latino incarcerated people” language to either only include Black or to include Latino when relevant. Separate note: by normalizing by the average value in the year prior to the pandemic vs the *slope* of that curve, we see slightly different stories for the % Latino curves. Panel D’s % Latino picks up on the visual change in the slope of the already-increasing % Latino curve in B, which increases even more after March 2020.

We have really taken this point to heart and have changed much of our language throughout the manuscript to accommodate this important distinction.

Lastly, to revisit the comparison of sentence length distributions from the previous comment, we refer again to the illustrative comparison between the sentence length distribution for Black/White/Latino people in Illinois and Texas to highlight that the important factor, again, ends up being baseline disparities in sentence length by race.

3. Jail holds. The authors’ reply to my concern about jail holds—that if those were a serious issue, we should not expect to see a 25% drop in jail populations (see their reply letter on p. 8-9)—is, unfortunately, not persuasive, and I think it is a critical issue they still need to address. It is likely they cannot address it empirically, since the data simply are not there. But they still need to wrestle with the issue much more directly, especially for a journal like *Nature*, where many of the readers may not be familiar with many of the more in-the-weeds policy issues involving the criminal legal system.

My basic concern with their reply is that jail populations are likely driven in large part by the “churn” of low-level misdemeanants coming and going, and the entire misdemeanor system ground to a halt in 2020: not just on the court side, but on the street-level enforcement side as well (which, to be clear, their focus on state-trooper traffic stops—which are not just limited to traffic stops, but traffic stops on highways or in small towns lacking their own police forces—cannot observe). To put the scale of misdemeanor enforcement in perspective, misdemeanor courts process about ~13 million cases a year, vs. ~1.5 million in felony courts—but only (often fairly serious) felony cases make it to prison. The near-total shutdown of misdemeanor enforcement means that local jails could simultaneously see relatively large increases in jail holds for felony convictions alongside really substantial declines in overall jail populations.

In other words, I think that the authors’ parenthetical “e.g.” on p. 9 about how they aren’t

ruling out that jail drops might have been bigger still but for jail holds is not really a parenthetical aside: I think it could actually be the real story.

And it's not immediately clear to me how Fig. A16 E-H addresses this; if anything, I think it may be evidence of the importance of jail holds. I think what they show is that, given the holds, prisons only admitted the most serious cases within each category. That's an interesting point to make (esp that it is true across all four categories, not just the highest--I'd expect jail managers would work hard to move out those convicted of the most serious violent crimes, but looks like the effect is across all levels), but I don't see how that shows that jail holds didn't matter. In fact, it suggests that the types of people coming from jails changed, that some (non-random) set of people were remaining in jails. Of course, some of this reflects changes in who the police were arresting during this time, but I'm not sure it explains all of it; I doubt that it does.

Now, it is true that we have little to no data on jail populations that is directly on point for what the authors want to say here. Which is really disappointing. But that doesn't change the fact that it poses a serious challenge to their causal claim that court slowdowns drove the change in prison populations, since courts have no control over how a person's confinement is handled post-conviction. The courts have some (sometimes quite a lot of) say over whether they are detained at all, but not where.

I don't have a good solution, empirically, to this. I think it may reflect a critical empirical black hole that arises from a combination of poor county-level jail data and the macro-level upheaval of Covid. But it still deserves a much more extensive discussion.

We continue to appreciate these comments, and they've forced us to contend with a few new datasets, analyses, and discussions in order to address. As Reviewer 1 emphasizes, "...we have little to no data on jail populations that is directly on point for what the authors want to say here. Which is really disappointing." We agree, but there are two new discussion points we can add here that may be informative.

First, we think that the exercise above of isolating the states where the trend from Fig1B does not apply is useful for understanding the influence of jails. That is, these five states have prison populations serving, on average, longer sentences. This is primarily driven by relatively fewer people in prison serving sentences that are shorter than two years. This,

Second, we can look specifically at states with a combined jail / prison system as a way of asking to what extent we still see the main effects observed in Fig1B. These are: Alaska, Connecticut, Delaware, Hawaii, Rhode Island, and Vermont. Below, we plot Fig 1B subset to include only these states.

Among states with unified jail/prison systems
(AK, CT, DE, HI, RI, VT)

While this does not full address the role that county jails played/play in driving the demographic dynamics we observe, we do feel that it—especially in conjunction with the overall reframing around the crucial role that disparities in sentence length distributions play in these trends—has sufficiently motivated changes to the manuscript’s language as it relates to admissions to prison during this period. Lastly, one especially promising avenue for future work is that, for several states, we have complete data about admissions and releases, including the nature of the admission (i.e., transfers, parole, sentencing, etc.). While this would be more case-study oriented, this is precisely the type of data that could help to resolve some of the questions that arise from this work.

4. Traffic stop data. I appreciate that the authors looked to other agencies to confirm what they saw with Texas. But even the NJ data focused on summonses doesn’t really address my core concern here, which is that traffic stops almost never result in the sorts of charges that lead to prison time (the most likely prison-eligible traffic stop is the one that finds sufficient quantities of drugs, but drug offenses comprise only ~15% of prison populations, making it unlikely that any sort of marginal change in drug-producing traffic stops would do much in 2020). Even summonses don’t really produce prison time. They may lead to more jail time, but as the authors note, that is a separate issue.

There are ways to tackle the racial impact of policing during Covid, but I don’t think that traffic stop data isn’t the most promising path forward See, for example, working paper

by Massenkoff and Chalfin (http://maximmassenkoff.com/papers/victimization_rate.pdf), which looks at how victimization risk changed over the pandemic. One could invert their approach for enforcement, too: how did the risk of non-traffic stops change during the pandemic, by looking at city-level data on race and nature of arrests and the population that was at risk of such stops. (The NYPD, at least, provides detailed annual data on stops and arrests; I'm sure other departments do too.)

To be clear. I think the causal mechanism they are suggesting makes absolute sense! But I don't think traffic stop data is a good way to examine the sorts of arrests that lead to prison time. And since prisons are the focus of this paper, the case-studies need to be more closely tied to that issue. And, again, I think this is especially true for a journal like *Nature*; if this were for *Criminology*, the readers would all be able to immediately appreciate the limitations of traffic stop data for prison growth models. But the authors argue for publication in *Nature* to try to reach out to a wider array of researchers—likely ones who are less familiar with the complicated jurisdictional fracturing in the criminal legal system, and thus less likely to appreciate the fundamental limitations of traffic stop data.

This, along with Reviewer 2's recommendations on the matter, led us to remove the discussion of traffic stops almost entirely, and we appreciate being pushed on this. Like Reviewer 1, we really do think the causal mechanism here makes (at least partial) sense, and we were absolutely struck by the fact that it shows up repeatedly across every state where we've found data (e.g. now we have data from CA, TX, NJ, CT, AZ, and are getting a few others). There is "a story" here for sure, but we're now convinced that the story doesn't belong in the main text of a story largely about incarceration / decarceration.

5. Problems with the NCRP data. I had some concerns with the authors' use of the NCRP, because some of their numbers seem off. For example, in Fig A9, the NCRP file I have shows none of the errors they get for any of the five states. For all five, my NCRP dataset tracks the Klein numbers closely. In fact, all five states are viewed by BJS and ABT Associates (who manage the NCRP) as being "term file" states (the more reliable form of data) since before 2010, and my NCRP data comports with that, not the finding in Fig A9.

Which makes me concerned about Figs A7 and A8. It's not clear to me why they use so few states, when the BJS has 30 states with "term file" data starting at least by 2010, often earlier; it has 33 if you count the three states that have good data to 2018. These states are AL, AZ (to 2018), CA, CO, DE, FL, GA, IL, IN, IA, KY, MA, MI (to 2018),

MN, MS, MO, MT, NE, NV, NJ (to 2018), NY, NC, OH, OK, PA, RI, SC, TN, TX, UT, WA, WI, WY. Note the authors also include ME, which the BJS says is reliable as of 2012 (when it starts in the paper).

It's unclear to me why the authors drop many of these, using only CA, CO, FL, GA, IL, KY, ME, MN, NY, OH, SC, along with ND and OR. Given the clear errors in Fig. A9, I'm concerned the reason may be a problem with the data file or the code, not the underlying data itself. I'd also note that Figs A7 and A8 include two states that the BJS does not think has good data during this period: North Dakota (which they say has a break in the data in 2014-2015 making comparisons hard, and had no data in 2016), and Oregon (which submitted no data in 2015 and 2016).

I realize that the NCRP data is a robustness check here, but I still wanted to flag some obvious errors for them to address.

Thank you so very much for pointing this out. We re-downloaded the 2019 dataset from <https://www.icpsr.umich.edu/web/ICPSR/studies/38048/datadocumentation#> and we were able to create a national graphic similar to FigA2, attached below. As Reviewer 1 mentions, this was indeed a robustness / sanity check for the data, but it is nevertheless better to avoid obvious errors here. In the figure below, states with light gray labels (e.g. Alaska, Oregon, Hawaii, Idaho, etc.) are not included in the term file. Washington, California, Nevada, Utah, Nebraska, Arizona, Colorado, Wyoming, Kansas, Texas, Iowa, Minnesota, Illinois, Indiana, Kentucky, Tennessee, Mississippi, West Virginia, Ohio, Wisconsin, Georgia, Florida, New Jersey, New York, and South Carolina are the states with the highest degree of correspondence between the two datasets. There are still a few states where the correspondence between our data and the NCRP data (e.g. Rhode Island, Missouri, Maryland, among others). This puzzles us, because in each of these states we were provided with (raw) data directly from these states' Departments of Correction. Nevertheless, the datasets are, for the most part, the same.

Lastly, to reiterate a general point: We greatly appreciate the care that Reviewer 1 took with these comments. They have improved the paper both technically and substantively.

Additionally, in the current Supplementary Information, we have included tables of the population and demographic data for each prison system studied here. Also included in these tables is a tag about whether or not the data is raw data from the state or *interpolated*. For most (35) prison systems, we have raw monthly data for the entire duration of the study period. For some, we only have data at the quarterly (8) or bi-annually (4); for these 12 states, we simply do a linear interpolation on the raw demographic data and sum these columns together to arrive at a total estimate for the number of people incarcerated each month in between the

quarterly/biannual dates. Note: As a validation, we do this same interpolation on states where we do have reliable monthly data, and the estimates are almost perfectly aligned (average R^2 of correlation with ground truth for bi-annual interpolation: 0.975; annual interpolation: 0.945). For five states: Michigan, New Jersey, South Carolina, Tennessee, and Virginia, we only have demographic data at the yearly level. In each of these states, we have population *totals* at the monthly level; with these, the task becomes to estimate the counts of incarcerated people by race each month, given the population totals. In some ways, this is an easier task, since we know the overall trend in the prison population. Here, again, we do a linear interpolation between the dates without missing values, multiplied by a factor of (interpolated_sum / actual_sum). Doing this same validation on states with reliable, monthly data reporting gives us high alignment again. Lastly, we note that every combination of including or excluding states based on their reporting frequency and quality still produce the same qualitative results, which we would expect given the extensive discussion above.

Upon publication, we hope to have an online dashboard ready to share with researchers, the public, and policymakers to use and build upon. Since we originally submitted this work back in January, we have tried to convince a number of states to update the ways and frequency with which they report demographic data about prison populations. Some have! (California, Alaska) We feel that a tool like this for the public, which contains *the public's* data, will go a long way towards normalizing and standardizing the disparate and frustrating ways that states currently share this important data.

Reviewer 2 -- Nature 2022-01-00306

I thank the authors for seriously engaging with and addressing my comments. More generally, I found their responses to the reviewer comments very clarifying. However, I think the authors can do more to incorporate the insights from these responses in the paper itself.

Throughout, we appreciate the insights and critical comments received. Responding to them has been instructive, and we really think that the process of incorporating them has made for a much improved paper that is clearer, more focused, with potentially a more direct impact.

In terms of the comments below, we have tried to do exactly what they suggest. We address them below and point to the location of where changes have been made in response to them.

I have three comments on the authors' responses to my prior comments.

1. I appreciate the authors thoroughly investigating the hypothesis that racial differences in average sentence length can go a long way in "explaining" the findings. I think the message of the paper would be much more clear if this point were a larger part of the paper's narrative. The bulk of the story seems to be that prison admissions decreased substantially during the pandemic (perhaps more so for minor offenses) and that admissions are less racially skewed than the prison population. This point isn't discussed directly in the text until page 11. I think it should come up as early as the abstract or introduction.

In short, our most substantive changes in this revision is precisely about this point. A mix of comments like this (and similar ones echoed by Reviewer 4) have really forced us to reframe / refocus the overall narrative of the paper. That is: 1) We observe the spike in the percent of incarcerated people who are Black, starting in March 2020. 2) We consider a few mechanisms that may have influenced this, but 3) ultimately, we discuss and provide further evidence for a much simpler statistical story built around pre-existing disparities in sentence length by race. A priori, we may have *expected* to see the trends found in Fig1B, based solely on sentencing data by race (i.e., amid sentence length differences by race, the average person released on any given day is disproportionately more likely to be white).

This connects to your broader structural recommendations below about how to streamline the story, but the edits required to accommodate the re-focusing on sentence length distributions ended up being quite substantive. We removed the traffic stops case study entirely, as well as cutting much of the discussion of admissions / releases via court

closures or states' policies on decarceration during the pandemic (these are still discussed in the Supplementary Information, but feature much less prominently in the main text). In its place, we now have a devoted section where we look at the effects of differences in sentence length by race, which is accompanied by several new analyses. First, we are able to explain what was previously under-treated in our earlier drafts: The difference between the relative change in the Black incarcerated population vs. that of the Latino population. Were too quick to use language that grouped the respective changes of incarcerated Black/Latino populations, and now we have a much more clarified understanding about why they may or may not diverge, depending on the state we're looking at. An especially salient example of this is to look at the time series of %Black and %Latino populations in Texas and in Illinois:

This figure is created by 1) randomly sampling 100 White/Black/Latino incarcerated people from the bulk population datasets released frequently by each state (the High Value Data Sets in Texas and Prison Population Datasets in Illinois). 2) We then record the median length of sentence from each of these three groups of 100 randomly selected people and 3) repeat this sampling procedure 1000 times. From this, we create a histogram showing the distribution of median sentence lengths for White, Black, and Latino people across each state. (A) and (B) show this histogram for both states. We see higher median sentence lengths for Black incarcerated people than White in both states; in Illinois, we see a high overlap between the distributions of median sentence lengths for Black and Latino incarcerated people, whereas in Texas, White and Latino incarcerated people have similar median sentence length distributions. According to our point about there being a nationwide *mechanism* (as opposed to trend, per se), these baseline differences in sentence length distributions in the two states would give us clear predictions about what would happen come March 2020. That is, based on differences in sentencing *alone*, we would expect to see spikes in the curves for %Black and %Latino incarcerated population in Illinois (C & D below) but not in Texas (E & F below).

While we currently include this figure in the Supplemental Information, we note that this (current) person-level data on incarcerated populations over time is rarely reported across states, and as such, it does not feature heavily in the main text. However, the figure itself is simply an additional justification of our framing of the national mechanism.

Last point about this: having this newly-focused, clarified treatment of differences in sentence length by race allows us to do a deeper dive in terms of the states that *do not* show the main effects from Figure 1B, of which we have identified five. (Note: This ties into remarks by Reviewer 1 about what exactly we’re defining as the observed “trend” from Fig1B—is it the down-up-down pattern we observe at the national level? Or is it merely an increase relative to pre-pandemic trends? We define the main effect as an increase relative to the slope of the pre-pandemic curve, sustained for at least five out of the first six months after March 2020.) The states that did not reproduce this trend are those who have either 1) relatively smaller incarcerated populations, but more importantly 2) have prison systems that, on average, includes fewer people with short-term sentences. This is crucial, since a core factor driving the main effects we observe is not simply differences in sentence length by race, but also in particular, differences driven in large part by more white incarcerated people serving shorter term sentences. If there are fewer people in prison who have been sentenced to <2 years, that raises the overall median sentence length by white incarcerated people, thereby making the distributions of sentence length by race much closer than they are typically in other states. We include examples of this now in the Supplementary Information, and have tried to make the finale of our Results section—and the bulk of the Discussion section—reflect and summarize this core argument. Lastly, we still include potential mechanisms of admissions and releases as possible state-specific factors that may tune the overall effect, but that these are by no means a national-level trend. In fact (and we’ll discuss this more with the following comment), using data from almost 20 states’ admission and releases, we show that while certain states (e.g. Texas, in FigA.22) may show increases in the proportion of admissions who are Black, this is largely not a nationwide trend and therefore should not be given as much weight as the sentencing observation.

We really thank Reviewer 2 for pressing hard on this because it really has given us a framing that is more concise and informative, also in line with the editor’s comments to us, that she wants us to produce a manuscript that can tell the broadest story about what drives the majority of these trends across as many states as possible.

- By the same token, can the authors provide statistics on the relative importance of changes in admissions versus changes in releases for the change in the prison population? It seems they can do this in Texas, at least. It would be great to see statistics on admissions and releases in other states, if possible.

More and more it became clear that we needed a similar large-scale data collection effort for admissions and releases, in addition to just population data. For this reason, we’re now including multiple states’ admissions and releases data, by race in Figure ____ (note, we have also added a co-author, Arush Sharma, who has been instrumental in the collection and analysis of these data). What we see is broadly interesting but far less of a clear, across-the-board effect as the population / demographic changes. First, as expected, admissions across these 18 states declined to almost zero during April or May 2020.

Ultimately, based on demographic data about the admissions and releases from 18 states’

prison systems, racial disparities in admissions and releases alone are not able to drive the broad trends observed in this study. In fact, if these were the only factors influencing prison population demographics, we would expect the *opposite* effect seen in Fig1B, since we observe a large increase in the proportion of white admissions after the start of the pandemic, amid large decreases in the proportion of Black admissions and relatively commensurate rates of releases. There are examples of individual states that show sudden increases in the relative amount of Black people admitted to prison at the start of the pandemic (see Texas, for example). Similarly, there are examples of large-scale releases causing an abrupt increase in the percent of Black incarcerated people (see a recent example in January 2022 in data from the Federal Bureau of Prisons, Figure below). Nevertheless, we do not see these factors as being anywhere as influential as disparities in sentencing.

The inclusion of these releases / admissions data also suggest that a fourth broad data collection effort will be required in the coming years. To give a sense of the longer-term goals of this collaboration, we hope to collect, clean, and make publicly available all of these different data sources so that researchers and policy makers can have a more up-to-date view of the current state of prisons in the U.S. For example, much of this information is available at a 2-3 year lag often, *despite* being produced by states' DOCs at a monthly level. This gap isn't necessarily especially urgent when doing longer-term retrospective studies of carceral policy, but when there are sudden disruptions or policy changes (e.g. pandemic, supreme court decisions, widespread social protests, etc.), having access to this kind of data is quite useful.

3. I appreciate the authors' attempt to expand the discussion of traffic stops and tie that evidence to court outcomes. However, I still think the paper would be stronger if it dropped the discussion of traffic stops. It is just very unclear what implications these

patterns have for prison statistics, if any. The California and New Jersey data, while nice additions, do not make this connection clear. Even among stops that lead to summons, very few will lead to any confinement sentence, and even fewer will lead to a prison sentence. I understand the point the authors are trying to make—that the pandemic affected the composition of who is interacting with police—but the interactions that the authors are measuring are not really the relevant ones for thinking about prison spells.

As we wrote to Reviewer 1: “[These] recommendations on the matter, led us to remove the discussion of traffic stops almost entirely, and we appreciate being pushed on this. Like Reviewer 1, we really do think the causal mechanism here makes (at least partial) sense, and we were absolutely struck by the fact that it shows up repeatedly across every state where we’ve found data (e.g. now we have data from CA, TX, NJ, CT, AZ, and are getting a few others). There is “a story” here for sure, but we’re now convinced that the story doesn’t belong in the main text of a story largely about incarceration / decarceration.”

In summary, I am suggesting a paper that:

- a. Documents changes in the racial composition of the prison population across all states.
- b. Points out how slowing down admissions (and potentially speeding up releases) would be expected to change the racial composition of the prison population given racial differences in sentence length.
- c. Gives some sense for the relative importance of changes in admissions versus releases.
- d. Uses the Florida and Arkansas case studies for a more granular discussion of changes in admission and release policy.

To reiterate, we truly are grateful for these comments. They, along with those from the other reviewers, have led us to re-frame and analyze further the crucial role that baseline disparities in sentence length distributions have in state prison systems producing the trends from Fig1B. We have used the suggestions directly above this as a guide for how to consolidate and restructure the overall narrative of the manuscript, and we hope that we have done so in a way that accurately captures the suggestions. Thank you.

Reviewer 3 -- Nature 2022-01-00306

The authors have sufficiently responded to my comments on the original manuscript. I recommend that Nature publish the paper, as I think it makes important empirical and analytical contributions. The article should appeal to specialist and non-specialist readers alike.

The authors are thrilled with Reviewer 3's praise of this work. We hope that with the revisions included from Reviewers 1 and 2 that this work will have the empirical and analytical impact that Reviewer 3 mentions above.

Reviewer Reports on the Second Revision:

Referees' comments:

Referee #1 (Remarks to the Author):

I want to start by saying that I think the restructuring of the paper has made a very real difference, and I appreciate the effort that the authors have put into this version. At this point, I only have a few small comments, all of which are more about things that struck me as not entirely clear, not on the substance.

1. The most significant one is, I think, more an issue of phrasing than substance, but an issue of phrasing that I think is important. In the new part of the paper, the authors decompose prison growth into admissions, releases, and sentence length. At one level, this seems redundant, since sentence length is just the time between admission and release, and the total prison population at any point is just the sum of admissions minus the sum of releases.

Because of this, it took me a few times reading it before I really realized that what I believe what the authors are saying is that the prison population in 2020 = (A) those admitted in 2020 + (B) those who were admitted before 2020 and remain after 2020 - (C) those released in 2020, and that the spike and reversion is caused primarily by the composition of B (what they call "sentence length"). Is this right? If so, I agree with the point, but think I would call it something other than sentence length, because it isn't really that, at least not directly. It's true that B differs from A and C due to racial differences in sentence length, but given the short window for the spike and regression, it's more about the "stock" of people in prison than either of the two "flow" variables (although, in this context, obviously, the term "stock" is terrible and something else should be used, and I used it here only to emphasize the stock-vs-flow point, given how it is commonly phrased).

I realize that this could seem needling, but it really did throw me off, and made me miss the valid point that was being made.

2. When talking about the racial composition of B, and how it differs from A and C, the authors may want to note the recent study by the Council on Criminal Justice, which makes a similar point (not about Covid, but about prison pops in general), but notes that a major explanatory reason for this is differences in violent offending and victimization, which complicates the (political) story about the causes here: <https://counciloncj.foleon.com/reports/racial-disparities/national-trends>. It's obviously not the time to include some major new section on this, but it may be worth at least discussing it briefly.

3. For Fig. A.2/Section A2, I'm not entirely sure why the authors say that only Maine, Oregon, and Wyoming don't fit the general trend. I get that those three see *declines* in disparities. But I feel like, for ex, OK/CT/DE (steady rises), or MO/WI (looks like declines as well) don't fit as well. If there is some way to clarify what really makes the three states they highlight uniquely non-conforming (or what pulls OK and MO into the conforming fold) that would be great--but I also think that Fig. A.2 makes the data transparent enough that it is easy for readers to draw their own assessments about

the importance of heterogeneity, so this is more an issue of just adding in some clarifying exposition.

4. Along those lines, it could be that Fig. A.4 provides the evidence I am looking for--I have to admit that I am completely unfamiliar with violin plots and had a hard time understanding what it was saying. That could just reflect my own lack of knowledge of that sort of plot, but it also makes me somewhat concerned that other readers may struggle too. I wonder if there is a way to convey that information in a more graphically-conventional format?

5. In Fig. A.5, is it worth noting who the big outliers at the top are as well? And following up on point (3), if A.5 is what makes it clear why the non-conforming listed at the start of Section A.2 are the non-conforming, that wasn't entirely clear to me either. Also, the highlighted states in A.5 isn't the same group as those listed on p. 24 at the start of Section A.2--are these different groups?

Referee #2 (Remarks to the Author):

The authors have sufficiently responded to my comments on the previous draft. I recommend that Nature publish the paper and I thank the authors for their careful and thoughtful engagement.

Author Rebuttals to Second Revision:

Referee #1 (Remarks to the Author):

I want to start by saying that I think the restructuring of the paper has made a very real difference, and I appreciate the effort that the authors have put into this version. At this point, I only have a few small comments, all of which are more about things that struck me as not entirely clear, not on the substance.

We are so sincerely pleased about this review process. Thank you for your work in improving this paper.

1. The most significant one is, I think, more an issue of phrasing than substance, but an issue of phrasing that I think is important. In the new part of the paper, the authors decompose prison growth into admissions, releases, and sentence length. At one level, this seems redundant, since sentence length is just the time between admission and release, and the total prison population at any point is just the sum of admissions minus the sum of releases.

Because of this, it took me a few times reading it before I really realized that what I believe what the authors are saying is that the prison population in 2020 = (A) those admitted in 2020 + (B) those who were admitted before 2020 and remain after 2020 - (C) those released in 2020, and that the spike and reversion is caused primarily by the composition of B (what they call "sentence length"). Is this right? If so, I agree with the point, but think I would call it something other than sentence length, because it isn't really that, at least not directly. It's true that B differs from A and C due to racial differences in sentence length, but given the short window for the spike and regression, it's more about the "stock" of people in prison than either of the two "flow" variables (although, in this context, obviously, the term "stock" is terrible and something else should be used, and I used it here only to emphasize the stock-vs-flow point, given how it is commonly phrased).

I realize that this could seem needling, but it really did throw me off, and made me miss the valid point that was being made.

We have added more to a few sentences in the third section of the results (bottom of page 9, and in the next paragraph on page 10) to clarify Reviewer 1's justifiable confusion about the difference between the population of people currently incarcerated, people who were admitted to prison during the pandemic, and people who were released during the pandemic—highlighting the key importance of sentencing disparities among the incarcerated population immediately prior to the pandemic.

2. When talking about the racial composition of B, and how it differs from A and C, the authors may want to note the recent study by the Council on Criminal Justice, which makes a similar point (not about Covid, but about prison pops in general), but notes that a major explanatory reason for this is differences in violent offending and victimization, which complicates the (political) story about the causes here: <https://counciloncj.foleon.com/reports/racial-disparities/national-trends>. It's obviously not the time to include some major new section on this, but it may be worth at least discussing it briefly.

We added a citation and brief discussion of the reference Reviewer 1 provided, which does help our broader discussion quite a bit.

3. For Fig. A.2/Section A2, I'm not entirely sure why the authors say that only Maine, Oregon, and Wyoming don't fit the general trend. I get that those three see *declines* in disparities. But I feel like, for ex, OK/CT/DE (steady rises), or MO/WI (looks like declines as well) don't fit as well. If there is some way to clarify what really makes the three states they highlight uniquely non-conforming (or what pulls OK and MO into the conforming fold) that would be great--but I also think that Fig. A.2 makes the data transparent enough that it is easy for readers to draw their own assessments about the importance of heterogeneity, so this is more an issue of just adding in some clarifying exposition.

This is a good point, and to clarify, we say that Maine, Oregon, Wyoming and Maryland and Missouri do not fit the trend. Nevertheless, Reviewer 1 brought up a good point that we should mention why some of these borderline states are included---if weakly---as conforming to the trend we identify. We have now added this note in the middle paragraph on page 5.

4. Along those lines, it could be that Fig. A.4 provides the evidence I am looking for--I have to admit that I am completely unfamiliar with violin plots and had a hard time understanding what it was saying. That could just reflect my own lack of knowledge of that sort of plot, but it also makes me somewhat concerned that other readers may struggle too. I wonder if there is a way to convey that information in a more graphically-conventional format?

In response to Reviewer 1's concerns as well as the editorial team's concerns, the legend for Figure A.4 (which is now included as A.3) is expanded much more. If possible, we'd like to keep the format (i.e., violin plots instead of transitioning them to box plots) because of broader trends away from under-informative box plots. That said, we acknowledge that more explanation is needed. For one, we add an explanation for how to interpret violin plots in the legend. We then add more information about the statistical test included (two sided t-test, with n=50 degrees of freedom, and exact t- and p-values instead of approximations previously). Lastly, we detail what

each of the components of a violin plot means (i.e., which lines correspond to the distribution's min, max, mean, what the width of the violin means).

5. In Fig. A.5, is it worth noting who the big outliers at the top are as well? And following up on point (3), if A.5 is what makes it clear why the non-conforming listed at the start of Section A.2 are the non-conforming, that wasn't entirely clear to me either. Also, the highlighted states in A.5 isn't the same group as those listed on p. 24 at the start of Section A.2--are these different groups?

In response to Reviewer 1's suggestion, we have added a sentence that highlights what the large outliers in Figure A.5 are (in this version, this figure has become A.4). The legend now includes: "The top four curves on the plot correspond to prison systems that are smaller in size (New Hampshire, North Dakota, Vermont, the District of Columbia), with demographic statistics that can be disrupted by small fluctuations in incarcerated populations."

Referee #2 (Remarks to the Author):

The authors have sufficiently responded to my comments on the previous draft. I recommend that Nature publish the paper and I thank the authors for their careful and thoughtful engagement.

Echoing our response to Reviewer 1: Your comments and recommendations for how to improve our paper truly transformed this work. We appreciate the care that you took with these reviews.

Editorial Note: The editors assessed the responses to R1's remaining comments and, after internal discussion, decided that they had been addressed appropriately.